# Forest fire size amplifies postfire land surface warming

Jie Zhao[1,2,3,10], Chao Yue[3,4,5,10]✉, Jiaming Wang[1], Stijn Hantson[6], Xianli Wang[7], Binbin He[8], Guangyao Li[1], Liang Wang[2], Hongfei Zhao[1] & Sebastiaan Luyssaert[9]

Climate warming has caused a widespread increase in extreme fire weather, making forest fires longer-lived and larger[1-3]. The average forest fire size in Canada, the USA and Australia has doubled or even tripled in recent decades[4,5]. In return, forest fires feed back to climate by modulating land–atmospheric carbon, nitrogen, aerosol, energy and water fluxes[6-8]. However, the surface climate impacts of increasingly large fires and their implications for land management remain to be established. Here we use satellite observations to show that in temperate and boreal forests in the Northern Hemisphere, fire size persistently amplified decade-long postfire land surface warming in summer per unit burnt area. Both warming and its amplification with fire size were found to diminish with an increasing abundance of broadleaf trees, consistent with their lower fire vulnerability compared with coniferous species[9,10]. Fire-size-enhanced warming may affect the success and composition of postfire stand regeneration[11,12] as well as permafrost degradation[13], presenting previously overlooked, additional feedback effects to future climate and fire dynamics. Given the projected increase in fire size in northern forests[14,15], climate-smart forestry should aim to mitigate the climate risks of large fires, possibly by increasing the share of broadleaf trees, where appropriate, and avoiding active pyrophytes.

In recent decades, large forest fires have become more frequent, with a doubling to tripling of the mean fire size in the western USA, Canada and eastern Spain, in response to factors such as climate warming, fuel management and rural depopulation[4,5,16]. Larger fires have been associated with higher burn severity, as measured by greater postfire decrease in vegetation greenness[17] or larger combustion of surface organic carbon[18], suggesting that they result in higher carbon dioxide ($CO_2$) emissions per unit burnt area and thus have a greater biogeochemical climate warming impact.

The biogeochemical effect of $CO_2$ emissions represents only about half of the near-surface climate impact of global forest fires[8], with the remaining half coming from biogeophysical effects due to changes in the land surface characteristics[6]. It remains to be established whether larger fires lead to greater biogeophysical effects that would, in turn, strengthen their biogeochemical effects, or have smaller biogeophysical effects that could (partly) offset their biogeochemical warming. Given that fire size is projected to increase in response to future warming in northern forest ecosystems[14,15], this study aims to enhance our understanding of the biogeophysical implications of increasing fire size.

To this end, we combined satellite-based datasets of global wildfire events for 2003–2016, surface radiometric temperature ($T$), surface albedo ($\alpha$), ecosystem evapotranspiration (ET) and incoming shortwave radiation, to investigate changes in surface energy processes with fire size for between 1 and 14 years after a fire (Methods). Moreover, satellite observations of fire behaviour, fire radiative power (FRP), leaf area index (LAI) and forest loss were used to investigate changes in fire intensity and fire severity with fire size (Methods).

Northern temperate and boreal forests (40° N–70° N) were selected for our research because (1) pronounced postfire surface warming has previously been reported there[8], suggesting a substantial contribution of biogeophysical processes in postfire surface climate impacts; (2) both fire size and burnt area are projected to increase in this region under future climate warming[14,15], potentially making climate feedback effects through fire size important; (3) this region has various forest types with distinct fire regimes[19], potentially leading to different postfire biogeophysical responses to fire size; and (4) wildfires in this region are mostly driven by climate conditions rather than by direct human activities[20], emphasizing the need for appropriate mitigation of large fires.

## Fire-size-dependent postfire surface warming

The study domain shows extensive fires over the period of 2003–2016 (Fig. 1a). Changes in summer (June–August) surface radiometric temperature ($\Delta T$) indicate a widespread warming effect one year after

[1]College of Natural Resources and Environment, Northwest A & F University, Yangling, China. [2]Shandong Provincial Key Laboratory of Water and Soil Conservation and Environmental Protection, College of Resources and Environment, Linyi University, Linyi, China. [3]State Key Laboratory of Soil Erosion and Dryland Farming on the Loess Plateau, Northwest A & F University, Yangling, China. [4]College of Forestry, Northwest A & F University, Yangling, China. [5]Institute of Future Agriculture, Northwest A & F University, Yangling, China. [6]Faculty of Natural Sciences, Universidad del Rosario, Bogotá, Colombia. [7]Natural Resources Canada, Canadian Forest Service, Northern Forestry Centre, Edmonton, Alberta, Canada. [8]School of Resources and Environment, University of Electronic Science and Technology of China, Chengdu, China. [9]Amsterdam Institute for Life and Environment (A-LIFE), Department of Ecological Sciences, Vrije Universiteit Amsterdam, Amsterdam, The Netherlands. [10]These authors contributed equally: Jie Zhao, Chao Yue. ✉e-mail: chaoyuejoy@gmail.com

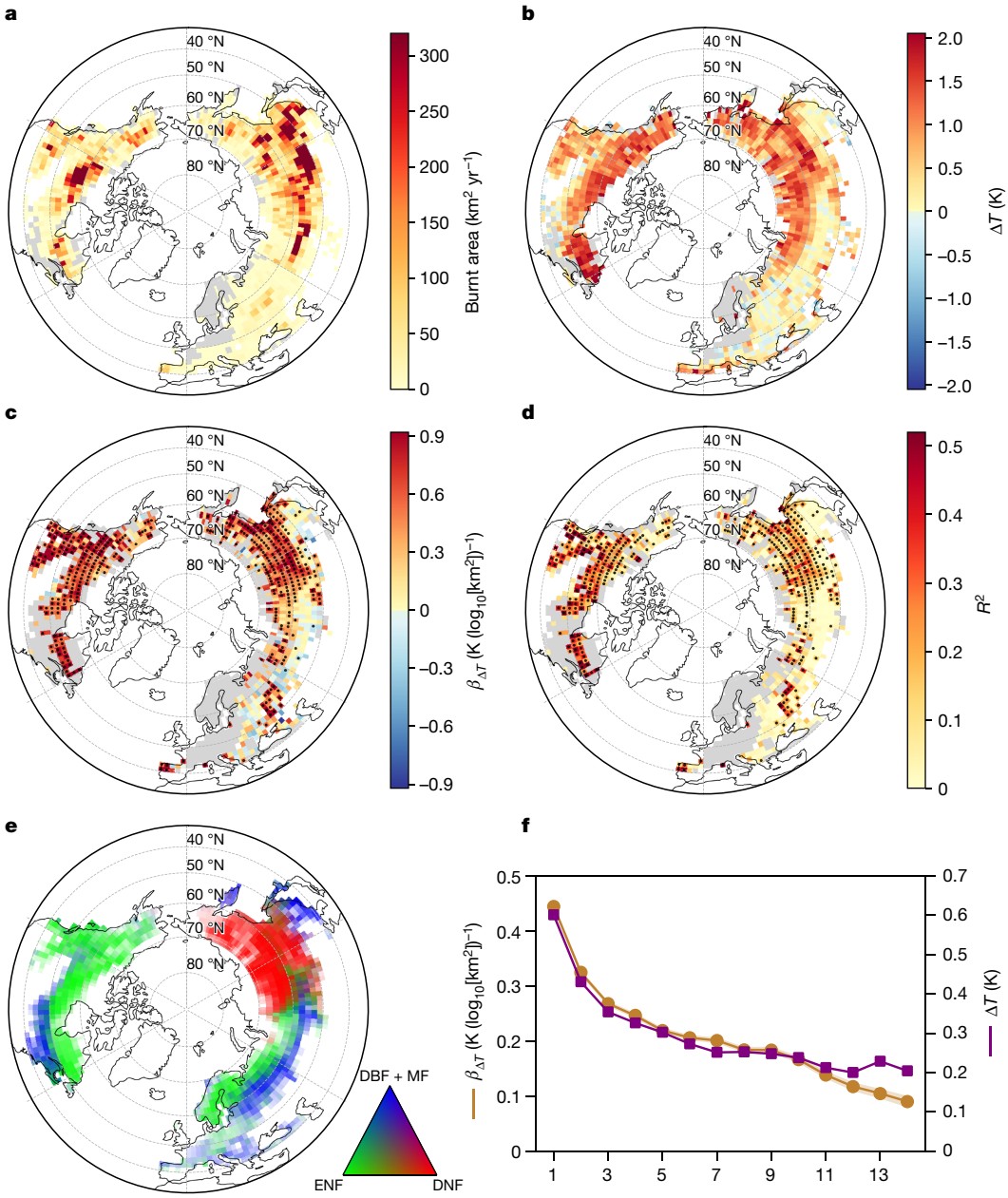

**Fig. 1 | Postfire surface radiometric warming in summer and its amplification by fire size. a**, Annual burnt area over 2003–2016. **b**, Surface radiometric temperature change ($\Delta T$) in summer (June–August) one year after fire. **c,d**, Slope $\beta_{\Delta T}$ and $R^2$ derived by fitting a linear regression model ($\Delta T = \alpha + \beta_{\Delta T} \times \log_{10}[\text{fire size}]$) in the grid cells with a minimum of 10 fires. The solid dots indicate pixels with locally significant regressions ($P < 0.05$, the two-tailed $t$-test), all of which have passed a more rigorous field significance test corrected for the false-discovery rate (FDR; $\alpha_{FDR} = 0.10$; Methods). **e**, Spatial distribution of forest type composition. The colour opacity indicates the proportion of forest area to land area. **f**, Regional mean postfire summer $\Delta T$ (in violet, significantly greater than zero for all years at $\alpha = 0.05$, one-tailed $t$-test, with negligible s.e.m.), and the domain-wide $\beta_{\Delta T}$ value derived by fitting a single linear regression model ($\Delta T = \alpha + \beta_{\Delta T} \times \log_{10}[\text{fire size}]$) across the whole study domain (in orange; $P < 0.05$ for all years, Student's $t$-test; the shaded areas show standard errors), both for up to 14 years after fire. All the maps have a 2° spatial resolution. The light-grey background indicates northern temperate and boreal forests (40° N–70° N) with a >10% ground coverage. Figure developed using Python open-source tools.

fire (Fig. 1b), consistent with previous findings[8]. The effect of fire size on summer $\Delta T$ was investigated by fitting a linear regression model ($\Delta T = \alpha + \beta_{\Delta T} \times \log_{10}[\text{fire size}]$) in each 2° grid cell of the study domain. The widespread positive $\beta_{\Delta T}$ values, which were statistically significant, suggest that fire size amplified postfire surface warming (Fig. 1c,d). High $\beta_{\Delta T}$ values were mostly found in North America and eastern boreal Asia where evergreen needleleaf forests (ENF) and deciduous needleleaf forests (DNF) dominate (Fig. 1c,e). By contrast, lower or statistically insignificant $\beta_{\Delta T}$ values were found in western, central and southeastern Siberia and eastern Europe, which are dominated by deciduous broadleaf forests (DBF) and mixed forests (MF) (Fig. 1c,e). This pattern suggests that $\beta_{\Delta T}$ is modulated by forest type, a hypothesis analysed later in this study.

The domain-wide $\beta_{\Delta T}$ value was 0.44 ± 0.01 K $(\log_{10}[\text{km}^2])^{-1}$ (Fig. 1f), showing that, in summer, for every doubling in fire size, postfire land surface becomes warmer by 0.13 ± 0.01 K, or 22% of the regional average warming (0.60 K; Fig. 1f). Both reported summer surface warming ($\Delta T$) and its amplification by fire size ($\beta_{\Delta T}$), although they decayed over time,

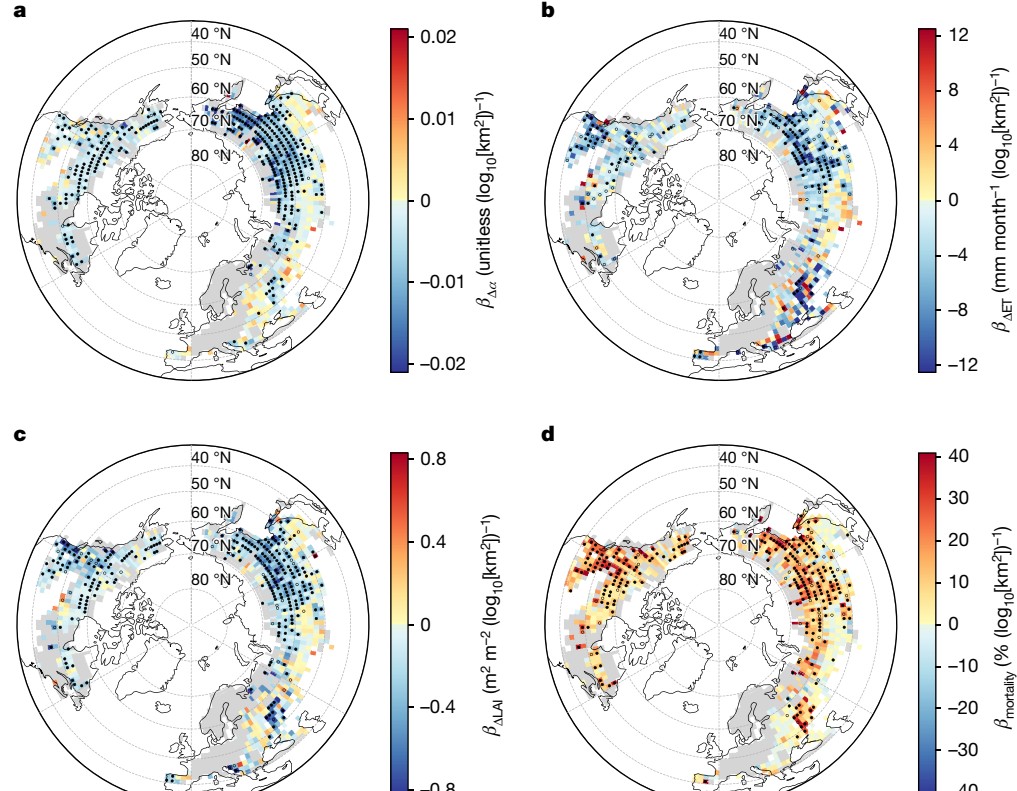

**Fig. 2 | Amplification of fire size on biogeophysical changes after fire.**
**a**–**d**, Linear regression models, of the form $y = \alpha + \beta \times \log_{10}[\text{fire size}]$, were fitted in 2° grid cells with a minimum of 10 fires, where $y$ stands for the changes in summer (June–August) surface albedo ($\Delta\alpha$) (**a**), summer ecosystem ET ($\Delta$ET) (**b**), summer LAI ($\Delta$LAI) (**c**) and forest mortality (as a percentage; **d**), all for one year after fire. Both solid and empty dots indicate pixels with locally significant regressions ($P < 0.05$, two-tailed $t$-test), but the solid dots indicate those having passed a more rigorous field significance test corrected for the FDR ($\alpha_{\text{FDR}} = 0.10$; Methods). Extended Data Figs. 1 and 7 show the spatial distribution of the mean value of postfire changes for these variables. Figure developed using Python open-source tools.

were found to persist for up to 14 years after fire, the longest period with records available (Fig. 1f).

Several studies report a moderate omission error in the burnt area of high-latitude regions by the global-coverage fire patch dataset of the Global Fire Atlas (GFA) used in this study[21]. A comparison of boreal forests of western Canada and Alaska, for which higher-quality fire patch datasets exist, shows that fire size tends to be underestimated by the GFA but the reported amplification by fire size in postfire summer surface warming was robust, with the regional datasets showing even larger $\beta_{\Delta T}$ values and better model fitting than the GFA (Supplementary Note 1).

## Underlying changes in surface energy process

The size-enhanced summer surface warming was driven by systematic changes in land surface energy processes following fire, almost all of which were amplified by fire size (Fig. 2a,b and Extended Data Fig. 1). As a result of the removal of the forest canopy and darkening of the ground surface[22], surface albedo in summer decreased one year after fire in most areas of the study region (Extended Data Fig. 1e), with greater reductions associated with larger fires (Fig. 2a). This caused more shortwave radiation to be absorbed by the land surface as fire size increases (Extended Data Fig. 1g,h). Moreover, decreasing ET with increasing fire size (Extended Data Fig. 1i and Fig. 2b) implies that less of the absorbed radiative energy was released as latent heat (Extended Data Fig. 1k,l), leading to a non-radiative warming effect.

In line with surface warming, outgoing longwave radiation increased after fire, with greater increases following larger fires (Extended Data Fig. 1c,d). The sum of sensible heat and ground heat fluxes, derived

by applying energy conservation, also increased after fire, but the increase with fire size is mostly statistically insignificant (Extended Data Fig. 1m,n), probably because it was calculated as a residual term in the energy budget and therefore integrates the uncertainties in the other variables.

The increase in outgoing longwave radiation following fire due to surface warming in summer exceeds the increased absorption of shortwave radiation, leading to surface radiative cooling, which was also amplified by fire size (Extended Data Fig. 1o,p). The reported land surface warming, despite this radiative cooling, hence indicates the dominance of non-radiative processes, that is, sensible heat, latent heat and ground heat fluxes, rather than the radiative processes, namely, incoming and outgoing short- and longwave radiations, in determining postfire surface temperature change in summer.

In winter (December–February), postfire changes in surface energy fluxes were largely the reverse of those in summer, with the exception of ET and latent heat flux, which showed almost no change (Extended Data Fig. 2). Accordingly, in contrast to summer, $\Delta T$ in winter was dominated by radiative processes, resulting in surface cooling consistent with an increase in surface albedo. Both surface cooling and albedo changes show some dependence on fire size, but this effect is mainly limited to North America (Supplementary Note 2 discusses the snow effect on winter cooling).

On the annual timescale, for the first four years after fire, the change in surface radiometric temperature was dominated by the summer signal, showing size-dependent warming. Afterwards, winter cooling became dominant (Extended Data Figs. 3a and 4a). However, over timescales of up to 14 years after fire, the surface radiative cooling effect persisted, largely as a result of increasing surface albedo in winter over

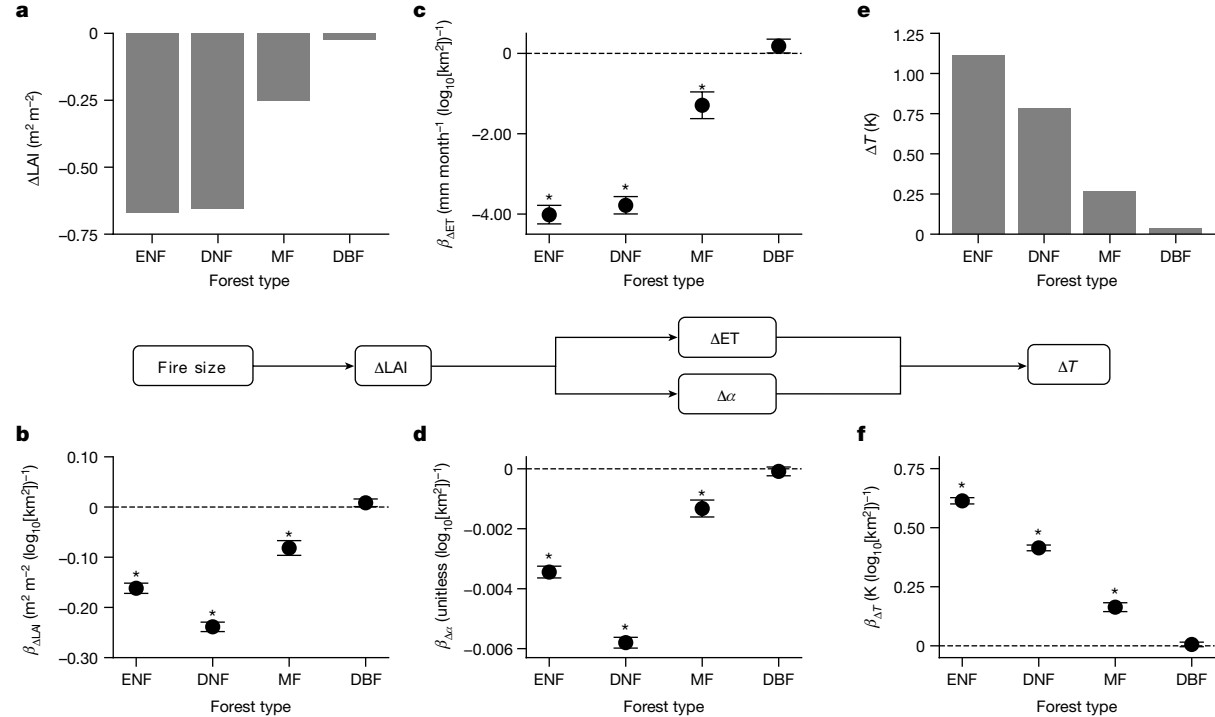

**Fig. 3 | Fire vulnerability and the amplification of fire size on biogeophysical changes after fire, stratified by forest type.** The cascade of biogeophysical drivers of postfire summer (June–August) surface radiometric warming is shown in the middle of the figure. **a,e**, Fire vulnerability is defined as postfire change in LAI (**a**) or surface radiometric temperature ($\Delta T$) (**e**) averaged over all fire patches. For all the forest types, the mean value of $\Delta$LAI (or $\Delta T$) is significantly lower (or greater) than zero at $\alpha = 0.05$ (one-tailed $t$-test), with negligible s.e.m. Tukey's honest significant difference test for multiple comparisons shows significant differences in fire vulnerability among forest types ($P < 0.05$). **b–d,f**, Multiple linear regression models incorporating $\log_{10}$[fire size], forest type and their interactive effects were fitted to derive the amplification effect of fire size for different forest types (that is, the $\beta$ value: the slope between postfire biogeophysical changes and $\log_{10}$[fire size]), using the dependent variables of $\Delta$LAI ($\beta_{\Delta LAI}$; **b**) and $\Delta$ET ($\beta_{\Delta ET}$; **c**), and changes in surface albedo ($\beta_{\Delta\alpha}$; **d**) and surface warming ($\beta_{\Delta T}$; **f**), all for the summer period one year after fire. The asterisks show significant regressions ($P < 0.05$); error bars show the standard errors. All the $\beta$ values show significant differences ($P < 0.05$, Student's $t$-test) among the forest types. Figure developed using Python open-source tools.

time, consistent with previous findings[19,23] (Extended Data Fig. 3h). The reported persistent summer surface warming (Fig. 1b), further supported by field observations[24], hence arises from non-radiative processes that outweigh the radiative processes. The observed amplifying effect of fire size in postfire land surface energy processes, particularly for summer, mostly holds throughout the entire 14-year trajectory (Extended Data Fig. 4).

## Direct size effect and fire behaviour effect

Size-dependent surface warming is related to postfire landscape heterogeneity as a result of the spatial extent of a burnt patch: we term this the direct size effect. This effect can be understood by thinking of a fixed total burn area: when burnt as several smaller fire patches, the resulting landscape will be more heterogeneous, leading to greater surface roughness than that associated with a single large fire. Consequently, the land surface containing small fire patches dissipates more of the absorbed energy as turbulent heat fluxes, leading to less surface warming (Extended Data Fig. 5). Similar mechanisms underlie previously reported lower magnitudes of land surface warming in highly fragmented as opposed to large-scale deforestation areas[25,26].

The direct size effect was further corroborated by our additional analysis using a Canadian dataset of stand-replacing forest fire and clear-cut harvest in mainly ENF (Supplementary Note 3). Following harvesting, similar to the situation after fire, the land surface becomes warmer in summer and the warming magnitude also increases with the harvest patch size. However, the slope between land surface warming and disturbance patch size ($\beta_{\Delta T}$) was significantly larger for fire than for harvest, with the difference being 0.16–0.21 K ($\log_{10}[\text{km}^2])^{-1}$ or 31–33% of the derived $\beta_{\Delta T}$ values for fire. The larger $\beta_{\Delta T}$ value for fire can be ascribed to the decreased surface albedo caused by increased surface charring, which indicates increasing fire intensity or fire severity with fire size (Supplementary Note 3).

Our analysis reveals positive linear relationships between fire size and fire duration (Extended Data Fig. 6a), and between fire size and fire spread rate (Extended Data Fig. 6c). Long fire durations are only possible under prolonged drought conditions, resulting in almost complete drying of the biomass that fuels the fire[27]. Lower fuel moisture enables more complete burning and also means that less energy is required for adjacent conductive fuel heating, resulting in faster fire propagation, as more energy is available for convective heating and radiation[28].

FRP measures the radiative energy released during combustion, making it a proxy for fire intensity[29]. The observed positive relationship between FRP and fire size (Extended Data Fig. 6e), thus, indicates greater fire intensity with larger fires. At the same time, both the decrease in change in LAI ($\Delta$LAI) and drop in forest mortality following fire (Extended Data Fig. 7) increased with fire size (Fig. 2c,d), suggesting increasing fire severity with size, as reported previously[17]. The independence of the algorithms used to derive fire size, fire intensity and fire severity datasets provides robust evidence for the co-varying fire behaviour effects that contribute to increasing postfire surface warming with fire size.

The correlation between fire behaviour and fire size implies that the effects of fire behaviour are implicitly accounted for when using fire size as the major explanatory variable to examine biogeophysical responses after fire (Supplementary Note 4).

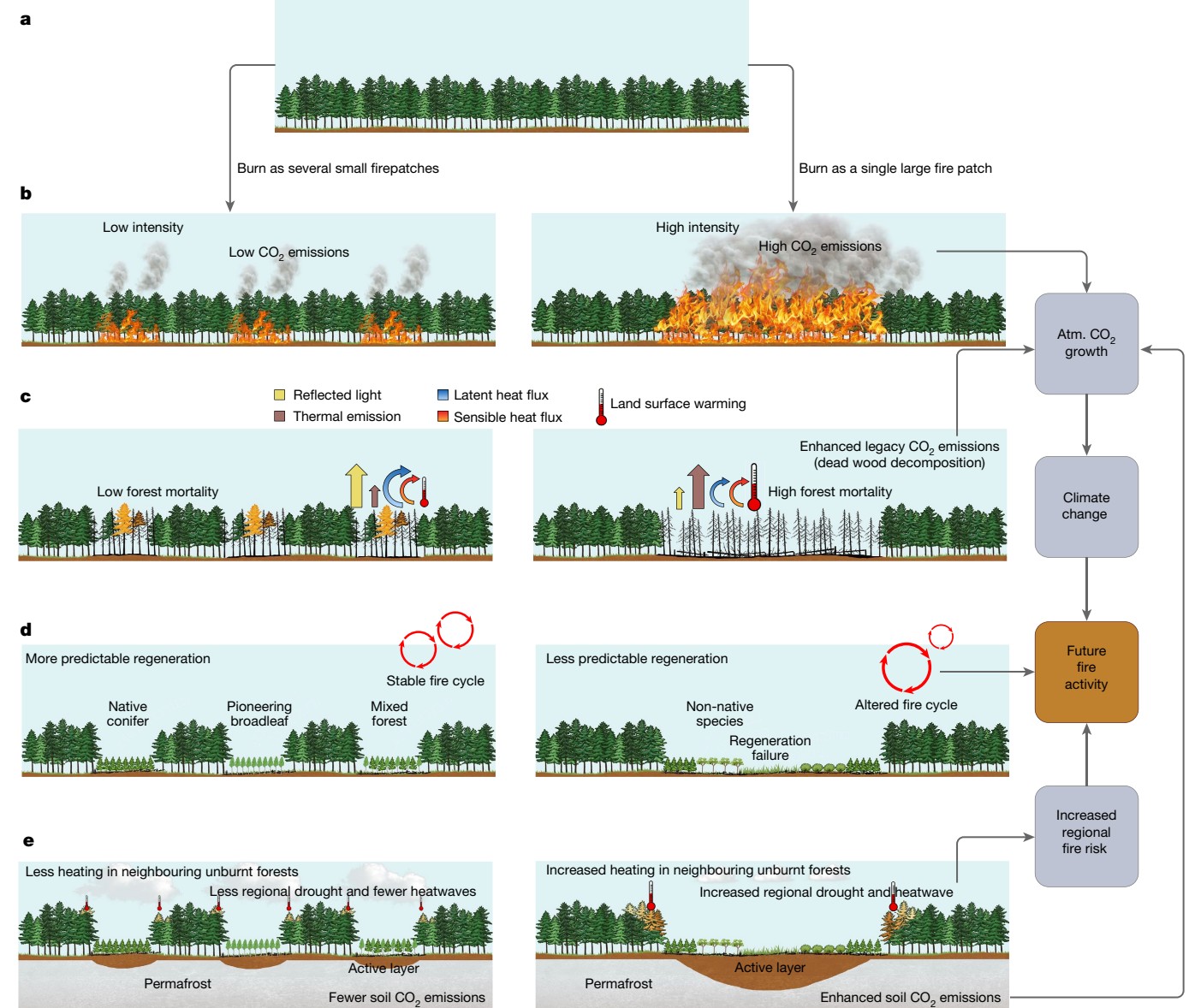

**Fig. 4 | Implications of increasing fire size on future climate and fire activity. a**, Prefire state using boreal ENF as an example, followed by burning the same total area as several small fires (left column) or a single large fire (right column). **b–e**, Effects of fire size on $CO_2$ emissions (**b**), postfire biogeophysical processes (**c**), stand regeneration (**d**) and regional- to global-scale climate feedback loops (**e**). Processes linked to fire size in **b** and **c** are reported in this study, whereas those in **d** and **e** are based on the literature[48–50]. Figure developed using Microsoft PowerPoint software.

## Forest type modulates size-dependent warming

As hypothesized above (Fig. 1c), forest type was found to modulate size-dependent warming. In general, the amplification strength of fire size in postfire biogeophysical responses is consistent with fire vulnerability across different forest types (Fig. 3). Postfire decrease in LAI was the greatest in ENF, slightly lower in DNF and much lower in MF, and almost negligible in DBF (Fig. 3a). The pattern of ΔLAI across different forest types was consistent with those of fire size, fire behaviour and postfire forest mortality (Extended Data Fig. 8).

Accordingly, $\beta_{\Delta LAI}$, the slope between ΔLAI and fire size, was also the highest in ENF and DNF, followed by MF and DBF (Fig. 3b). Differences in $\beta_{\Delta LAI}$ propagate consistently into differences in $\beta_{\Delta ET}$ and $\beta_{\Delta \alpha}$ (Fig. 3c,d), ultimately leading to different size sensitivities to postfire surface warming (that is, $\beta_{\Delta T}$; Fig. 3f). This pattern is also consistent with that of surface warming magnitude across different forests, that is, summer ΔT is the highest in ENF, followed by DNF and MF, with negligible warming in DBF (Fig. 3e).

The above results were confirmed by an alternative analysis based on the share of broadleaf-containing forests rather than on a discrete classification of forest types (Extended Data Fig. 9). The modulation of forest type on fire size sensitivity of postfire biogeophysical responses remained for up to a decade after fire (Supplementary Fig. 1). These findings justify proposing species selection to mitigate the surface climate impacts of growing forest fire size in boreal forests where appropriate.

## Management implications

Recent decades have seen a worldwide increase in extreme fire weather[1], with its influence already looming in temperate and boreal regions. In the forests of Canada, the USA and Australia, both mean and extreme fire size have increased by 65–450% over the past four to six decades, along with an increase in burnt area (Extended Data Fig. 10). Given that the reported amplified land surface warming by fire size also holds for the central to western continental USA and Australia (Supplementary

Figs. 2 and 3), the implications of growing fire size on surface climate may extend beyond northern forests to other regions where climate warming can increase large fires (Australia was included in this part of the analyses only to make this point). Although fire-control spending has increased in both Canada and the USA (Supplementary Fig. 4), growing burnt area and fire size there highlight the difficulty of preventing large fires in sparsely populated areas, which are mostly climate driven[4,30] and possibly exacerbated by historical fire exclusion[31].

Our observation that an increasing share of broadleaf forests weakens size-dependent postfire surface warming provides additional support for earlier proposals[9,32] to increase broadleaf species in boreal forests to mitigate climate warming and forest fires, because passive pyrophyte broadleaves are less prone to fire and have a higher surface albedo than needleleaf species. Considering that about two-thirds of boreal forests, mainly in their warmer and wetter southern areas, are under human management[33], increasing broadleaf species at the time of stand regeneration seems a practical alternative to active fire suppression. For example, strips of coniferous forests could, after being harvested, be replaced by broadleaf trees, which could then serve as firebreaks to reduce fire spread in an otherwise homogeneous coniferous landscape[10]. Moreover, field- and remote-sensing-based observations have shown that the ongoing intensification of fire regime in North American boreal forests has led to an increased abundance of deciduous broadleaves[34,35] at the cost of evergreen conifers, resulting in a natural control on fire frequency[36,37].

Nonetheless, given that the fire mitigation effect of broadleaf species is mostly documented for North American crown fire systems[10,38] and plausible in the Eurasian dark taiga where crown fires dominate[39], the effect of increasing broadleaves on fire frequency in light taiga, where surface fires dominate, requires further research. Furthermore, caution should be taken—or careful local-scale planning is needed—to increase the share of broadleaves in areas where broadleaf forests are a risk for human-dominated spring fires[40], or where permafrost stability is partially maintained by the protective thick organic layer beneath needleleaf forests[41].

## Feedback to future climate and fire activity

The increasing fire size and the associated biogeophysical responses reported in this study have a range of implications for future climate dynamics and fire activity (Fig. 4). On the local scale, the individual density and species assembly of postfire recruitment depend on propagule dispersal distance[11,42], which further depends on fire size[43]. Enhanced soil warming[24] following large fires may differentially alter seedling growth of different tree species, leading to transitions in forest species composition[12]. Lower soil moisture[44] following fire, combined with surface warming, can even render complete regeneration failure[45,46]. These processes influencing stand regeneration may affect the local fire cycle for decades (Fig. 4d).

The reported enhanced surface warming following larger fires was also found to occur over nearby unburnt forests (Supplementary Fig. 5), contributing to regional surface warming. The decade-long persistent warming after fire, combined with reduced ET, might reduce the cloud cover[47] and facilitate the development of droughts and heatwaves, thereby elevating fire risk on a regional scale (Fig. 4e).

On the global scale, higher fire intensity and forest mortality associated with larger fires imply greater $CO_2$ emissions either through direct combustion (Fig. 4b) or indirectly through postfire dead-wood decomposition[48,49] (Fig. 4c). In boreal regions, postfire soil warming also implies enhanced permafrost thaw and soil organic carbon decomposition[50] (Fig. 4e). Both processes contribute to global atmospheric $CO_2$ growth and the resultant climate change, promoting positive climate-fire feedback.

However, postfire surface radiative cooling and atmospheric radiative cooling induced by fire-emitted aerosols also cool the climate[7],

adding complexity to the overall impact of growing fire size on climate change. Nonetheless, a projected increase in the occurrence of large fires under future climate warming[14,15] in forests in the Northern Hemisphere might push these fire-related biophysical and climate processes out of their historical boundaries, making the prevention of large fires and the mitigation of their climate risks imperative to forest management.

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

# Methods

## Datasets of forest fire, land surface characteristics, surface energy and water fluxes

**Fire events.** Information on individual fire events was obtained from the GFA database[51] for the period of 2003–2016. This dataset was constructed using the daily 500 m MODIS Collection 6 MCD64A1 burnt area product[52]. The GFA dataset has been validated against fire event data derived using medium-resolution (30 m) satellite images and forestry agency statistics for the continental USA[51]. Fire size, fire duration and fire spread rate are provided in the GFA and have been reported to be largely in agreement with the validation data, although with a slight underestimation[51]. The GFA has a minimum fire size of 0.25 km² but to avoid observational uncertainties associated with small fire patches, only fires larger than 1 km² (roughly four burnt pixels in the 500 m MODIS dataset) were included in our analysis, ultimately leaving 84,510 fires.

A global, systematic evaluation of the MCD64A1 burnt area data showed that over the boreal forest, the product has an omission error of 27%, which is roughly compensated for by a commission error of 24%, but in temperate forest, it omits a large proportion of fires[53]. Additional studies show that over boreal Eurasia and over Canada and Alaska, MCD64A1 underestimated the burnt area by about 20% (refs. 21,54). Given that our research domain (40° N–70° N) is mostly covered by boreal forest and that our focus is on how postfire biogeophysical processes respond to changes in fire size rather than changes in burnt area, the mapping accuracy of MCD64A1 was considered acceptable. Moreover, to the best of our knowledge, the GFA is the only openly available fire patch dataset spanning multiple years and covering the entire spatial domain of our research area, simultaneously providing information on fire duration and fire spread rate—all these features make it a suitable product for this study. The spread rate for a given fire patch was calculated as the mean value of the spread rates for all the underlying 500 m burnt pixels of a given patch[51].

Nonetheless, the omission of the burnt area by MCD61A1 in boreal regions probably leads to the underestimation of fire size, which is confirmed by comparing the GFA with three regional fire patch datasets of higher quality than the GFA for Canadian and Alaskan boreal forests (Supplementary Note 1). The first two regional datasets cover both Canadian and Alaskan forests and were generated by the Arctic-Boreal Vulnerability Experiment (ABoVE) project by combining fire perimeter information from the Alaskan Interagency Coordination Center and Natural Resources Canada with either burn severity information derived from Landsat images (ABoVE-Landsat) or with the burn date information from the MODIS MCD14ML active fire product (ABoVE-MODIS). The third dataset covers Alaska only and was obtained from the Monitoring Trends in Burn Severity (MTBS) project, which is based on Landsat images. These three datasets are not subject to the omission/commission errors associated with MCD64A1. The major conclusion of this study—that the forest fire size amplifies postfire surface warming—was tested using these high-quality fire patch datasets and found to be robust (Supplementary Note 1).

The GFA is also known to erroneously divide a single fire patch spanning two MODIS tiles (each tile covering a 10° × 10° region) into two patches. However, our analysis revealed that fire patches subject to this error accounted for negligible percentages of the total number of fires and total burnt area (Supplementary Note 5). The analysis in this study was complemented by a comparison of the results based on the GFA with those based on another MCD64A1-based fire patch dataset (GlobFire), which is free of tiling error. This comparison showed consistent results derived using the two datasets, confirming the negligible influence of the tiling error in the GFA on our key findings (Supplementary Note 5).

**Fire intensity.** FRP for 2003–2016 was obtained from the MCD14ML active fire product[55], which contains the geographical coordinates of individual pixels with a nominal resolution of 1 km, where ongoing active burning is detected. FRP denotes the radiative energy emitted per unit time during burning over the area of the observed active fire pixel and has been widely used as a proxy for fire intensity[56]. An active fire pixel is considered to belong to a fire event if its centre is located in a 500 m buffer zone of the event perimeter and if its detection date lies between the start and end dates of the concerned fire event.

Although an active fire pixel has a nominal area of 1 km², its actual area increases as the satellite sensor view moves away from the nadir. Hence, the view angle information was used to calculate the actual area of a given active fire pixel, over which the given FRP was measured. More specifically, the area of an active fire pixel was calculated as the product of the along-scan (image-scanning direction) pixel length and the along-track (satellite movement direction) pixel length, as described previously[57,58]. The mean FRP per area (MW km⁻²) of a fire event was finally obtained by dividing the total FRP of all the active fire pixels in a fire patch by their total area.

**Forest type.** The spatial distribution of forest type was derived from the 300 m global land cover maps for 2002–2015 produced by the European Space Agency (ESA) Climate Change Initiative (CCI) land cover project. To match the resolution of the GFA dataset, the 300 m land cover maps were resampled to 500 m resolution using the nearest-neighbour method. All the land cover types belonging to the 'tree cover' type in the original classification system were treated as a forest. On the basis of the resampled 500 m ESA CCI land cover data, the annual proportions of forest area to total land area (including all the land cover types except for water bodies) were calculated for each 2° grid cell for 2003–2016, followed by the calculation of the multiannual mean proportion (shown as colour opacity in Fig. 1e). Four types of forest were distinguished: ENF, DNF, DBF and MF (a mixture of broadleaf and needleleaf). The relative proportions of ENF, DNF and MF + DBF to forest area over each 2° grid cell were calculated to show the spatial pattern of forest type composition across the study domain (Fig. 1e).

The dominant forest type for each fire event was determined as follows: for an event to be classified as dominated by ENF, DNF or DBF, at least 50% of its 500 m burnt area pixels must be covered by that specific forest type. If the proportion of any forest type is less than 50%, but the total area of all the forest types covers more than 50% of fire event area, then the event is classified as MF.

**Forest mortality.** The Global Forest Change dataset (version 1.6) was used to obtain forest mortality following fire. This dataset provides annual forest loss information derived using Landsat data at a 30 m resolution for 2003–2018 (ref. 58). Ratios of the number of 30 m Landsat pixels undergoing forest loss to the total number of 30 m forest pixels (defined as those with a tree cover of >15% for 2000, also provided in the Global Forest Change dataset) of each individual 500 m burnt pixel were calculated, and were further averaged to obtain the forest mortality for each fire event.

Extended tree mortality, occurring up to ten years after fire, has been reported in field observations from temperate and boreal forest biomes[59–61], but most mortality events occur in the first three years after fire. Satellite-based forest mortality derivation must account for such extended mortality events. However, salvage logging might lead to the false attribution of forest loss to fires. To balance the omission error due to neglecting long-term tree mortality and the commission error of potentially including salvage logging events, fire-induced forest mortality was defined as including forest deaths in the year of fire and one year after fire (a two-year time window). Although a three-year time window, that is, including forest deaths up to two years after fire and in the year of fire, was used in a previous study[8], we found using three- or two-year time windows yielded negligible differences in the derived forest mortality, but accounting for forest loss only in the year of fire occurrence reduced the estimated mortality

by a half (Supplementary Figs. 6 and 7). Nonetheless, all the mortality rates derived using the time windows of different lengths significantly increase with fire size (Supplementary Figs. 6 and 7).

**LAI.** The MODIS Collection 6 MOD15A2H LAI product with a 500 m spatial resolution and an 8-day time step for 2002–2017 was used. All the observations available at the 8-day time step were used, with no gap filling for the missing data. The quality of this dataset has previously been assessed using ground-based LAI measurements and by comparison with other LAI products[62,63].

**Land surface radiometric temperature.** The 8-day MOD11A2 Collection 6 daytime (10:30) and night-time (22:30) land surface radiometric temperature (LST) product for 2002–2017 was used. This product was derived from the MODIS sensor onboard the Terra satellite and has a spatial resolution of 1 km. Only the available observations with a reported error less than 2 K were used, with no gap filling for the missing data. The dataset was resampled to 500 m resolution using the nearest-neighbour method to match the 500 m spatial resolution of the MCD64A1 burnt area data underlying the GFA fire event dataset. The observed daytime and night-time surface temperatures were averaged to obtain daily surface temperature (the equivalents of Fig. 1b–d but for daytime and night-time LST are shown in Supplementary Fig. 28).

**Surface albedo.** The MODIS Collection 6 MCD43A3 surface albedo product at a 500 m resolution for 2002–2017 was used. The 16-day temporal resolution of this dataset provides an appropriate trade-off between the availability of sufficient angular samples and the temporal stability of the surface albedo[64,65]. All the observations available at the 16-day time step were used, with no gap filling for the missing data. White-sky albedo, derived by integrating a bidirectional reflectance distribution function over both incoming and outgoing hemispheres and independent of illumination or atmospheric conditions[22], was used in this study. Specifically, the white-sky albedo for the total shortwave spectrum (0.3–5.0 μm) was used.

**Shortwave radiation.** Incoming shortwave solar radiation was used to calculate changes in absorbed shortwave radiation by the land surface due to postfire changes in surface albedo. Monthly gridded shortwave radiation ($SW_{in}$) at 1° spatial resolution for 2003–2016 from the global surface radiative flux data of CERES_EBAF_Edition4.1 was used. This dataset was obtained from the NASA Clouds and the Earth's Radiant Energy System (CERES). The coarse resolution of the shortwave radiation data is consistent with our research aim of examining the local biogeophysical effects of fires while ignoring fire-induced cloud-regime changes and their associated effects on incoming shortwave radiation.

**Ecosystem ET.** The MODIS Collection 6 MOD16A2 ET product (2002–2017) with a 500 m resolution and an 8-day time step was used. This product is based on the Penman–Monteith equation and uses meteorological reanalysis data (for example, surface air temperature, surface air pressure, humidity, radiation and fraction of photosynthetically active radiation) and land surface characteristics (for example, land cover, LAI and albedo) from MODIS products as inputs[66]. ET was defined as the total water loss to the atmosphere as an integration of soil evaporation, wet canopy evaporation and plant transpiration[66].

**Snow cover.** The daily MODIS Collection 6 MOD10A1 snow cover product at a 500 m resolution for 2004–2017 was used to explore the effect of snow on postfire biogeophysical changes in winter (Supplementary Note 2). Snow cover was detected using the normalized difference snow index. All the observations available at the daily time step were used, with no gap filling for the missing data.

## Analysis of postfire biogeophysical responses to fire size

This study quantified the impacts of forest fire on biogeophysical variables staring from 1 year to up to 14 years after fire, in summer (June–August), winter (December–February) and on annual timescales. The mean values for summer, winter and the annual value of these variables were first calculated by averaging the observations with the respective time steps of different variables, over the corresponding time lengths of summer, winter and all year round (annual), followed by the application of the space-and-time approach to derive fire impacts (described below).

**Quantifying postfire biogeophysical responses.** Fire impacts on various biogeophysical variables (namely, LAI, LST, surface albedo and ET) were quantified as the change in variable the year after fire compared with the year before fire. The MCD64A1 burnt area data underlying the GFA dataset were used to derive postfire biogeophysical impacts for each 500 m burnt pixel. A space-and-time approach was used to isolate fire impacts by subtracting the background variation from the gross change over time. This method has been demonstrated to be effective in quantifying changes in land surface temperature due to fire disturbance or forest cover change[67–69].

Taking LST as an example, the space-and-time approach assumes that for a given burnt pixel, the gross change in LST between the years after and before fire ($\Delta T_{gross}$) is the sum of the LST change induced by fire ($\Delta T$) and a residual change ($\Delta T_{res}$) caused by interannual large-scale climate variations (equation (1)):

$$\Delta T_{gross} = \Delta T + \Delta T_{res}. \tag{1}$$

It, therefore, follows that

$$\Delta T = \Delta T_{gross} - \Delta T_{res}. \tag{2}$$

In equation (2), $\Delta T_{gross}$ can be readily obtained for a burnt pixel, and $\Delta T_{res}$ can be approximated as the mean $\Delta T_{gross}$ of the unburnt control pixels and neighbouring pixels, and is considered comparable to the burnt pixel. The identification of an unburnt control pixel requires the following four conditions to be satisfied.

First, the candidate control pixel needs to be in an area of $50 \times 25$ km$^2$ (over a grid of longitude × latitude, corresponding approximately to $100 \times 50$ pixel$^2$ in the 500 m MODIS dataset) surrounding the target burnt pixel. Second, the candidate control pixel must not have been burnt during the same year as the target burnt pixel nor in the following year. Third, it must be covered with the same forest type as the burnt pixel. Fourth, to exclude potential impacts on $\Delta T_{res}$ by disturbances other than fire, for example, harvest or wind damage, the cumulative tree cover loss (as a percentage) of the candidate control pixel during the year when the target pixel burnt and the following year cannot exceed 20%.

The size of the local window ($50 \times 25$ km$^2$) used above was determined following several previous studies that used a similar approach to quantify surface radiometric temperature change following forest cover change or fire disturbance (for example, a window of $50 \times 50$ km$^2$ was used in ref. 68; a window of $50 \times 28$ km$^2$ was used in ref. 8; and a window of $25 \times 25$ km$^2$ was used in ref. 70). Alternative search windows of $25 \times 25$ km$^2$ and $50 \times 50$ km$^2$ were also used to quantify the linear relationship between postfire summer surface warming and $\log_{10}$[fire size]. The derived slopes were almost identical for the different window sizes (Supplementary Fig. 8), indicating that our results are robust across a fourfold change in search window size.

Fire-induced LST change at fire event level is defined as the average of the $\Delta T$ values of all the burnt pixels for a given event. Fire effects on LAI (ΔLAI), albedo (Δα) and ET (ΔET) were calculated following a similar approach.

**Resolving changes in land surface energy fluxes after fire.** Changes in surface radiometric temperature following fire are the outcome of changes in the surface energy balance, which can be expressed as

$$\Delta R_n = \Delta SW_{in} - \Delta SW_{out} + \Delta LW_{in} - \Delta LW_{out} = \Delta LE + \Delta(H + G), \quad (3)$$

where $\Delta R_n$ is the change in net radiation that measures the surface radiative impact of fires; $\Delta SW_{in}$, $\Delta SW_{out}$, $\Delta LW_{in}$ and $\Delta LW_{out}$ are changes in the incoming and outgoing shortwave and longwave radiation, respectively. $\Delta LE$ and $\Delta(H + G)$ are changes in latent heat flux and changes in the sum of sensible heat flux and ground heat flux, respectively. The change in the energy storage of the land surface was not included.

This analysis considered only local effects on surface temperature due to forest fires by comparing burnt pixels with their surrounding unburnt pixels. Hence, feedback effects such as fire-induced cloud-regime changes (which mainly occur beyond local scales) were not considered. This simplification leads to the assumption that no changes occurred in $SW_{in}$ or $LW_{in}$ due to fire, and therefore, the change in net radiation can be expressed as

$$\Delta R_n = -\Delta SW_{out} - \Delta LW_{out}. \quad (4)$$

$\Delta SW_{out}$ can be derived by combining the surface albedo change ($\Delta \alpha$) and $SW_{in}$:

$$\Delta SW_{out} = \Delta \alpha \times SW_{in}. \quad (5)$$

As fire influences surface temperature, outgoing longwave radiation also changes. $LW_{out}$ can be calculated from the surface radiometric temperature ($T$) and emissivity ($\varepsilon$) using the Stefan–Boltzmann law:

$$LW_{out} = \varepsilon \sigma T^4, \quad (6)$$

where $\sigma$ is the Stefan–Boltzmann constant ($5.67 \times 10^{-8}$ W m$^{-2}$ K$^{-4}$).

The MOD11C3 product provides surface emissivity estimates for the middle and thermal infrared bands that can be used to obtain $\varepsilon$ using an empirical equation[71]:

$$\varepsilon = 0.2122\varepsilon_{29} + 0.3859\varepsilon_{31} + 0.4029\varepsilon_{32}, \quad (7)$$

where $\varepsilon_{29}$, $\varepsilon_{31}$ and $\varepsilon_{32}$ represent the surface emissivity for MODIS bands 29 (8,400–8,700 nm), 31 (10,780–11,280 nm) and 32 (11,770–12,270 nm), respectively.

$\Delta LW_{out}$ can then be approximated by the first-order differential of equation (6). But given its strong nonlinearity, a distinction is made between changes in daytime and night-time LST:

$$\Delta LW_{out} = \frac{1}{2}\sigma[(T_{day}^4\Delta\varepsilon + 4\varepsilon T_{day}^3\Delta T_{day}) + (T_{night}^4\Delta\varepsilon + 4\varepsilon T_{night}^3\Delta T_{night})], \quad (8)$$

where $\Delta T_{day}$, $\Delta T_{night}$ and $\Delta\varepsilon$ refer to postfire changes in daytime and night-time surface temperatures and surface emissivity, respectively. A single value of $\Delta\varepsilon$ was used for both daytime and night-time because only a daily value is provided in the MOD11C3 product.

The latent heat flux (LE; unit: W m$^{-2}$) can be derived from ET (unit: mm day$^{-1}$) as

$$LE = \rho L_v \times ET, \quad (9)$$

where $\rho$ is the density of water (a constant value of $1.0 \times 10^3$ kg m$^{-3}$ was used) and $L_v$ is the latent heat of vapourization for water (a constant value of $2.5 \times 10^6$ J kg$^{-1}$ was used). The conversion coefficient between ET (mm day$^{-1}$) and LE (W m$^{-2}$) can be calculated using equation (9) as 28.94 W m$^{-2}$ (mm day$^{-1}$)$^{-1}$. Therefore, the change in latent heat flux ($\Delta LE$) is given by

$$\Delta LE = \Delta ET \times 28.94 \text{ W m}^{-2} \text{ (mm day}^{-1})^{-1}, \quad (10)$$

Finally, changes in the sum of the sensible heat flux and ground heat flux can be derived as

$$\Delta(H + G) = -\Delta SW_{out} - \Delta LW_{out} - \Delta LE. \quad (11)$$

Note that all the terms on the right side of the above equation have their associated observation errors that we have not explicitly addressed here, which are then included in the term on the left side.

**Linear relationships between biogeophysical changes and fire size.** Simple linear regressions of the form $y = \alpha + \beta \times \log_{10}[\text{fire size}]$, between postfire biogeophysical responses ($\Delta LAI$, $\Delta\alpha$, $\Delta ET$ and $\Delta LST$) and changes in land surface energy fluxes and fire size, were conducted to detect the impact of fire size on postfire biogeophysical processes. The derived regression slope of fire size is defined as fire size sensitivity (the $\beta$ value). Following the observation that fire size fits a log-normal distribution in different ecosystems around the globe[72], $\log_{10}[\text{fire size}]$ was used as the independent variable. Such a logarithmic transformation of fire size or burnt area is widely used in forest fire studies[17,30,73]. The minimum fire size required for a fire event to be included in the regression was set at 1 km$^2$ (that is, four 500 m MODIS burnt pixels), to remove possible observational uncertainties associated with small fire patches.

Furthermore, simple linear regressions between fire size and fire behaviour variables (fire duration and fire spread rate), and between fire intensity, forest mortality and fire size, were also performed to examine changes in fire behaviour and fire severity with fire size.

For all the above variables, linear regressions with $\log_{10}[\text{fire size}]$ were fitted in each 2° cell with a minimum of 10 fires to display the spatial pattern of the $\beta$ value and the possible influence of forest type (Figs. 1c,d and 2, Extended Data Fig. 6 and Supplementary Fig. 7). Moreover, we applied a more rigorous field significance test corrected for the FDR ($\alpha_{FDR} = 0.1$) following ref. 74. For observations showing moderate to strong spatial autocorrelation, the global significance level ($\alpha_{global} = 0.05$) is about half that of $\alpha_{FDR}$ (ref. 74). The test corrected for FDR was carried out using the stats.multitest module in the statsmodels Python package based on the Benjamini–Hochberg method. Pixels with locally significant regressions ($P < 0.05$) and those passing the field significance test corrected for FDR are shown in our results.

For changes in LST and land surface energy fluxes after fire, a single linear regression with fire size, including all fires in the study domain, was also performed to obtain the domain-wide $\beta$ values (Fig. 1f and Extended Data Fig. 4). A multiple linear regression model with interaction terms between fire size and forest type was used to test whether significant differences exist in the $\beta$ values among different forest types (Fig. 3). The forest type was treated as a categorical variable with its values being ENF, DNF, MF or DBF. A two-tailed $t$-test for the interaction term was used to examine whether significant differences existed in the regression slopes (namely, $\beta$ values) among the forest types ($P < 0.05$). The values of the determination coefficient ($R^2$) of multiple linear regressions are as follows: $\Delta LAI = 0.23$, $\Delta ET = 0.12$, $\Delta\alpha = 0.20$ and $\Delta T = 0.31$.

Multiple linear regression models using postfire change in LST as the dependent variable, and $\log_{10}[\text{fire size}]$, $\Delta LAI$, fire intensity (FRP) and forest mortality as independent variables, were conducted both in each 2° grid cell and by including all fires in the study domain, to examine the direct size impact after accounting for co-varying fire behaviour variables with size (Supplementary Note 4.1 and Supplementary Fig. 23a).

Fire vulnerability was defined in terms of the mean values of postfire changes in LAI and LST, and forest mortality of all fires. The Tukey's honest significant difference test for multiple comparisons was used to test significant difference in fire vulnerability among forest types (Fig. 3).

We examine the impact of the edge effect, where a linear relationship between postfire surface warming and fire size would be potentially driven by a decreasing fraction of the patch area located near its edge as its size increases. This analysis shows that the reported enhanced postfire surface warming (and other biogeophysical variables) with fire size is unlikely to be an artefact of the edge effect (Supplementary Note 6).

**Long-term biogeophysical responses after fire and their changes with fire size.** The space-and-time method used to quantify biogeophysical and surface energy flux changes one year after fire (that is, $\Delta T$, $\Delta ET$, $\Delta \alpha$ and changes in surface energy fluxes) and the linear regressions of these variables with $\log_{10}$[fire size], as described above, were extended for up to 14 years after fire—the longest period with records available. Control pixels that were used to derive the residual changes (namely, $\Delta T_{res}$ in equation (2) and for other similar variables) were limited to neighbouring pixels with the same forest type that did not experience any burning and had a cumulative forest cover loss of <20% for the period from the year of fire occurrence up to the analysis year.

### Recent trends in fire size and burnt area
Historical data for forest burnt areas and the number of fires for Canada, the USA and Australia were collated from a variety of sources (Supplementary Table 1).

For Canada, the fire event dataset consisted of point data retrieved from the Canadian National Fire Database (NFDB). This dataset includes the location and size of each individual fire event for 1960–2020. The NFDB, maintained by Natural Resources Canada, is derived from various Canadian fire management agencies including provinces, territories and Parks Canada. This dataset has been widely used in the analysis of long-term fire regime changes in Canada[4,75].

For the USA, the burnt perimeter dataset for 1984–2020 from the MTBS project was used. The MTBS project mapped the fire extent and size of all large fires (that is, greater than 1,000 acres or 4.05 km$^2$ in the west and greater than 500 acres or 2.02 km$^2$ in the east) over the conterminous USA, Alaska and Hawaii[76]. This dataset has been previously used to analyse historical trends of large fires in the USA[5].

We included Australia in our analysis, despite its location in the Southern Hemisphere, because its forests are prone to fire and can be largely classified as temperate. For Australia, polygon datasets of wildfire events (excluding prescribed fires) for 1981–2020 were obtained from the government agencies of land management and fire suppression for New South Wales, Western Australia, South Australia, Victoria, Queensland and Tasmania (Supplementary Table 1). These datasets have been extensively used in studies addressing the response of fire regimes to climate change in Australia[77,78]. The vegetation type for each fire polygon was determined based on the location of its centroid and its associated EAS CCI land cover type for 2001 to ensure that only forest fires were included.

Although the Canadian fire event data (NFDB) contain very detailed fire records including small fires (<2 km$^2$), in some agencies, the reporting of such fires only started at around 1980 (ref. 4). Moreover, previous studies have shown that small fires, although numerous, make up only a small fraction of the total area burnt[79]. To ensure consistency, only forest fires larger than 2 km$^2$ for Canada, USA and Australia were included in our analysis of temporal trends for fire size and burnt area.

Temporal trends in annual mean fire size, extreme fire size (95th percentile) and burnt area (Canada, 1960–2020; USA, 1984–2020; Australia, 1981–2020) were calculated using the Mann–Kendall test for trend and the Theil–Sen slope estimator (Extended Data Fig. 10), both of which are robust to potential observation outliers[1]. The predicted values for the start and end years were obtained from the fitted Theil–Sen regression. The relative change (%) in mean fire size and extreme fire size was calculated as the difference in the predicted value between the end and start years divided by the predicted value for the start year.

Further details are available in the Supplementary Information and refs. 80–86.

## Data availability

All data supporting our results are publicly available. All MODIS data are available at https://lpdaac.usgs.gov/. GFA data are available at http://www.globalfiredata.org. GlobFire data are available at https://gwis.jrc.ec.europa.eu/. FireCCILT11 data are available at ftp://anon-ftp.ceda.ac.uk/neodc/esacci/fire/data/burned_area/AVHRR-LTDR/pixel/v1.1/. FireCCI51 data are available at ftp://anon-ftp.ceda.ac.uk/neodc/esacci/fire/data/burned_area/MODIS/pixel/v5.1/. Global Forest Change data are available at http://earthenginepartners.appspot.com/science-2013-global-forest/download_v1.6.html. CERES_EBAF-Edition4.1 data are available at https://asdc.larc.nasa.gov/project/CERES/CERES_EBAF_Edition4.1. The ESA CCI global land cover products are available at http://maps.elie.ucl.ac.be/CCI/viewer/index.php. The open-source Python package cartopy, available on GitHub at https://github.com/SciTools/cartopy.git, was used for plotting all the maps. The datasets used to generate all the figures in the Article and its Supplementary Information are publicly available at Figshare (https://doi.org/10.6084/m9.figshare.21443799).

## Code availability

The scripts used to generate all the figures in the Article and its Supplementary Information are available at Figshare (https://doi.org/10.6084/m9.figshare.21443799).

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

**Acknowledgements** We thank N. Andela for developing the fire event dataset; the MODIS team that provided the core datasets for this analysis; J. Canadell for helping retrieve the Australian fire data; and R. Scholten for comments that improved the quality of this study. C.Y. was funded by the National Key Research and Development Program of China (2023YFB3907403), the National Science Foundation of China (U20A2090), the Second Tibetan Plateau Scientific Expedition and Research Program (2022QZKK0101) and the Strategic Priority Research Program of the Chinese Academy of Sciences (XDB40000000). S.L. was funded by Horizon 2020, HoliSoils (SEP-210673589) and Horizon Europe INFORMA (101060309).

**Author contributions** C.Y. conceived the study. J.Z. and J.W. performed the analyses, with support from G.L., C.Y., S.L. and S.H. The manuscript was written by C.Y. and S.L., with help from J.Z. and other co-authors. All co-authors reviewed and commented on the manuscript.

**Competing interests** The authors declare no competing interests.

**Additional information**
**Correspondence and requests for materials** should be addressed to Chao Yue.

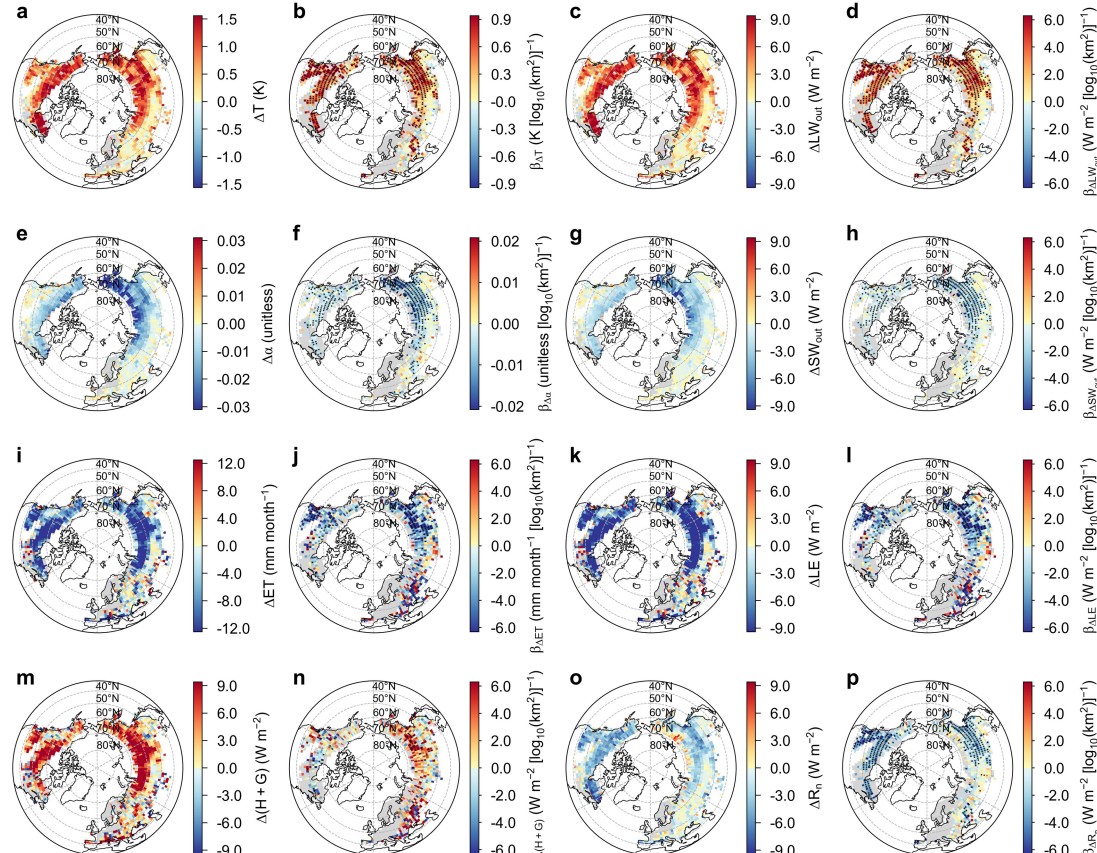

**Extended Data Fig. 1 | Biogeophysical and surface energy flux changes in summer (June–August) one year after fire and their amplification by fire size over northern forests (40°N–70°N).** The first and third columns show the mean values of postfire changes in land surface temperature (ΔT, **a**), outgoing longwave radiation (ΔLW_out, **c**), surface albedo (Δα, **e**), reflected shortwave radiation (ΔSW_out, **g**), ecosystem evapotranspiration (ΔET, **i**), latent heat flux (ΔLE, **k**), the sum of sensible and ground heat fluxes (Δ(H + G), **m**) and net radiation (ΔR_n, **n**) by averaging all fires within each 2° grid cell. The second and fourth columns show linear regression slopes (β) derived by fitting a linear regression model (y = α + β × log₁₀(fire size)) for each 2° grid cell with more than 10 fires. In the regression model, y stands for ΔT (**b**), ΔLW_out (**d**), Δα (**f**), ΔSW_out (**h**), ΔET (**j**), ΔLE (**l**), Δ(H + G) (**n**), and ΔR_n (**p**). Both solid and empty dots indicate pixels with locally significant regressions (*p* < 0.05, the two-tailed t-test), but solid dots indicate those having passed a more rigorous field significance test corrected for the false discovery rate (α_FDR = 0.10, see Methods). The light grey background in all maps indicates northern forests with a > 10% ground coverage. Figure developed using the Python open-source tools.

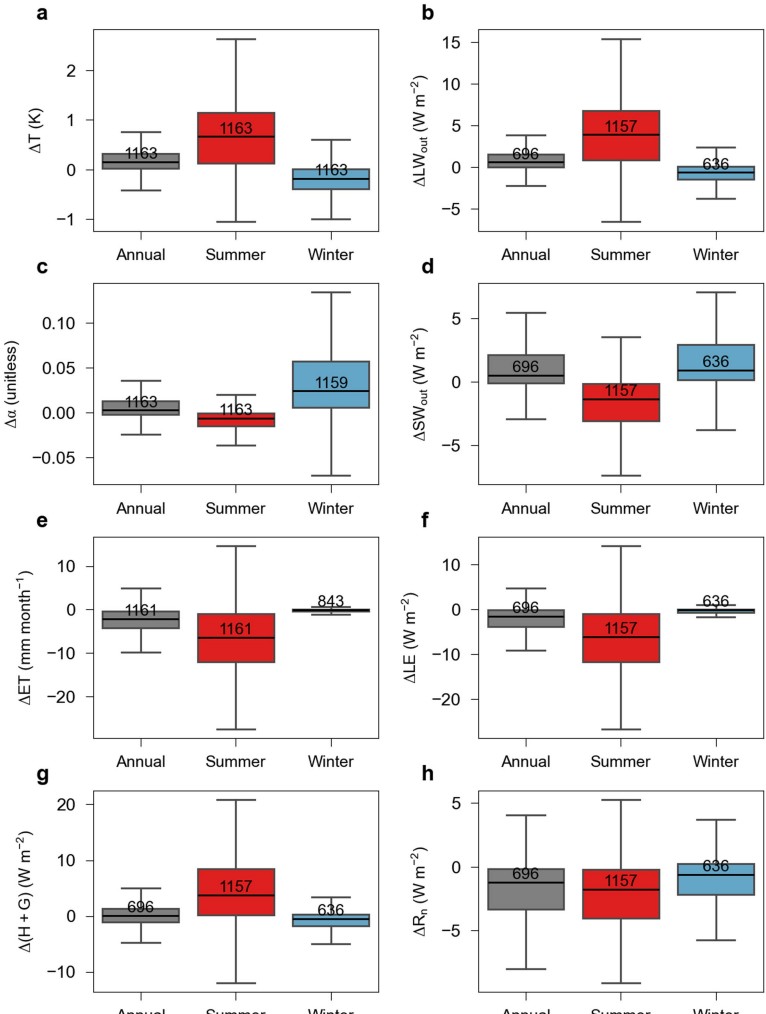

**Extended Data Fig. 2 | Postfire biogeophysical and surface energy flux changes for summer (red, June–August), winter (blue, December–February) and the annual time scale (gray) one year after fire for northern forests (40°N–70°N).** Postfire changes in land surface temperature ($\Delta T$) (**a**), outgoing longwave radiation ($\Delta LW_{out}$) (**b**), surface albedo ($\Delta\alpha$) (**c**), reflected shortwave radiation ($\Delta SW_{out}$) (**d**), ecosystem evapotranspiration ($\Delta ET$) (**e**), latent heat flux ($\Delta LE$) (**f**), the sum of sensible and ground heat fluxes ($\Delta(H + G)$) (**g**), and net radiation ($\Delta R_n$) (**h**) were derived for each 2° grid cell with more than 10% forest coverage. Boxplots show the statistical distributions of the values for 2° grid cells. The center line of the boxplots represents the median value, with the box limits indicating upper and lower quartiles and whiskers showing 1.5 × interquartile range. Figure developed using the Python open-source tools.

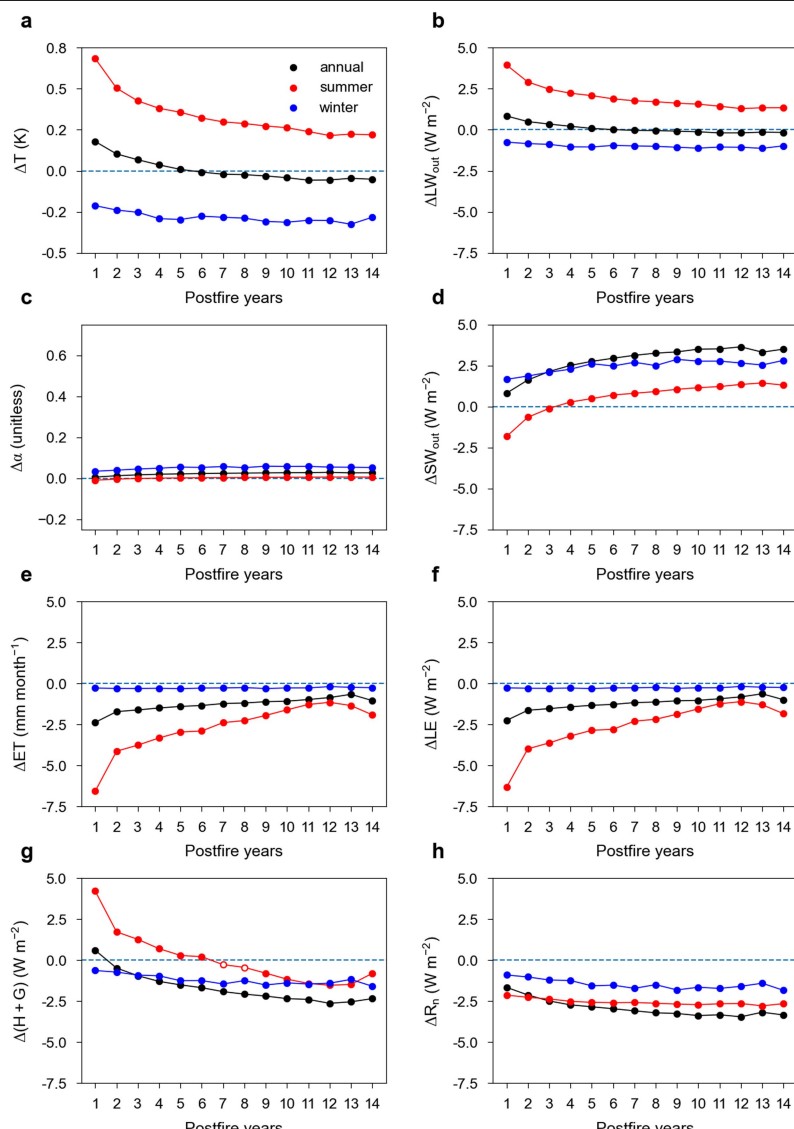

**Extended Data Fig. 3 | Long-term biogeophysical and surface energy flux changes for up to 14 years after fire for northern forests (40°N–70°N).** Postfire trajectories of the mean values of all fire patches are shown for changes in land surface temperature ($\Delta T$) (**a**), outgoing longwave radiation ($\Delta LW_{out}$) (**b**), surface albedo ($\Delta\alpha$) (**c**), reflected shortwave radiation ($\Delta SW_{out}$) (**d**), ecosystem evapotranspiration ($\Delta ET$) (**e**), latent heat flux ($\Delta LE$) (**f**), the sum of sensible and ground heat fluxes ($\Delta(H+G)$) (**g**) and net radiation ($\Delta R_n$) (**h**). The red, blue and black dots represent the regional mean values for summer (June–August), winter (December–February) and the annual time scale, respectively. Solid dots represent mean values significantly different from zero ($p < 0.05$, two-tailed t-test), whereas empty dots represent those not significantly different from zero ($p > 0.05$, two-tailed t-test). The values of s.e.m. for all variables are too small to be visible and hence error bars showing s.e.m. are omitted. Figure developed using the Python open-source tools.

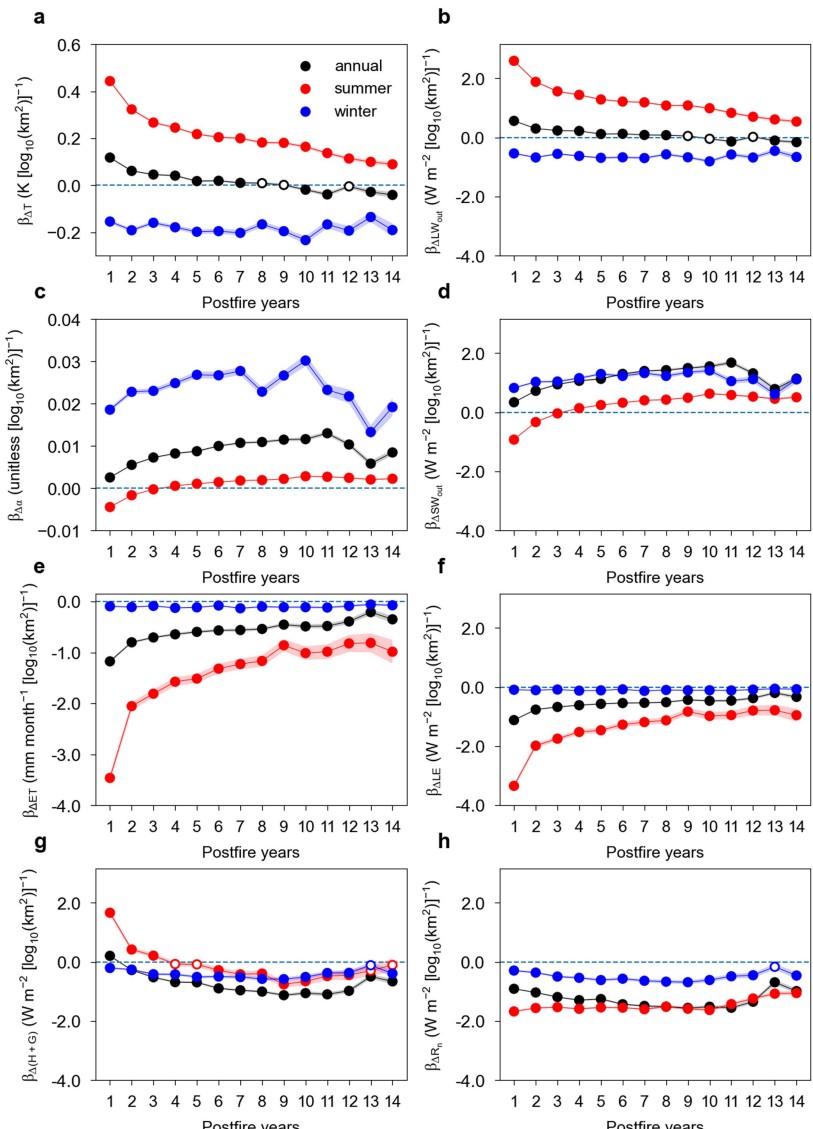

**Extended Data Fig. 4 | The effect of fire size on different postfire biogeophysical and surface energy flux changes for summer (red, June–August), winter (blue, December–February) and the annual time scale (black) for up to 14 years in northern forests (40°N–70°N).** Linear regression models ($y = \alpha + \beta \times \log_{10}(\text{fire size})$) were fitted across the whole study domain, where y stands for postfire changes in land surface temperature ($\beta_{\Delta T}$, **a**), outgoing longwave radiation ($\beta_{\Delta LWout}$, **b**), surface albedo ($\beta_{\Delta\alpha}$, **c**), reflected shortwave radiation ($\beta_{\Delta SWout}$, **d**), ecosystem evapotranspiration ($\beta_{\Delta ET}$, **e**), latent heat flux ($\beta_{\Delta LE}$, **f**), the sum of sensible and ground heat fluxes ($\beta_{\Delta(H+G)}$, **g**), and net radiation ($\beta_{\Delta Rn}$, **h**) for different years after fire. Solid dots represent the β values from significant regressions ($p < 0.05$, the student's t-test) while the empty dots represent insignificant regressions ($p \geq 0.05$). Shading indicates the 95% confidence intervals, which are shown for all variables, although in some cases may be difficult to discern due to the narrow width of the confidence interval. Figure developed using the Python open-source tools.

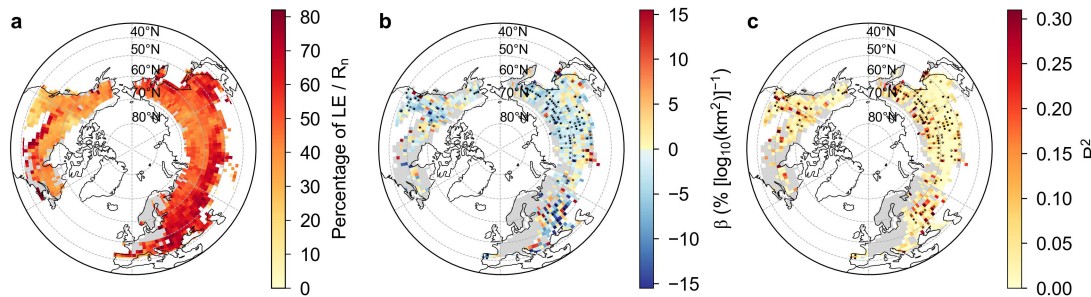

**Extended Data Fig. 5 | The percentage (%) ratio of latent heat flux to net radiation (LE/$R_n$) for the summer (June–August) one year after fire and its amplification by fire size across northern forests (40°N–70°N) at a 2° resolution.** Panel (**a**) displays the mean values of postfire percentage of LE/$R_n$. Panels (**b**) and (**c**) show the regression slope (β) and the coefficient of determination ($R^2$) derived by fitting a linear regression model ($y = \alpha + \beta \times \log_{10}$(fire size)) within 2° grid cells with more than 10 fires, where y represents the percentage of LE/$R_n$. Both solid and empty dots indicate pixels with locally significant regressions ($p < 0.05$, the two-tailed t-test), but solid dots indicate those having passed a more rigorous field significance test corrected for the false discovery rate ($\alpha_{FDR} = 0.10$, see Methods). The light grey background in all maps indicates northern forests with a > 10% ground coverage. Figure developed using the Python open-source tools.

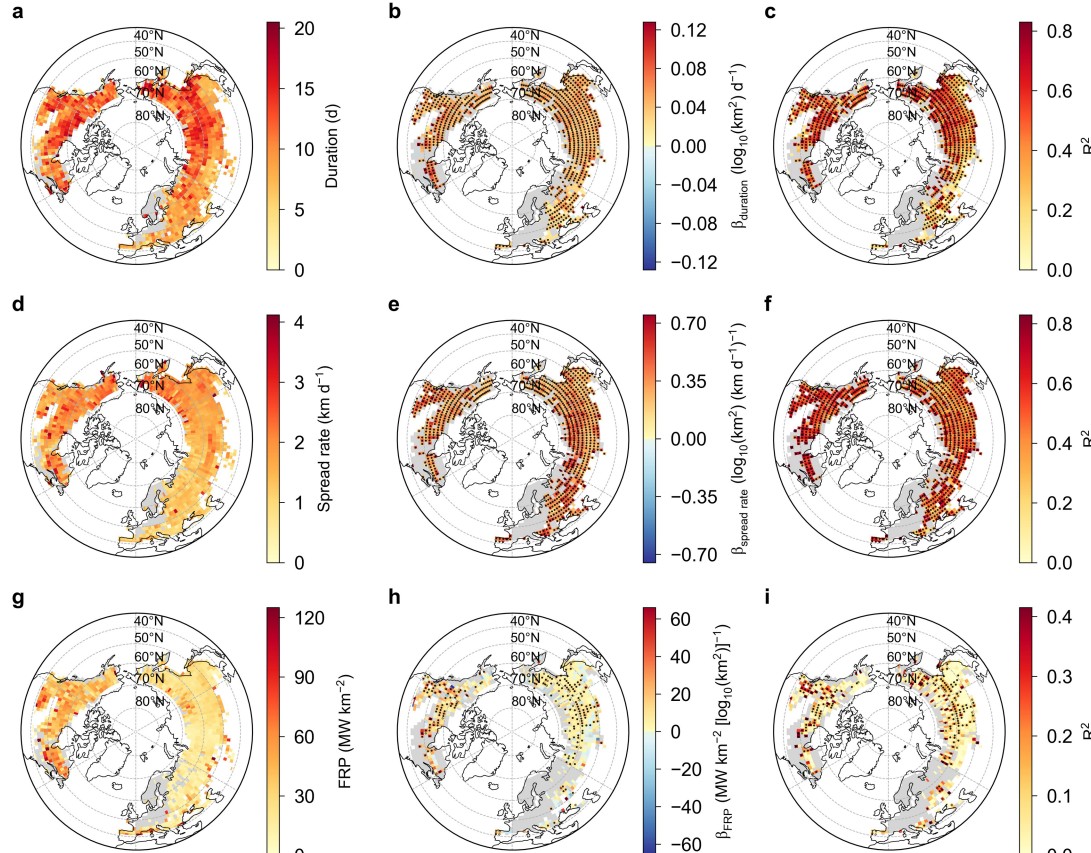

**Extended Data Fig. 6 | Spatial patterns of fire behavior and the linear relationships between fire behavior variables and fire size across northern forests (40°N–70°N) at a 2° resolution.** The first column displays the mean values of fire duration (**a**), fire spread rate (**d**) and fire radiative power (FRP, **g**) by averaging all fire patches within each 2° grid cell. Panels (**b**, **e**) and (**c**, **f**) show the regression slope (β) and the coefficient of determination (R²), respectively, which are derived by fitting linear regression model ($\log_{10}$(fire size) = α + β × x) within 2° grid cells, where x stands for fire duration (**b**, **c**) and fire spread rate (**e**, **f**). Panels (**h**) and (**i**) show the β and R² derived by fitting a linear regression model (FRP = α + β × $\log_{10}$(fire size)) within 2° grid cells. Only grid cells with >10 fire events were included in the regression analysis. Both solid and empty dots indicate pixels with locally significant regressions ($p < 0.05$, the two-tailed t-test), but solid dots indicate those having passed a more rigorous field significance test corrected for the false discovery rate ($\alpha_{FDR}$ = 0.10, see Methods). The light grey background in all maps indicates northern forests with a >10% ground coverage. Figure developed using the Python open-source tools.

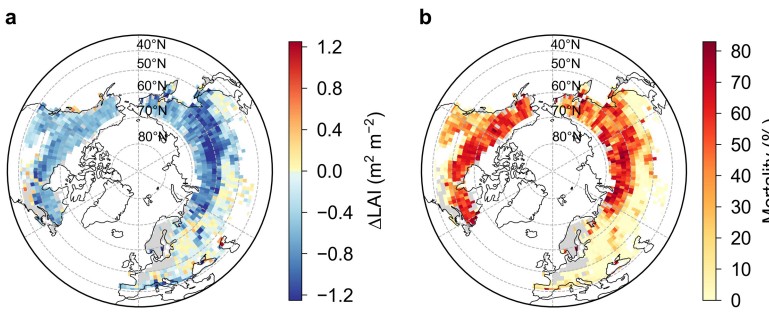

**Extended Data Fig. 7 | Spatial patterns of leaf area index change (ΔLAI) and fire-induced forest stand mortality one year after fire across northern forests (40°N–70°N).** Panels (**a**) and (**b**) display the mean values of summer (June–August) ΔLAI and fire-induced forest stand mortality at a 2° resolution, respectively. The light grey background in all maps indicates northern forests with a > 10% ground coverage. Figure developed using the Python open-source tools.

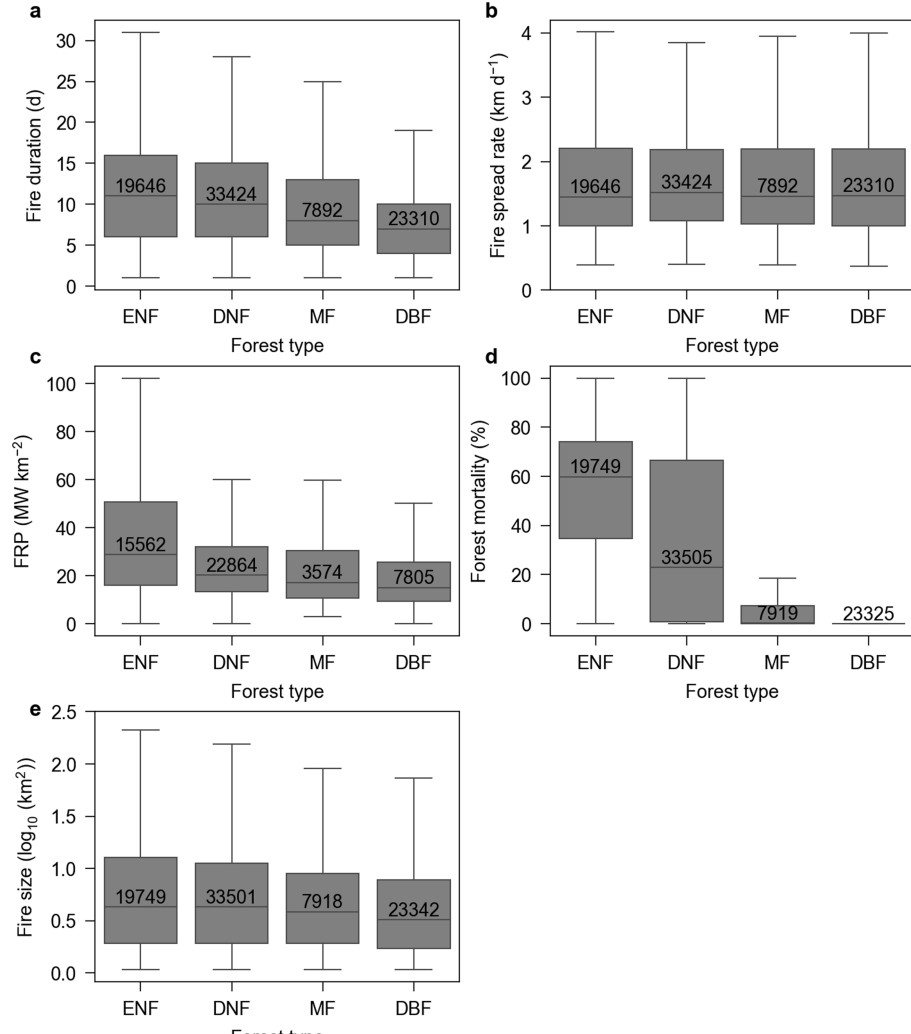

**Extended Data Fig. 8 | Comparing fire behavior, postfire forest mortality, and fire size of fire patches of different forest types for northern forests (40°N–70°N).** Boxplots are shown for forest fire duration (**a**), fire spread rate (**b**), fire radiative power (FRP, **c**), fire-induced forest mortality (**d**), and fire size (**e**) for four forest types: evergreen needleleaf forest (ENF), deciduous needleleaf forest (DNF), MF (mixed forest) and deciduous broadleaf forest (DBF). The center line of the boxplots is the median value, with box limits indicating upper and lower quartiles and whiskers showing 1.5 × interquartile range. The numbers on boxplots show the number of fire patches for each forest type. Figure developed using the Python open-source tools.

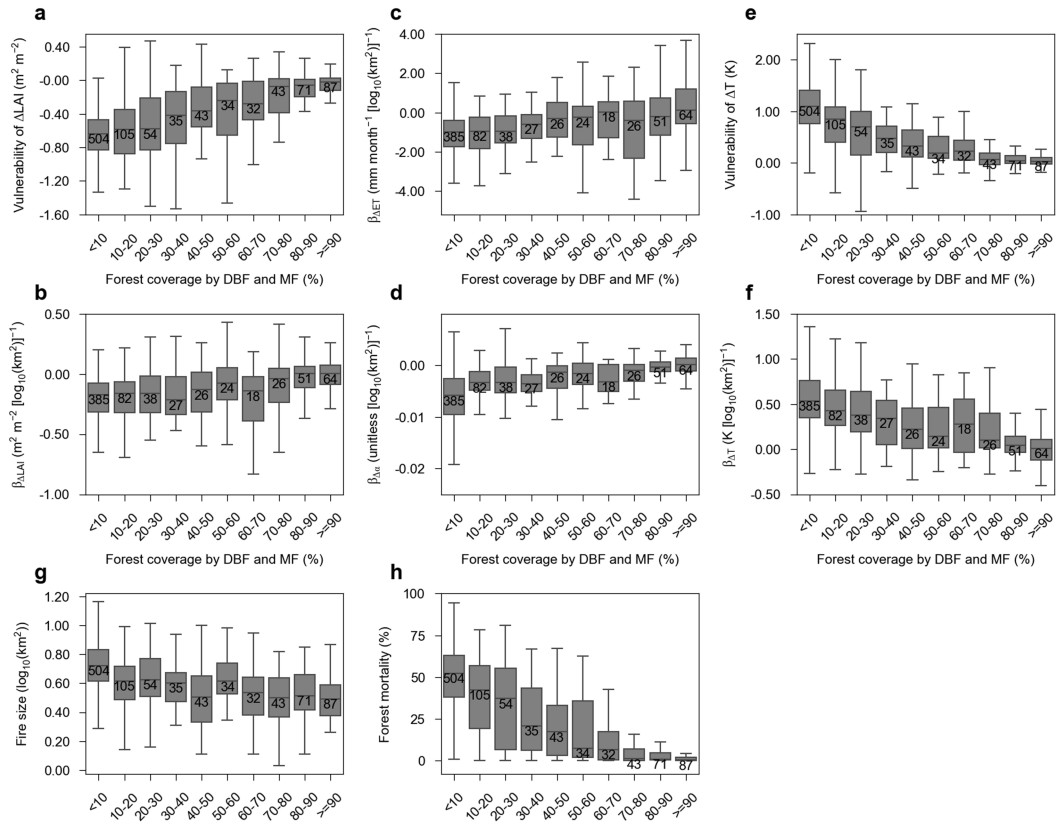

**Extended Data Fig. 9 | Fire size, fire vulnerability and the fire size effect on biogeophysical changes one year after fire, ordered by increasing share of mixed forests and deciduous broadleaf forests for northern forests (40°N–70°N).** Fire vulnerability is defined as the change in LAI (ΔLAI, **a**) and surface radiometric temperature (ΔT, **e**) after fire, and fire-induced forest mortality (**h**). Simple linear regression models (y = α + β × log₁₀(fire size)) were fitted for each 2° grid cell with more than 10 fires to derive the fie size effect (β) on postfire changes in LAI ($\beta_{\Delta LAI}$, **b**), ecosystem evapotranspiration ($\beta_{\Delta ET}$, **c**), surface albedo ($\beta_{\Delta \alpha}$, **d**) and surface radiometric temperature ($\beta_{\Delta T}$, f) in summer one year after fire. Boxplots are shown for fire size (**g**), fire vulnerability (**a**, **e**, **h**) and fire size effect on biogeophysical changes (**b**, **c**, **d**, **f**) using samples of 2° grid cells falling in different bins of DBF and MF forest coverage (in 10% intervals). The number above each box indicates the number of 2° grid cells in each bin. The center line of the boxplots represents the median value, with box limits indicating upper and lower quartiles and whiskers showing 1.5 × interquartile range. Figure developed using the Python open-source tools.

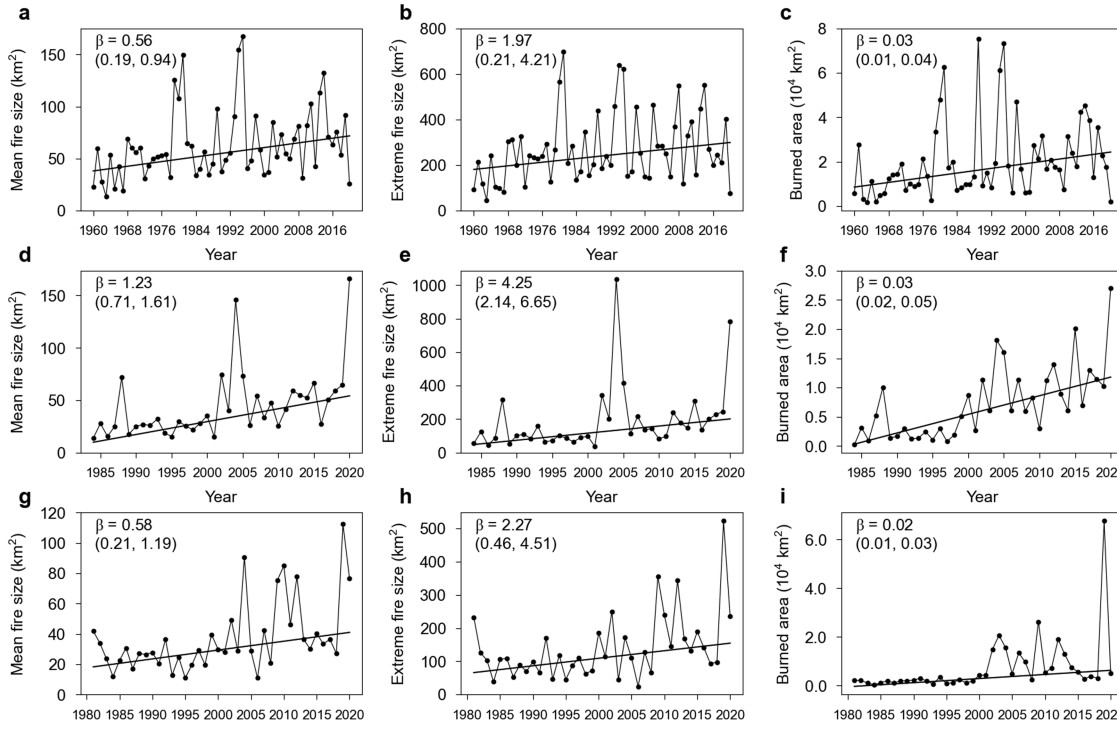

**Extended Data Fig. 10 | Temporal trends in the mean fire size, extreme fire size (95th fire size), and total burned area for all forests in Canada, USA, and Australia.** The first, second and third rows correspond to Canada (**a**–**c**), USA (**d**–**f**), and Australia (**g**–**i**), respectively. The first, second and third columns present mean fire size, extreme fire size, and total burned area, respectively. Significance testing for temporal trends was carried out using the Mann-Kendall test (for Canada, n = 61; for USA n = 37; for Australia, n = 40), with all regressions being significant ($p < 0.05$). β values representing the temporal trends were determined using the Theil-Sen estimator. The numbers in brackets represent the 95th confidence intervals for the derived slopes. The relative growth in the mean (extreme) fire size was quantified as follows: 88% (65%) for Canada over 1960–2020, 448% (317%) for the USA over 1984–2020, and 124% (135%) for Australia over 1981–2020. Figure developed using the Python open-source tools.