## [Peer Review file · Nature]

Manuscript Title: Forest fire size amplifies postfire land surface warming

Editorial Notes:

Redactions – Third Party Material

Reviewer Comments & Author Rebuttals

Reviewer Reports on the Initial Version:

Referees' comments:

Referee #1 (Remarks to the Author):

Overview: This is a useful paper that highlights links between fire size and a host of post-fire impacts that have many implications for fire management and restoration in the face of a warming climate that enables larger fires in temperate and boreal forests. The authors show some of the strongest empirical links that I have seen with 'usually-noisy' fire statistics. The most novel result is the demonstrated link between fire size and post-fire biophysical impact, namely that realized through elevated land surface temperature. Previous studies have codified the general relationship presented here in terms of post-fire land surface temperature changes, including at global scales, and how these relationships change in the years post fire. But not the explicit link to fire size. There are two points of clarification that should be addressed in the main paper given the broader readership and audience, which may result in misuse of the results for those who do not read carefully. First, the current study looks at the immediate (1-year) post fire change, and it should at the very least be clarified in the main text that such effects may not scale into the subsequent years as shown in previous studies. It seems important to justify the use of 1-yr post-fire here and potential limitations. Similarly, the authors refer to surface temperature as LST from remotely sensed data. Many may infer this as near-surface or 2-m air temperature, which is quite different and less sensitive to land-cover. While elevated LST would be reflected in 2-m air temperature, the magnitude should be somewhat less.

Technical Corrections/Considerations

Line 46: Is the word 'half' appropriate here? If the authors numerically intend this to be 50% of the impact, then it works, however, I think this is pretty definitive and that the word half should be softened or changed to "portion".

Line 63, Why is temperate cut off at 40 degree latitude and not further equatorward?

Line 66, Can this log-normal distribution of fires be extended to temperate forests?

Line 74: It would be good to explicitly highlight the approach for that allowed you to perform the analysis on a 2-degree grid.

Line 95: How dependent is this cooling signal in winter on snow cover? Most of the forests here have seasonal snow cover, although the equatorward edge might be a bit different.

Line 106: I might add that you would need critical drought (live/dead fuel moistures receptive to full combustion), but also a prolonged period of fire conducive weather (e.g., continued dry conditions) for such longer-duration fire events.

Line 126, Many of the broad patterns that underlie species distributions are partially dictated by climate. Namely, the ENF are often in places with greater drought stress and receive less precip/more solar in the summer while DBF are more adapted to places with reduced water stress. I don't know if there is an easy way to screen for this, but it might be acknowledged as the advocacy of wide-scale planting of DBF may be short sighted.

Line 154: There is an assumption here that burn severity scales with fire size. Any studies that support or refute this idea might be worth contextualizing here. For example, Cansler and McKenzie 2014 showed a positive relationship between fire size and severity
<https://esajournals.onlinelibrary.wiley.com/doi/full/10.1890/13-1077.1>

Line 168: Perhaps, but in other systems a more frequent fire regime can lead to type conversion and replacement of shrub or grass which might have net C impacts and limited benefits from LST/ET perspective. This might be a system that has higher biogeochemical post-fire effects than biophysical from a climate perspective.

Line 297: It might be useful to remind the reader of the minimum fire size in GFA. I believe this should be ~25ha or so. Since LST has a resolution of 1-km, there could be some artifacts at small scales/small fires where a LST pixel coincides with partially burned and partially unburned land. Can the authors

identify a screening approach to ensure their scaling results are not a function of this? I am more confident the effect they identify is real based on the scaling for larger fires.

Line 338: It might be good to justify the 2-year window given the broad audience of the journal. This could be associated with indirect mortality post-fire, direct fire mortality, and/or time-lag in detection tied with phenological measures

Line 353: Is there a difference between delta daytime vs delta nighttime LST? From the energetic perspective, daytime seems like it would be most impacted and potentially somewhat less downward LW from vegetation at night could have an opposite effect.

Line 370: This is very coarse and will not capture the heterogeneity of cloud cover or aspect slope on downward SW. There are some gridded downward SW data from reanalysis at 0.1 degree from ERA-5 land. However, this is not a very critical part of the analysis and might not matter a lot.

Line 384: This needs to be better stated in the text. The influence on albedo, LST, is likely rapidly fading as vegetation regrows based on other studies on this topic.

Line 402: Is Tres based on a single candidate pixel, or do you find all pixels meeting the criteria here to create an average delta T res?

Line 435: Since the Stefan-Boltzmann equation is very nonlinear, it might also better to use $LW_{out} = (\epsilon \sigma T_{daytime}^4 + \epsilon \sigma T_{nighttime}^4) / 2$

Line 561: How credible are records of number of fire counts in these locations? Fire counts are prone to many uncertainty given the nature of accounting with inhomogeneity in datasets due to a host of factors like duplicates, inclusion of records (e.g., Short et al., 2015) <https://www.publish.csiro.au/wf/WF14190>

Figure 3:

Why are the letters here in these figures? I would expect this information to be self-contained within the figure and caption

The statistics in panels a,b,d,f are a bit confusing and might be unnecessary. Instead, you can perhaps use colors to show statistical significance as this figure is pretty complicated for being black and white.

Figure 4: Does this figure only show results for temperate and boreal in each country (e.g., excludes subtropical forests in Australia/US)? There are forests shown in green equatorward of 40 and unsure if those are counted here.

Figure ED4: The scaling relationship for fire duration, spread rate, and FRP are quite remarkable. I am also interested in any ideas as to why the FRP relationships are substantially stronger in N. America vs Eurasian forests as Fig ED5e,f show these results to be much more heterogeneous.

Figure ED9: The costs here either refer to different actions or have different nomenclature. For example, fire prevention and fire suppression are pretty different items. It would be good to clarify.

Of the broader impacts of the work that might be further emphasized is the effect that elevated LST post-fire for very large fires would pose on tree regeneration in warmer sites. Postfire regeneration often favors cooler/wetter years, and hence another feedback here could be through elevated LST → reduced tree regeneration → reduced C sequestration. Thus, the biophysical feedback could strengthen the biogeochemical feedback. I don't expect this as an additional analysis, but it could be considered as part of the discussion.

Referee #2 (Remarks to the Author):

This manuscript analyzes the relationships between fire size and biogeophysical responses in the northern forest. The authors found that during the summer months immediately following wildfire, the bigger the fire, the stronger its effect on outgoing shortwave radiation, latent heat flux, outgoing longwave radiation, the combination of sensible and ground heat fluxes, land surface temperature, and surface albedo.

Overall, this manuscript uses the method of Alkama, R., & Cescatti, A. 2016. "Biophysical climate impacts of recent changes in global forest cover." It is not surprising that the results of the two studies are similar since fire is one of the forest disturbances. This manuscript focuses only on forest cover loss due to fire in the northern latitude. And while it is still interesting how fires affect land surface temperature,

the authors already published that in 2021 for North American boreal forests. The only difference between the current study and 2021 is the expansion of the study area and mainly analyzing changes in biogeophysical responses in relation to fire size. However, I have a number of concerns regarding the data, method, and interpretation of the results.

First is the quality of the data used in the manuscript. Fire Atlas (Andela et al. 2019), from where fire boundaries were extracted, contains many uncertainties which were not acknowledged in this manuscript. For example, MODIS BA product and derived from it Fire Atlas are not ideal data sources for high latitudes. Chen et al. (2021), in "Missing Burns in the High Northern Latitudes: The Case for Regionally Focused Burned Area Products," reported that "MCD64A1 missed nearly half the total burned areas in the Alaskan and Canadian boreal forests and the tundra during the 15-year period." In Canada and Alaska, the ABOVE dataset based on Landsat is much superior, and it is unclear why the authors did not use the ABOVE product. Additionally, I attached figures with several examples where Fire Atlas cuts fires if they occur between MODIS tiles, therefore, inaccurately calculating the fire size of the individual fires.

Trends in fire size were calculated using OLS, which might not be robust, especially considering that in many regions, the trend can be driven just by one or a few years with extremely large fires (see Fig. 4). Additionally, several studies reported a negative relationship between fire size and human activities. Therefore, it is possible that the increase in fire size due to warming can be offset by increasing population density, which was not addressed in the paper.

Processing artifacts is another of my concerns. The edge-to-fire ratio that is higher for small fires may affect the reported changes in biogeophysical responses in relation to fire size.

Edge pixels that are only partially burned can have less effect on land surface temperature and other biogeophysical responses. I illustrate this in the last figure attached. The fire on the left consists of 9 pixels ("X"), 8 of which are partly burned (blue "X"). The fire on the right consists of 31 pixels, 15 of which are edge pixels. Therefore, the proportion of edge-to-fire pixels for the smaller fire is twice higher compared to a bigger fire. Suppose a fully burned pixel contributes to a certain amount of increase in land surface temperature. In that case, it is reasonable to assume that a half-burned pixel contributes to a smaller increase in land surface temperature. If that is the case, large fires would appear to have a higher climate impact. In reality, the proportion of burned land within fire pixels drives that difference.

Since MODIS BA product misses most of the small fires, it is possible that some of the pixels that are labeled as none burned experienced fire at least partially, which would propagate error into equation (2).

The accuracy of the historical data is also questionable. For China, the authors used 4 different data sources that provided data for random years. How comparable are those data sources? Can they be used together for time-series analysis? Additionally, as the author mentioned, fire size distribution is highly skewed toward small fires, which is why a log scale was used in the analysis. Therefore, it raises the question of why the authors decided to calculate the fire size trend using the mean. I argue that dividing the total amount of burned area by the number of fires is not an appropriate approach to

analyze changes in fire size. The correct way would be to calculate the 95th percentile or median, the two most common approaches in the literature.

Furthermore, why the authors included Australia in the analysis when the study is about “northern forests (40°N–70°N)”?

I found the way the authors presented their results is misleading. The supplementary text is crucial as it shows that the main conclusion is only valid for the summer months. The relationships between fire size and biogeophysical responses are mostly reversed in winter. Annually, those responses are much weaker, if present at all (see Extended Data Fig. 2). Considering that the supplementary text is only a page, and Extended Data Fig. 2 provided far more information than main Fig. 2, it is not clear why the authors did not want to include those results in the main body of the manuscript. Additionally, the authors did not find trends in fire size to be significant in any of their study areas. Therefore, their main conclusion, “The feedback loop in which climate warming results in larger fires and larger fires result in more warming renders the climate impacts by fires beyond the effects of changes in burned area per se.” is not supported by their analysis.

Moreover, “postfire warming” that the authors presented as the main conclusion was only detected during the year immediately after the fire, which was not particularly clear from the abstract. The authors talked about accelerating climate warming due to large fires, while a large body of literature states otherwise, which was not addressed in this manuscript. For example, Chen et al. (2018), in “Strong cooling induced by stand-replacing fires through albedo in Siberian larch forests,” found that “the cooling effect, however, grows rapidly and continuously until the 11th year ..., after which it gradually lessens.” Similarly, in North America, wildfires were found to impose a cooling effect after the fire, according to Randerson et al. (2006, <https://doi.org/10.1126/>), McMillan et al. (2008, <https://doi.org/10.1029/2007gb003038>), Amiro et al. (2006, <https://doi.org/10.1016/j.agrformet.2006.02.014>), Rogers et al. (2015, <https://doi.org/10.1038/ngeo2352>), and Randerson et al. 2012, <https://doi.org/10.1029/2012gl051790>).

Author Rebuttals to Initial Comments:

Overall responses to the referees:

We thank both referees for their efforts reviewing our paper and for their comments which have greatly helped improve the manuscript. Please find below the original referee comments in black and our responses in blue. All the text substantially revised in the manuscript has been marked in blue to make the revisions easier to follow. All figures, tables and line numbers in the following responses refer to those in the revised manuscript, which includes the main text, Extended Data Figures and Tables, Supplementary Text and Supplementary Figures.

Comments by Referee #1

[C1] Overview: This is a useful paper that highlights links between fire size and a host of post-fire impacts that have many implications for fire management and restoration in the face of a warming climate that enables larger fires in temperate and boreal forests. The authors show some of the strongest empirical links that I have seen with ‘usually-noisy’ fire statistics. The most novel result is the demonstrated link between fire size and post-fire biophysical impact, namely that realized through elevated land surface temperature. Previous studies have codified the general relationship presented here in terms of post-fire land surface temperature changes, including at global scales, and how these relationships change in the years post fire. But not the explicit link to fire size. There are two points of clarification that should be addressed in the main paper given the broader readership and audience, which may result in misuse of the results for those who do not read carefully. First, the current study looks at the immediate (1-year) post fire change, and it should at the very least be clarified in the main text that such effects may not scale into the subsequent years as shown in previous studies. It seems important to justify the use of 1-yr post-fire here and potential limitations. Similarly, the authors refer to surface temperature as LST from remotely sensed data. Many may infer this as near-surface or 2-m air temperature, which is quite different and less sensitive to land-cover. While elevated LST would be reflected in 2-m air temperature, the magnitude should be somewhat less.

[Response] We thank the referee for the effort in reviewing our paper and for the positive general comments.

(1) We agree that it’s important to look at the biogeophysical impacts and the associated fire size effects for the subsequent years after fire. Following the referee’s comments, we extended the initial analysis for a period of up to 14 years after fire in the revised manuscript, the longest period allowed by the available data records. The extended analysis found that: a) the reported land surface warming in summer persists for the entire postfire trajectory, but the annual surface temperature effect due to fire shifted from a warming effect to a cooling one about 4 years after fire, as winter cooling started to outweigh the summer warming; b) the

scaling relationships between postfire biogeophysical changes and fire size hold for almost the entire postfire trajectory. The persistent amplifying effects of fire size hence strengthened our initial conclusion, which was based on the results for only one year after fire. Please refer to Section 2.4 in the Methods section (lines 622–632), the revised Fig. 1 and Extended Data Fig. 3, Extended Data Fig. 4, Extended Data Fig. 9, and lines 86–88, lines 120–128 in the main text, and Supplementary Text 2.3 for these results.

(2) To clarify that this study quantified changes in satellite-observed radiometric surface temperature rather than the 2-m air temperature, throughout the whole revised text, we used, where appropriate, the expression ‘surface **radiometric** temperature’ to clearly indicate that we are looking at radiometric temperature inverted by satellite observations. We also added the results of surface radiative cooling effects due to fire to be consistent with previous studies and added a caution regarding the difference between the overall atmospheric cooling through the radiative process and the change in land surface temperature. Relevant revisions include the Method summary in the main text (line 61), and the captions of Fig. 1 (line 322), the caption of Fig. 3 (line 351, line 352), and lines 108–112 in the main text.

Our point-by-point responses to the technical issues raised by the referee are given below.

Technical Corrections/Considerations

[C2] Line 46: Is the word ‘half’ appropriate here? If the authors numerically intend this to be 50% of the impact, then it works, however, I think this is pretty definitive and that the word half should be softened or changed to “portion”.

[Response] We indeed mean “**about** half” here. Liu et al., (2019) reported a 0.0046 K global mean surface warming one year after fire due to the biogeophysical impacts and a 0.0075 K near surface air warming due to the CO₂ emissions. The biogeochemical warming thus accounted for 62% of the total of both effects. This percentage, however, was likely overestimated, because the biogeochemical effects of grassland and savanna fires were included in Liu et al., (2019) but their biogeophysical effects were not. Indicating the exact percentage seemed unnecessary given the context. Based on these considerations, we agree that using “about half” might be better. Hence the original sentence has been changed to: “*The biogeochemical effect through CO₂ emissions represents only **about** half of the near-surface climate impact of global forest fires*” (line 49–50 in the revised main text).

[C3] Line 63, Why is temperate cut off at 40 degree latitude and not further equatorward?

[Response] The cutoff of our research domain was based on following considerations:

(1) First of all, we focused on temperate and boreal forests rather than tropical forests because fires in tropical forests are often deliberately set by humans for the purpose of deforestation (Morton et al., 2008; van der Werf et al., 2008) and hence tropical forest fire size does not necessarily reflect the effects of fire weather and fire behavior. Additionally, in tropical forests, fire detection by the MODIS burned area product is much less effective than the active fire count product (Hantson et al., 2013), causing large omission errors in burned area of as much as -74.1% (Boschetti et al., 2019).

(2) Within global temperate and boreal biomes, the species characteristics of fire-prone forests are quite different between the northern and southern hemispheres. For example, fire-prone temperate forests in Australia are dominated by eucalyptus species that encourage fires by producing inflammable oils and can resprout rapidly after fire (Fairman et al., 2019), whereas in North American boreal forests, fires normally lead to a century-long ecosystem succession, often with species shifts in the mid- or late-successional stages. Forests in boreal North America and boreal Eurasia also have distinct fire regimes. In northern temperate and boreal forests, fire susceptibility also differs among broadleaf and coniferous forests (Astrup et al., 2018). Our second key focus is the difference in postfire biogeophysical responses among different forest types, which makes the most sense in northern temperate and boreal forests, explaining our decision to exclude the southern hemisphere.

(3) Liu et al. (2019) found prominent postfire land surface warming for regions $>40^{\circ}\text{N}$ demonstrating a strong influence of fire on land surface energy processes. Hence, we selected the region of 40°N – 70°N as our study domain.

We noticed that the definition of ‘northern lands’ is a little arbitrary in the literature, but most studies used a separating parallel falling between 30°N and 50°N . For example, Rogers et al., (2015) defined the ‘boreal region’ as land $>40^{\circ}\text{N}$; Piao et al., (2014) defined ‘northern lands’ as land $>30^{\circ}\text{N}$; McGuire et al., (2009) and Hayes et al., (2011) defined ‘pan-arctic region’ or ‘northern high-latitude land’ as land $>45^{\circ}\text{N}$. These examples support our definition of ‘northern temperate and boreal forests’ as land $>40^{\circ}\text{N}$.

That being said, if our research domain was further extended equatorward (as suggested by the referee) but still excluding tropical regions, the largest difference in comparison to our current domain would be that more land in continental US would be included. But we show in the revised manuscript that the reported scaling relationship between postfire surface warming and fire size holds for the whole continental US (Extended Data Fig. 12a in the revised manuscript). Therefore, extending further equatorward will not change our key conclusion.

The above considerations are now briefly described in the revised main text (lines 65-73).

[C4] Line 66, Can this log-normal distribution of fires be extended to temperate forests?

[Response] Generally yes. First of all, the highly right-skewed distribution of fire size was found in all types of ecosystems across the globe, including forests, grasslands, savannas or savanna woodlands, and croplands (Hantson et al., 2015). Besides the log-normal form, another form of power law distribution (a special case of the log-normal one) has also been widely used to fit fire size distribution (Cui and Perera, 2008): both of these require the logarithmic transformation of fire size. However, a recent study (Hantson et al., 2016) showed that the log-normal distribution outperforms the power law one for most regions of the world, suggesting the generality of log-normal distribution. In the revised manuscript, we have changed the original sentence to: “Following the observation that fire size fits a log-normal distribution in different ecosystems across the globe, the logarithm of fire size was used as the independent variable” (lines 574–576 in the revised main text).

[C5] Line 74: It would be good to explicitly highlight the approach for that allowed you to perform the analysis on a 2-degree grid.

[Response] Fig. 1a was based on binning fire patches into different size groups with the average of ΔT being derived for each group. This masked the within-group spread in ΔT . Fig. 1c highlights that the scaling relationship holds even without any grouping. They were basically the same analysis with minor differences. To enhance the clarity of our explanation, we have changed the relevant sentence to: “The scaling relationship between ΔT and fire size held when regressions were repeated within each 2° grid cell but were made at native fire-event level” in the revised main text (lines 89-90).

[C6] Line 95: How dependent is this cooling signal in winter on snow cover? Most of the forests here have seasonal snow cover, although the equatorward edge might be a bit different.

[Response] The snow effect on winter cooling over North America covering 30°N–70°N is now presented in Supplementary Text 2.2 and Supplementary Fig. 4. We have extended the southern border to 30°N, still within the limits of the temperate region, in order to include areas with less snow in winter to better elucidate the role of snow in winter postfire biogeophysical processes. To summarize the results, consistent decreases in $\Delta\alpha$ (albedo change), ΔT and snow cover were observed with decreasing latitude, suggesting a strong regulation by snow in postfire winter surface cooling. Our additional analysis of Australia winter warming following fire and the comparison of its snow cover with northern temperate and boreal forests (40°N–70°N) further supports the crucial role of snow in driving winter cooling in the latter region (Extended Data Table 1; lines 181–183 in the main text).

[C7] Line 106: I might add that you would need critical drought (live/dead fuel moistures receptive to full combustion), but also a prolonged period of fire conducive weather (e.g., continued dry conditions) for such longer-duration fire events.

[Response] We agree with the referee on the raised points. Our original text was: “Long fire durations are only possible under persistent drought conditions, when the biomass that fuels the fire dries almost completely”. The usage of ‘persistent drought conditions’ conveys a similar meaning as in the second point raised by the referee, i.e., ‘a prolonged period of fire conducive weather or continued dry conditions’, whereas the usage of ‘when the biomass that fuels the fire dries almost completely’ conveys a similar meaning as in the first point raised by the referee, i.e., ‘critical drought to dry live/dead fuel moistures receptive to full combustion’. We fully agree with the referee on the importance of clearly elucidating this point, unfortunately, given the length limit, we cannot elaborate it. However, the ‘prolonged period of fire conducive weather’ suggested by the reviewer sounds better than ‘persistent drought’, so we changed ‘persistent drought conditions’ to ‘**prolonged drought conditions**’ in the revised manuscript (line 135).

[C8] Line 126, Many of the broad patterns that underlie species distributions are partially dictated by climate. Namely, the ENF are often in places with greater drought stress and receive less precip/more solar in the summer while DBF are more adapted to places with reduced water stress. I don't know if there is an easy way to screen for this, but it might be acknowledged as the advocacy of wide-scale planting of DBF may be short sighted.

[Response] We suspect that the referee was referring to line 162 rather than 126. There we focus on the boreal forests which account for most of our study domain.

We agree with the referee that climate exerts a broad control over the distribution of forest species because species have evolved various adaptations to survive their respective climate conditions. In boreal Eurasia, for example, the dominance of deciduous larch forests in eastern Siberia is ascribed to their high tolerance of extreme cold and frost drought (Sitch et al., 2003; Beer et al., 2007). On a local to regional scale, where climate conditions are broadly similar, other factors could also play a role. Previous studies show that in the transitional zone in central Siberia, where multiple species coexist, frequent surface fires in larch forests mediate forest species distribution by removing regenerating saplings of evergreen conifer and deciduous broadleaf species (Schulze et al., 2012).

In Canadian and Alaskan boreal forests, a number of studies show that broadleaf species are pioneer, early-successional species after stand-replacing fires (e.g., Johnstone et al., 2010a; Beck et al., 2011; Rogers et al., 2013), the fire regime commonly found in this region. In fact, forest succession and the associated habitat changes are strongly mediated by fire in North American boreal forests (Johnstone et al., 2010a). Mature ENF forests can accumulate a thick,

acid organic layer of dead plant materials over mineral soil that maintains a high level of soil moisture, preventing fire occurrence in normal climate conditions but also impeding the germination of DBF seeds. When fires do occur in drought years, the organic layer is (partially) removed by fire, exposing the mineral soil surface, which facilitates seed germination and the subsequent colonization of DBF trees. With ecosystem succession, shade-tolerant ENF trees gradually grow beneath the shade intolerant DBF canopy and, with time, ENF trees out-compete DBF ones and the ecosystem succeeds to its climax. In particular, a few studies show that the chance for DBF trees to establish depends highly on the thickness of the remaining organic layer after fire, with more complete burning by fire leading to greater chances (Johnstone et al., 2010b; Barrett et al., 2011).

We argue that these natural mechanisms, and the fact that broadleaf and mixed forests are widespread in boreal regions (our Extended Data Fig. 1), imply that there is a reasonable chance of modifying the local environment through human management and of assisting species migration in order to increase the share of DBF trees in this region. This is particularly the case if we consider that about two-thirds of boreal forests are under active management (Gauthier et al., 2015). In fact, one of the reasons, as argued by Astrup et al., (2018), underlying the fact that climate-friendly species shifting has not happened is not because of the issue of the environmental and climate suitability, but rather whether the wood market can embrace such a change in timber quality (i.e., from conifers to broadleaf).

Despite our long arguments here, we agree with the referee that caution should be taken in advocating such a widespread change. The implementation strategy for such a potential change should also be well conceived. Considering the space limit, we have modified the relevant sentences in the revised manuscript to: “*Given that nearly two-thirds of boreal forests are under human management, judicious species selection at the time of stand regeneration could thus be a preventive alternative to active fire suppression.*” (lines 203–205 in the revised manuscript).

[C9] Line 154: There is an assumption here that burn severity scales with fire size. Any studies that support or refute this idea might be worth contextualizing here. For example, Cansler and McKenzie 2014 showed a positive relationship between fire size and severity <https://esajournals.onlinelibrary.wiley.com/doi/full/10.1890/13-1077.1>

[Response] It is a good point. In our original text we mentioned ‘large fires burn more intensively than small fires’ but omitted the aspect of severity. According to a review by Keeley (2009), ‘fire severity’ has a strong relevance to fire intensity, and is more generally used than the term ‘burn severity’. Fire severity can refer to fire-induced loss in live or dead biomass but also to forest mortality, whereas ‘burn severity’ is more popular among the remote-sensing community and is often based on metrics similar to fire severity. Hence our

reported increases in forest damage measured by Δ LAI and forest mortality with increasing fire size could be considered as increasing ‘fire severity’ with size, similar to Cansler and McKenzie (2014) cited by the referee.

Following the referee’s suggestion, we have added “At the same time, forest damage, as measured both by the drop in LAI and postfire forest mortality, increased with fire size, suggesting increasing fire severity with size as reported in previous studies¹⁴.” in lines 142–145 in the revised manuscript. We have also revised the concerned original sentence to: “large fires lead to enhanced surface warming because they burn more intensively and more severely than small fires” (line 192 in the revised manuscript).

[C10] Line 168: Perhaps, but in other systems a more frequent fire regime can lead to type conversion and replacement of shrub or grass which might have net C impacts and limited benefits from LST/ET perspective. This might be a system that has higher biogeochemical post-fire effects than biophysical from a climate perspective.

[Response] The replacement of a pre-fire forest ecosystem by shrub or grass as suggested by the referee does happen in tropical forests and, although not yet a widespread phenomenon, in temperate and boreal forests. In southeastern Amazon forests, after repeated burns (up to six times in a period of seven years) the forest has been reported to show signs of transitioning to an alternative state where woody canopy was reduced by half and invasive grasses were favored (Balch et al., 2015).

In boreal forests the probability is low for repeated burning in the initial decades after a fire because it takes time to accumulate the necessary fuel load needed for burning. Therefore, boreal forests in general show considerable resilience in terms of maintaining a forested state rather than shifting to shrub or grass following fire disturbances (Hart et al., 2019). Due to ongoing climate warming, different tree species may successfully establish following a disturbance, potentially altering species distribution (Reich et al., 2022). Moreover, large-scale widespread drought conditions can overwhelm the control of burning by fuel load and lead to repeated burning before the regenerating stand reaches productive maturity. The shortened fire return interval, in particular when combined with postfire warming and drought, can lead to complete regeneration failure and a treeless landscape (Whitman et al., 2019; Baltzer et al., 2021).

Given that the manuscript focusses on the northern forest biomes, changes in forest composition and possible regeneration failure are now mentioned in the revised manuscript: “Postfire increase in soil temperature may differentially alter the growth rate of seedlings of different tree species, leading to transitions in forest species composition. A lower soil

moisture after fire combined with surface warming can even render complete regeneration failure.” (lines 215–217).

[C11] Line 297: It might be useful to remind the reader of the minimum fire size in GFA. I believe this should be ~25ha or so. Since LST has a resolution of 1-km, there could be some artifacts at small scales/small fires where a LST pixel coincides with partially burned and partially unburned land. Can the authors identify a screening approach to ensure their scaling results are not a function of this? I am more confident the effect they identify is real based on the scaling for larger fires.

[Response] The minimum fire size valid for our analysis is explained in lines of 469–471 in the original manuscript as: ‘*For both regression types, the minimum fire size required for a fire event to be included was set as 1km² (i.e., four 500m MODIS burned pixels), to remove possible observational uncertainties associated with small fire patches*’. However, to increase the clarity, we now explain this at the very start of the revised Methods section where the GFA data is introduced (lines 384–386): “*GFA has a minimum fire size of 0.25 km² but to avoid observational uncertainties associated with small fire patches, only fires with a size larger than 1 km² (i.e., four 500m MODIS burned pixels) were included in our analysis*”

We were able to show that the reported scaling relationships were not an artifact relating to partial burning by small fires or related to the border of a fire patch, please refer to our responses to the comments C4 & C5 raised by Referee #2.

[C12] Line 338: It might be good to justify the 2-year window given the broad audience of the journal. This could be associated with indirect mortality post-fire, direct fire mortality, and/or time-lag in detection tied with phenological measures.

[Response] Long-term dynamics in tree mortality after fire have been reported in several studies based on field observations. These studies include fires in boreal forests in Canada (Brown and DeByle, 1987; Angers et al., 2011), temperate coniferous forests in South Dakota in US (Keyser et al., 2006), Mediterranean forests (Catry et al., 2010), and the Amazon forests (Barlow et al., 2003). Extended tree mortality lasting for up to ten years after fire has been consistently reported, but the majority of mortality events occurred within the first three years after fire. On the other hand, salvage logging following fire may lead to the false attribution of forest loss to fire disturbance. To balance the omission error by neglecting long-term tree mortality and the commission error by potentially including salvage logging events, we decided to use a 2-year window after fire. An alternative approach of using a 1-year window (i.e., by accounting for forest loss within the year of fire and one year after fire) yielded slightly lower mortality rates than the 2-year window but the difference is almost negligible. The value of the scaling coefficient between mortality and fire size was almost the same whether a 1-year or a 2-year window was used (Extended Data Fig. 15 in the revised

manuscript). Including only mortality events within the year of fire led to a much lower mortality rate for a given fire size but the scaling relationship still holds (Extended Data Fig. 15 in the revised manuscript). In the end, considering both the 2–3-year length of extended postfire mortality reported in the literature and the negligible difference between the mortality rates obtained using 1-year and 2-year windows, we decided to carry on using a 2-year window.

The reasoning presented here has been incorporated in the revised manuscript (lines 439–450, Extended Data Fig. 15).

[C13] Line 353: Is there a difference between delta daytime vs delta nighttime LST? From the energetic perspective, daytime seems like it would be most impacted and potentially somewhat less downward LW from vegetation at night could have an opposite effect.

[Response] Following the referee’s suggestion, we investigated postfire surface temperature changes during both daytime and nighttime and their scaling with fire size. Fires led to surface warming during daytime and cooling during nighttime in northern hemisphere summer, but the magnitude of daytime warming was much larger than nighttime cooling (Supplementary Fig. 2). Both daytime and nighttime ΔT scaled with fire size, with daytime warming being much more sensitive to fire size ($\beta_{\Delta T}=1.03 \text{ K } [\log_{10}(\text{km}^2)]^{-1}$) than nighttime cooling ($\beta_{\Delta T}=-0.02 [\log_{10}(\text{km}^2)]^{-1}$). Hence the reported $\beta_{\Delta T}$ value of $0.51 \text{ K } [\log_{10}(\text{km}^2)]^{-1}$ for daily surface warming in summer (Fig. 1a) was an outcome of a strong daytime sensitivity being partly offset by the nighttime sensitivity that is two orders of magnitude lower. Both daytime warming and nighttime cooling, and their scaling with fire size, were found remaining for a trajectory of up to 14 years after fire, leading to persistent daily warming amplified by fire size (Supplementary Fig. 3).

These results have been incorporated in Supplementary Fig. 2, Supplementary Fig. 3 and Supplementary Text 2.1 (lines 83–89) in the revised manuscript. Below we provide some explanations for nighttime cooling but, given that this is not the most innovative component of our results, we avoid going into too much detail regarding cooling mechanisms in the revised manuscript.

The observed daytime warming is well explained by postfire reductions in ecosystem evapotranspiration, however, the mechanisms behind the nighttime cooling are not completely clear. Nighttime surface cooling has also been reported for deforestation (Lee et al., 2011; Schultz et al., 2017), a process similar to fire in that both involve canopy loss. Contributing factors for the observed nighttime cooling due to forest canopy loss might include: (1) turbulent heat exchanges between the warm air aloft and the land surface being reduced because the canopy loss has reduced the surface roughness. (2) Downward longwave

radiation from clouds might have been reduced because reductions in evapotranspiration during daytime might have reduced cloud formation. We suspect that the reduced longwave radiation from forest canopy per se as suggested by the referee has a minor influence, because for dense-canopy temperate forest, the ‘surface’ radiometric temperature observed by the satellite refers to the top of the canopy. Boreal forests can have sparse canopies (especially in the high latitudes), making the satellite-observed surface temperature a mixed signal from both ground soil and canopy surface. However, shortly after fire, due to the partial or complete canopy removal, a large contribution from mineral soil is expected. Hence the ‘land surfaces’ observed by the satellite before and after fire could be quite different. The observed surface radiometric temperature should contain more signals from the canopy surface prior to the fire occurrence than afterwards. In this case, the longwave radiation from the vegetation canopy per se during nighttime contributes to the surface temperature of mineral soil but not to that of the canopy surface. (3) soil moisture has been reported to have decreased after fire in boreal forests (Harden et al., 2006; Holden et al., 2015), which might further reduce soil heat capacity and the resulting daytime heat storage, and consequently reduce the heat transfer from the subsoil to the soil surface during nighttime.

[C14] Line 370: This is very coarse and will not capture the heterogeneity of cloud cover or aspect slope on downward SW. There are some gridded downward SW data from reanalysis at 0.1 degree from ERA-5 land. However, this is not a very critical part of the analysis and might not matter a lot.

[Response] A 0.1° grid cell at a latitude of 60°N covers approximately 55 km², whereas the fire patches contributing most to the total burned area in boreal regions ranged between 30 km² and 300 km² (Lehsten et al., 2014). As we focus on the local biogeophysical effects of fires and ignored fire-induced cloud-regime changes and their associated effects on incoming shortwave radiation, using coarse-scale incoming shortwave radiation data is actually more compatible with our assumptions.

[C15] Line 384: This needs to be better stated in the text. The influence on albedo, LST, is likely rapidly fading as vegetation regrows based on other studies on this topic.

[Response] Following the referee’s suggestion, in the revised manuscript we analysed postfire biogeophysical changes for a period of up to 14 years after fire. Please refer to our responses to the first question of Comment C1.

[C16] Line 402: Is T_{res} based on a single candidate pixel, or do you find all pixels meeting the criteria here to create an average ΔT_{res} ?

[Response] We found all candidate pixels and calculated the average ΔT_{res} . This has now been explained in line 399 of the Methods section.

[C17] Line 435: Since the Stefan-Boltzmann equation is very nonlinear, it might also better to use $LW_{out} = (\epsilon \sigma T_{daytime}^4 + \epsilon \sigma T_{nighttime}^4) / 2$

[Response] It is a good point. Following the referee's suggestion, we updated the calculation of ΔLW by separating the contributions from changes in surface radiometric temperature for daytime and for nighttime, respectively. The corresponding sections and figures have been updated in the revised manuscript. Please refer to lines 550–555 in the revised Methods and the revised Extended Data Fig. 2.

[C18] Line 561: How credible are records of number of fire counts in these locations? Fire counts are prone to many uncertainty given the nature of accounting with inhomogeneity in datasets due to a host of factors like duplicates, inclusion of records (e.g., Short et al., 2015) <https://www.publish.csiro.au/wf/WF14190>

[Response] The burned area and fire count data for China were all obtained from the Chinese government's forest department, which, to the best of our knowledge, is the most reliable and authoritative source for long-term forest fire data in China and widely used in the literature (Zong et al., 2022; Lü et al., 2006). In the revised manuscript, this dataset has been compared with three other independent remote-sensing datasets to examine its validity. Please refer to lines 663–674 in the revised manuscript for its updated documentation and Extended Data Fig. 21 for the comparison with satellite-derived burned areas.

For European countries, the historical statistical data on forest fires were from the European Fire Database, which is published by the European Forest Fire Information System, the official European Union data portal for fires. The dataset has been used extensively for examining European forest fires trends, particularly in the Mediterranean regions which dominate forest fires in Europe (San-Miguel-Ayanz et al., 2013; Artes et al., 2019). To validate the temporal trend in burned area for the selected European countries, the statistical data were compared with three other independent satellite-derived burned area datasets. Please refer to lines 680–688 and Extended Data Fig. 22 in the revised manuscript.

Therefore, overall, we believe that analyzing long-term forest fire trends using statistical data from China and the selected European countries is acceptable.

[C19] Figure 3: Why are the letters here in these figures? I would expect this information to be self-contained within the figure and caption. The statistics in panels a,b,d,f are a bit confusing and might be unnecessary. Instead, you can perhaps use colors to show statistical significance as this figure is pretty complicated for being black and white.

[Response] Different letters indicate significant differences among forest types ($p < 0.05$) as stated in the original figure caption. The use of lowercase letters to denote statistically significant differences is quite common in the literature. The alternative would be to describe

the results as ‘differences among forest types are all significant except for ENF & DFN for panels a, b & d’ in the caption. But we consider the latter option to be less clear. Hence we used letters. However, to increase the readability, we have plotted panels a and e (i.e., the vulnerability panels) as bar plots in blue. Also, the statistics (R^2 , p-value) in panels b, c, d, f have been removed. Please refer to the revised Fig. 3.

[C20] Figure 4: Does this figure only show results for temperate and boreal in each country (e.g., excludes subtropical forests in Australia/US)? There are forests shown in green equatorward of 40 and unsure if those are counted here.

[Response] Figure 4 includes **all** forests in the countries and regions being analyzed. This is now explained in the revised figure caption. The reason for including forests beyond our main research domain (40°N–70°N) is now stated in the revised main text (lines 74–77): “*The study domain was extended to include all forests in China, selected European countries, the continental USA, and Australia, for the analyses of trends in the average fire size and extreme fire size. Extending the domain for this specific aspect enabled us to demonstrate the broader implications of our findings.*”. The scaling relationship between postfire surface warming and fire size for Canadian forests is shown in Supplementary Text 1 (Supplementary Fig. 1), and those for continental US and Australia are shown in Extended Data Fig. 12. The Australian case is particularly useful as it helps illustrate the role of snow cover in driving postfire winter cooling in northern forests, because Australian fires showed a winter warming effect largely due to the near-absence of any snow effect (Extended Data Fig. 12b, Extended Data Table 1).

[C21] Figure ED4: The scaling relationship for fire duration, spread rate, and FRP are quite remarkable. I am also interested in any ideas as to why the FRP relationships are substantially stronger in N. America vs Eurasian forests as Fig ED5e,f show these results to be much more heterogeneous.

[Response] We agree with the referee that, compared to fire duration and spread rate, the scaling relationship for FRP is spatially more heterogeneous. Part of the reason for this might be that fire duration, spread rate and fire size were all obtained from the single GFA dataset and hence coherences could be expected among these three variables (but note that R^2_{duration} also showed low values where mixed and broadleaf forests dominated).

The lower β_{FRP} in Eurasian forests compared to N. American ones could be explained by differences in their fire regimes. Previous studies have reported smaller fire size and frontline intensity in boreal Eurasian forests than in N. American ones (de Groot et al., 2013). Our results confirmed the smaller fire intensity by showing a smaller FRP (another measure of fire intensity but different from frontline intensity) in Eurasia than in N. America for the same fire size (Extended Data Fig. 23a). Eurasia also showed a smaller β_{FRP} than N. America. In addition, β_{FRP} also showed a decreasing trend with increasing coverage of mixed and

broadleaf forests similar to the trends for postfire forest mortality and postfire surface warming (Extended Data Fig. 23b). Extended Data Fig. 1 shows that the DBF and MF forests are mainly distributed across the northeastern US and eastern Canada (the provinces of Ontario, Quebec and Newfoundland and Labrador) but fires are relatively rare in these regions. In contrast, large tracts of MF and DBF forests in western and central Siberia are subject to fires and have lower β_{FRP} values (because β_{FRP} decreases with increasing DBF and MF coverage). Hence the spatial heterogeneity in Fig. ED5e and Fig. ED5f can be explained by the different fire regimes in Eurasia and N. America and by the difference in their forest type distributions.

In the revised manuscript, Extended Data Fig. 23 was added, with the associated descriptions being incorporated in the revised caption of Extended Data. Fig. 6.

[C22] Figure ED9: The costs here either refer to different actions or have different nomenclature. For example, fire prevention and fire suppression are pretty different items. It would be good to clarify.

[Response] Thank you for pointing this out. The meaning of fire control expenditure is explained in detail for each country or country group in the revised figure caption (now Extended Data Fig. 13). Here, it is important to look at historical changes in fire control expenditure for each individual country or country group rather than comparing between different countries. Hence it is acceptable that ‘fire control’ has different scopes for different countries. Given the differences in the forest fire situation and management strategies among different countries, it is difficult to find a single common indicator to measure their efforts in mitigating forest fires.

[C23] Of the broader impacts of the work that might be further emphasized is the effect that elevated LST post-fire for very large fires would pose on tree regeneration in warmer sites. Postfire regeneration often favors cooler/wetter years, and hence another feedback here could be through elevated LST → reduced tree regeneration → reduced C sequestration. Thus, the biophysical feedback could strengthen the biogeochemical feedback. I don’t expect this as an additional analysis, but it could be considered as part of the discussion.

[Response] Field observations show that warming alone, or warming combined with reduced precipitation, can reduce tree seedling growth of the currently dominant conifer species in boreal regions and increase their seedling mortality, whereas it can enhance the seedling growth of the broadleaf species which are currently subordinate in boreal forest (Reich et al., 2022). Hence land surface warming following fire disturbance has the potential to alter postfire forest species composition. Consistent with the referee’s comment, shortened fire return intervals with postfire warming and drought can reduce the seedling density of regenerating stands (Whitman et al., 2019) or even lead to complete regeneration failure

(Baltzer et al., 2021). The latter case will definitely further strengthen the biogeochemical climate feedbacks due to fires.

These points have been incorporated in the revised main text as: “Postfire increase in soil temperature may differentially alter the growth rate of seedlings of different tree species, leading to transitions in forest species composition. A lower soil moisture after fire combined with surface warming can even render complete regeneration failure.” (lines 215–217 in the revised main text).

Comments by referee #2

This manuscript analyzes the relationships between fire size and biogeophysical responses in the northern forest. The authors found that during the summer months immediately following wildfire, the bigger the fire, the stronger its effect on outgoing shortwave radiation, latent heat flux, outgoing longwave radiation, the combination of sensible and ground heat fluxes, land surface temperature, and surface albedo.

[C1] Overall, this manuscript uses the method of Alkama, R., & Cescatti, A. 2016.

“Biophysical climate impacts of recent changes in global forest cover.” It is not surprising that the results of the two studies are similar since fire is one of the forest disturbances. This manuscript focuses only on forest cover loss due to fire in the northern latitude. And while it is still interesting how fires affect land surface temperature, the authors already published that in 2021 for North American boreal forests. The only difference between the current study and 2021 is the expansion of the study area and mainly analyzing changes in biogeophysical responses in relation to fire size. However, I have a number of concerns regarding the data, method, and interpretation of the results.

[Response] Although we appreciate the referee’s efforts in reviewing our paper and the comments that helped to improve the manuscript, we disagree with the referee on the similarity in the results of our study and that of Alkama and Cescatti (2016) and on the lack of advances in this study in comparison to our paper published in 2021 (Zhao et al., 2021).

The Alkama and Cescatti paper looked at how land surface temperature changes with the percentage of forest cover loss over 0.05° grid cells (Alkama and Cescatti, 2016); this is reflected in their x-axis coordinates that saturate at 100% in their Figure 1. In our case there is no upper boundary for fire size, at least from a theoretical point of view. In the results of Alkama and Cescatti, the temperature change scales with the ground fraction of forest cover loss, probably because of the pure spatial extent effect, i.e., the more forests are lost within a given 0.05° grid cell, the greater the temperature change is. In contrast, in our study, the scaling between temperature change and fire size is driven by changes in fire behaviour (both

intensity and severity) and the associated cascading biogeophysical effects. Although they use related methods and jargon, the two studies are fundamentally different. The Alkama and Cescatti paper, by design, could not and did not report a scaling relationship with fire size driven by increasing fire intensity and fire severity. Moreover, they didn't and couldn't reveal the previously overlooked feedback between climate warming and forest fire amplified by the fire size effect.

In addition, our case study for Canadian fires, shown in our response to Comment C5 of Referee #2, further supports the argument that our analysis is fundamentally different from the Alkama and Cescatti paper. There, both the effects of fire size and the fraction of ground actually being burned within the fire patch (which could be considered as a proxy for forest cover loss) on postfire surface temperature change were examined. The results revealed a positive effect of burned percentage on postfire surface warming consistent with the Alkama and Cescatti paper, but after accounting for this co-varying factor, the scaling relationship between warming and fire size still holds. Please refer to the detailed descriptions given in our response to Comment C5.

This scaling relationship is one of the core findings of this study. This finding was by no means presented or even suggested in the earlier study of Zhao et al., (2021). In fact, any potential impacts of fire size on surface temperature change were not examined in Zhao et al., (2021). The second prominent difference between this study and Zhao et al., (2021) is how changes in fire behavior have driven forest damage (leaf fire index, forest mortality) and the cascading biogeophysical effects with fire size: these were not presented at all in Zhao et al., (2021). Third, Zhao et al., (2021) was limited to Canadian and Alaskan forests, whereas this study includes the whole northern temperate and boreal forest domain. The expansion in the study domain in comparison to Zhao et al., (2021) did not simply result in "more of the same", as suggested by the referee, but allowed us to reveal the role of forest type in regulating postfire responses to fire size. This finding provides insights in how broadleaf species could be used to mitigate the impacts of large fires. Again, this crucial point was not present in Zhao et al., (2021).

[C2] First is the quality of the data used in the manuscript. Fire Atlas (Andela et al. 2019), from where fire boundaries were extracted, contains many uncertainties which were not acknowledged in this manuscript. For example, MODIS BA product and derived from it Fire Atlas are not ideal data sources for high latitudes. Chen et al. (2021), in "Missing Burns in the High Northern Latitudes: The Case for Regionally Focused Burned Area Products," reported that "MCD64A1 missed nearly half the total burned areas in the Alaskan and Canadian boreal forests and the tundra during the 15-year period." In Canada and Alaska, the ABOVE dataset based on Landsat is much superior, and it is unclear why the authors did not use the ABOVE

product. Additionally, I attached figures with several examples where Fire Atlas cuts fires if they occur between MODIS tiles, therefore, inaccurately calculating the fire size of the individual fires.

The referee raised three issues in this comment:

(1) The referee cites Chen et al., (2021) concluding that “MCD64A1 missed nearly half of the total burned areas in the Alaskan and Canadian boreal forests and the tundra during the 15-year period” and argues that the Global Fire Atlas is not an ideal data source for high latitudes.

[Response] We would agree with the referee that the Global Fire Atlas (GFA) is not perfect for high latitudes if by ‘high latitudes’ the referee meant regions at $>60^{\circ}\text{N}$, because in Chen et al., (2021), only regions at $>60^{\circ}\text{N}$ were used to investigate the performance of MCD64A1. However, we selected GFA for our main analyses for three reasons: (1) The burned area dataset advocated in Chen et al., (2021) (i.e., the ABBA data, also based on MODIS data) provides burnt areas rather than the fire scars needed in this study to derive fire size information. (2) Our research domain is northern temperate and boreal forest (40°N – 70°N). According to GFA, burned areas within the region of $>60^{\circ}\text{N}$ accounted for only 25% of the total burned area of 40°N – 70°N . Hence the MCD64A1-based GFA data remained the only dataset completely covering our research domain which at the same time provided information on fire behaviour. (3) To address the referee’s comment, a comparison study, using multiple datasets for western Canadian and Alaskan boreal forests, was conducted and reported in our revised manuscript. The results confirmed the robustness of the reported scaling relationship between postfire surface warming and fire size irrespective of the underlying dataset. The uncertainties in the estimated $\beta_{\Delta T}$ associated with using GFA have now been fully explored in light of the results derived by using other datasets (See Supplementary Text 1 in the revised manuscript and lines 83–86 in the main text).

On the other hand, a global, systematic evaluation of the MCD64A1 burned area data conducted by the official data production team (Boschetti et al., 2019) has shown that it has good performance over the entire boreal forest biome (with a relative bias of -4% when an omission error of 27% is partly offset by a commission error of 24%). Another study using high-resolution satellite data validating MCD64A1 for boreal Eurasia found that over forests MCD64A1 underestimated burned area by 20% (Zhu et al., 2017). A third and recent study developed a new burned area dataset for the whole of Canada and US Alaska, also under the ABoVE project, which showed that MCD64A1 underestimated burn area in this region by 23% compared to the new dataset (Potter et al., 2022). Considering these pieces of evidence, the omission error in burned area by MCD64A1 for boreal forests reported in Chen et al., (2021) seems to be at the high end.

Finally, given that our focus is on how postfire biogeophysical processes responds to fire size rather than burned area, it is the errors in mapped fire size distribution rather than in burned area per se that influenced our key conclusion (the former is now discussed in Supplementary Text 1).

The justification for using MCD64A1, its potential limitations and the uncertainties in the quantified $\beta_{\Delta T}$ associated with its bias in fire size distribution are now included in lines 387–407, lines 83–86 in the revised manuscript, Supplementary Text 1 and Supplementary Text 3.

(2) The referee suggested using the Landsat-based ABOVE dataset covering Canada and Alaska.

[Response] Following the referee’s suggestion we visited the official data portal of the ABoVE project. Both the ABoVE-Landsat dataset (https://daac.ornl.gov/ABOVE/guides/ABOVE_Fire_Severity_dNBR.html) and the ABoVE-MODIS (https://daac.ornl.gov/ABOVE/guides/Wildfires_Date_of_Burning.html) dataset are now used in our analysis. These two products, along with the MTBS dataset, GFA, GlobFire (detailed in Supplementary Text 3 in the revised manuscript), resulting in a total of five datasets covering roughly the same region (Alaska and western Canada), were used to examine the relationship between summer postfire surface warming (ΔT) and fire size to explore the uncertainties in the quantified scaling coefficient ($\beta_{\Delta T}$). The results showed that the scaling relationship between ΔT and fire size holds irrespective of the underlying dataset used, with R^2 ranging 0.84 to 0.99 and all p-values < 0.01 , but $\beta_{\Delta T}$ tends to be conservative (i.e., underestimated) by using the global fire patch datasets compared to the regional ones because they show a higher frequency of small fires than the regional datasets.

The results described above are presented as Supplementary Text 1 in the revised manuscript and have been included in lines 83–86 in the main text.

(3) The referee shows several cases where Global Fire Atlas cut a single fire crossing two MODIS tiles into two individual fires and raised the concern that this can cause it to underestimate fire size.

[Response] We acknowledge this issue and its potential limitations in the revised Methods section (lines 400-407). Given that each MODIS tile covers an area of 10° by 10° (i.e., about 600,000 km² at a latitude of 60°N), it is expected that most of the fire patches will be located completely within the borders of individual tiles. This expectation is confirmed by an analysis of the 2016 fire season data across the entirety of the northern forests of 40°N–70°N as an

example, which showed that the number of patches in contact with MODIS tile borders accounted for only 0.63% of the total number of fire patches (n=10952), with their burned area accounting for just 3.74% of the total burned area. These percentages represent the upper boundary of the importance of fire patches subject to the tiling issue. Hence the influence of the tiling issue on our results is expected to be very limited.

To test the influence of the tiling issue on our results, we used GlobFire (Artes et al., 2019), another fire patch dataset based on the same MCD61A1 burned area data underlying Global Fire Atlas but without any known tiling issues. The two datasets generated similar results in terms of our key findings. This confirms our expectation that the tiling issue in GFA has a negligible influence on our results. These results are presented in Supplementary Text 3, lines 403–407 in the revised manuscript.

[C3] Trends in fire size were calculated using OLS, which might not be robust, especially considering that in many regions, the trend can be driven just by one or a few years with extremely large fires (see Fig. 4). Additionally, several studies reported a negative relationship between fire size and human activities. Therefore, it is possible that the increase in fire size due to warming can be offset by increasing population density, which was not addressed in the paper.

[Response] The large interannual variations in burned area and in fire size are intrinsic for fire activities and hence the observed extreme values for some specific years can hardly be considered as outliers. In principle OLS can still provide unbiased parameter estimates using observations with extreme values as long as these extreme values are not outliers.

Nonetheless, following the referee's suggestion, in the revised manuscript, we used the Theil–Sen estimator, which is insensitive to outliers and can accommodate heteroscedastic data, to estimate the trends in burned area and fire size, and used the Mann–Kendall (MK) test, a robust non-parametric method, to test their significance. These methods have also been used in previous studies on temporal trend detection in extreme fire weather which also showed great interannual variations (Jain et al., 2021). The directions of temporal trends for both the burned area and the mean fire size for all regions and countries derived using the new method were consistent with those derived using OLS, although the magnitudes of the trends differ. Please refer to lines 636–637 in the Methods, the updated Fig. 4 and Extended Data Fig. 11 in the revised manuscript for these revisions.

We agree with the referee that increases in fire size due to climate warming can be offset by increasing population density and in particular, human land use and land fragmentation (Andela et al., 2017). We believe this was the case in China and this has been briefly discussed in our original submission. But an exact attribution of changing fire size to changes in climate and in human population density is beyond our research scope.

[C4] Processing artifacts is another of my concerns. The edge-to-fire ratio that is higher for small fires may affect the reported changes in biogeophysical responses in relation to fire size. Edge pixels that are only partially burned can have less effect on land surface temperature and other biogeophysical responses. I illustrate this in the last figure attached. The fire on the left consists of 9 pixels (“X”), 8 of which are partly burned (blue “X”). The fire on the right consists of 31 pixels, 15 of which are edge pixels. Therefore, the proportion of edge-to-fire pixels for the smaller fire is twice higher compared to a bigger fire. Suppose a fully burned pixel contributes to a certain amount of increase in land surface temperature. In that case, it is reasonable to assume that a half-burned pixel contributes to a smaller increase in land surface temperature. If that is the case, large fires would appear to have a higher climate impact. In reality, the proportion of burned land within fire pixels drives that difference.

[Response] We construct a simple theoretical model to show that the change in postfire surface warming driven by the ‘edge effect’ is clearly different from that driven by a scaling relationship with fire size. Suppose that the postfire temperature change is ΔT_i for the interior and ΔT_e for the edge (note that according to the ‘edge effect’, it is assumed $\Delta T_i > \Delta T_e$), then the temperature change for the entire patch (ΔT) could be shown to saturate and approach ΔT_i as an asymptote with increasing fire size. This is clearly different to our results, in which ΔT shows no sign of saturation.

As an empirical example, changes in ΔT_i , ΔT_e and ΔT in summer were calculated for increasing fire sizes in US Alaska and Canada for 2003–2006 using the Global Fire Atlas dataset. The results showed that ΔT_i was larger than ΔT_e for all fires, but both ΔT_i and ΔT_e scaled with fire size. The observation that ΔT_i (thus excluding the edge effect), increased with fire size ($\beta_{\Delta T} = 0.55 \text{ K } [\log_{10}(\text{km}^2)]^{-1}$) demonstrates that the reported scaling relationship is not an artifact of the edge effect.

These results are presented in detail in Supplementary Text 4 and are briefly explained in lines 617–621 in the Methods in the revised manuscript.

[C5] Since MODIS BA product misses most of the small fires, it is possible that some of the pixels that are labeled as none burned experienced fire at least partially, which would propagate error into equation (2).

[Response] Our new analysis for forests in Alaska and western Canada using two 30m Landsat-based and three 500m MODIS-based datasets (Supplementary Text 1) revealed that the MODIS-based fire patch datasets with a global coverage (Global Fire Atlas and GlobFire) did not necessarily miss small fires as is suggested by the referee. First, the minimum fire size included in our analysis is 1 km² for both Global Fire Atlas and GlobFire, smaller than that provided in ABoVE-Landsat (2 km²) or MTBS (4 km²), both being based on 30-m resolution

Landsat images. Second, both Global Fire Atlas and GlobFire contained more small fires than the three regional datasets of ABoVE-Landsat, MTBS and ABoVE-MODIS (Supplementary Fig. 1b), suggesting that the underlying algorithms to construct fire patch data using MODIS burned pixels played a larger role in determining fire size distribution than the spatial resolution of the burned product being used (500 m in this case).

We agree with the referee that it is possible that some of the pixels labeled in the MODIS data as not burned may have experienced partial fire. Vice versa, some of the pixels labeled as burned can contain partially unburned areas and this can propagate errors in the quantification of fire size sensitivity (Equation (2)). However, due to a lack of such unburned fraction information in the MODIS product, a systematic assessment of the influence of this issue (i.e., sub-pixel unburned fractions within 500m burned pixels) for the whole northern temperate and boreal forest is beyond the reach of these state-of-the-art datasets (see also our responses to Comment C2 on the reason why Global Fire Atlas is the only suitable dataset for our research purpose).

Nonetheless, to have an idea of the potential influence of the ‘unburned fraction’ issue on the quantified $\beta_{\Delta T}$, we performed a specific analysis using a recently released 30m-resolution forest fire disturbance dataset across Canada (White et al., 2017), first re-gridded to 500m resolution followed by further construction of fire patches. More specifically, the original 30m resolution burned area data for 2003–2015 (the period when surface temperature data are available) were aggregated to 500-m burned pixels if the percentage of component 30m pixels identified as being burned exceeded a threshold of 20%. We intentionally set this threshold to a low value to test the effect of the ‘unburned fraction’ issue. All 500m burned pixels within a distance of 5 kilometers and within the same fire season were then aggregated into individual fire patches. Finally, the change in summer land surface temperature one year after fire (ΔT), patch size, and the burned fraction (F_{burned} , in percentage) for each individual fire patch were derived. We then fitted a multiple linear regression as follows:

$$\Delta T = \beta_0 + \beta_{\Delta T} \cdot \log_{10}(\text{size}) + \beta_F \cdot F_{\text{burned}} + \varepsilon$$

where ΔT is surface temperature change, $\log_{10}(\text{size})$ is the logarithm of fire size, $\beta_{\Delta T}$ is the scaling coefficient of postfire surface warming to fire size, F_{burned} represents the percentage of area actually burned within the fire patch, ε is the random error. Using the data across Canada described above we obtained $\beta_{\Delta T} = 0.48 \text{ K } [\log_{10}(\text{km}^2)]^{-1}$ with the coefficient of determination (R^2) as 0.23 and a p-value < 0.05 . These results indicate that, after accounting for unburned areas in the 500m burned pixels, the reported scaling relationship between surface temperature change and fire size still holds. The value of β_F was derived as 0.004 K

(%)⁻¹, showing that indeed pixels being more completely burned leads to greater postfire surface warming.

The derived $\beta_{\Delta T}$ is closer to the values derived using Global Fire Atlas and GlobFire (0.46–0.51 K [$\log_{10}(\text{km}^2)$]⁻¹) than those using the three regional datasets (0.69–0.73 K [$\log_{10}(\text{km}^2)$]⁻¹) for Alaskan and western Canadian forests (Supplementary Fig. 1), because the fire size distribution derived using the data of White et al., (2017) is closer to the two global fire-patch datasets (Figure R1 shown below).

Figure R1 Distribution probability density curves of fire size for different fire patch datasets.

The analyses presented here and in Supplementary Text 1 show that: (1) the omission of small fires, or fire size distribution, does not necessarily relate to the spatial resolution of the underlying burned area dataset; (2) fire size distribution, rather than the spatial resolution of the underlying burned area dataset, strongly influences the derived fire size sensitivity.

The uncertainties of the derived $\beta_{\Delta T}$ using Global Fire Atlas are thoroughly discussed in Supplementary Text 1 and Supplementary Text 3. As we would like to present the analysis for Canada reported here in another study, it is not included in the revised manuscript.

[C6] The accuracy of the historical data is also questionable. For China, the authors used 4 different data sources that provided data for random years. How comparable are those data sources? Can they be used together for time-series analysis? Additionally, as the author mentioned, fire size distribution is highly skewed toward small fires, which is why a log scale was used in the analysis. Therefore, it raises the question of why the authors decided to calculate the fire size trend using the mean. I argue that dividing the total amount of burned area by the number of fires is not an appropriate approach to analyze changes in fire size. The

correct way would be to calculate the 95th percentile or median, the two most common approaches in the literature.

[Response] A re-examination of this issue enhanced the homogeneity and temporal consistency of government statistical data sources for forest burned areas in China (see lines 663–674 in the revised Methods). In addition, the government statistical data for both China and the selected European countries were compared with three other independent remote-sensing datasets to confirm their validity (see Extended Data Figures 21 and 22; lines 667–674, lines 681–688 in the revised manuscript).

We agree with the referee that it would be more proper to use the median or 95th percentile fire size to represent temporal trends in fire size. Following the referee’s suggestion, for Canada, USA and Australia, where detailed information for individual fire events was available, temporal trends were calculated for the median or 95th percentile fire size. These results are presented in Extended Data Fig. 11. The median or 95th percentile fire size showed an increasing temporal trend consistent with the mean fire size.

For China and the selected European countries, no information on individual fire events was available to allow the calculation of the median or 95th percentile fire size, therefore we continue to use the mean fire size for these countries. To ensure consistency across all countries, the mean fire size is still used for all countries/regions displayed in Fig. 4 but the results for the median and extreme fire sizes are cited in its caption. Using the mean fire size is also easier to comprehend for general readers of Nature with no fire expertise.

Please refer to Extended Data Fig. 11 for the results of the median and 95th percentile fire size. The related methods are described in lines 715–716 in the Methods section of the revised manuscript.

[C7] Furthermore, why the authors included Australia in the analysis when the study is about “northern forests (40°N–70°N)”?

[Response] please refer to our responses to Comment C20 by Referee #1 where a similar question was raised.

[C8] I found the way the authors presented their results is misleading. The supplementary text is crucial as it shows that the main conclusion is only valid for the summer months. The relationships between fire size and biogeophysical responses are mostly reversed in winter. Annually, those responses are much weaker, if present at all (see Extended Data Fig. 2). Considering that the supplementary text is only a page, and Extended Data Fig. 2 provided far more information than main Fig. 2, it is not clear why the authors did not want to include those results in the main body of the manuscript.

[Response] According to Nature's formatting guide, we are aiming for a 6-page article with 4 modest display items (figures and tables). This requires a focus on what we consider the most important and exciting findings. Such a focus propagates into the content of the figures, which should be clear and concise rather than being overloaded with data. For this reason, the revised manuscript still focusses on the dynamics of the surface energy budget in summer. Nevertheless, the analysis of the energy budget during the winter months is further expanded in the revised Extended Data Figures and Supplementary Information. These sections now contain enough material for at least two manuscripts on, i.e., the difference between day and night time changes and the seasonal differences.

We started Fig. 1 focusing on summer for two reasons. First, summer is the growing season, and it is also the season when most heat waves and forest fires happen in the northern temperate and boreal region (Hirschi et al., 2010; Giglio et al., 2013). A warming impact by fire on the land surface in summer is hence expected to be far more influential than the cooling in winter. Second, we used the summer results to prepare readers for understanding those for winter and for the annual average. Note that we were not hiding the results of winter cooling and the (weaker) annual warming: they were clearly described in the main text. Also note that for Australian forests, our new analysis shows that postfire changes in surface temperature remained warming even in winter probably because of relatively weak snow influence, and the warming again scaled with fire size (Extended Data Fig. 12b and Extended Data Table 1). In fact, the winter cooling in northern temperate and boreal forests is, to a large extent, due to their snow cover (Supplementary Text 2.2).

Hence, it is not only the warming or cooling signal per se, but the underlying science mechanisms, that are important and that is what our paper is intended to present. We acknowledge that any way of presenting scientific findings comes with a strong personal flavor. In any case, we find that presenting the most impactful results with less complexity best suits our needs and the style of the journal.

We don't think it's a good idea to replace Fig. 2 with the Extended Data Fig. 2 as implied by the referee, if we understand correctly. Our key reason is that the surface energy processes driving surface temperature change following land cover change or forest fires have been well explained by previous studies. These results per se were not novel, nor our key focus. Instead, our major novelty resides in the scaling relationships between postfire biogeophysical responses and fire size. The two key underlying drivers for the surface energy processes, which scaled with fire size, were changes in surface albedo and evapotranspiration, which were further driven by changes in LAI and forest mortality. This explains our choice for Fig. 2 and our choice to put the detailed processes in Extended Data Fig. 2. Again, we aim to make

sure our figures in the main text present essential findings but at the same time remain concise and easy to follow, in particular for the broad audience of Nature.

[C9] Additionally, the authors did not find trends in fire size to be significant in any of their study areas. Therefore, their main conclusion, “The feedback loop in which climate warming results in larger fires and larger fires result in more warming renders the climate impacts by fires beyond the effects of changes in burned area per se.” is not supported by their analysis.

[Response] The three countries where we showed the scaling relationship between postfire surface warming and fire size holds, Canada, the continental US and Australia, all showed significant increases in mean fire size, median fire size and extreme fire size (represented by the 95th percentile fire size), with all trends being significant according to the Mann-Kendall test and the slopes being derived using the Theil-Sen estimator. These results are shown in the revised Fig. 4 in the main text, Extended Data Fig. 11, Extended Data Fig. 12 and Supplementary Fig. 1.

The extended analysis for a period of up to 14 years after fire showed that postfire warming in summer persists for the whole trajectory of 14 years after fire. The postfire surface radiometric temperature change in winter, however, remained a constant cooling effect (Extended Data Fig. 3, Supplement Text 2.3). Hence in the revised manuscript this sentence was revised to: “As forest fire occurrence is generally limited by the climate conditions near the land surface, the size-amplified surface warming in the fire season of summer could accelerate the feedback loop between climate warming and forest fires beyond the effects of changes in burned area per se.” (lines 221–224). In the abstract, the original sentence was revised to: “A possible feedback loop in which climate warming results in larger fires and larger fires result in more warming would thus render the climate impacts by fires beyond the effects of changes in burned area per se.” (lines 35–37) This sentence is presented as **an implication** of our study which could arise from scientific reasoning rather than a direct outcome of our analyses. Please also see our related responses to Comment C11.

[C10] Moreover, “postfire warming” that the authors presented as the main conclusion was only detected during the year immediately after the fire, which was not particularly clear from the abstract.

[Response] The initial analysis for one year after fire is now extended for a period of up to 14 years after fire in the revised manuscript. Please refer to our responses to a similar comment raised by Referee #1 (i.e., our response to the first question of Comment C1 by Referee #1).

[C11] The authors talked about accelerating climate warming due to large fires, while a large body of literature states otherwise, which was not addressed in this manuscript. For example, Chen et al. (2018), in “Strong cooling induced by stand-replacing fires through albedo in

Siberian larch forests,” found that “the cooling effect, however, grows rapidly and continuously until the 11th year . . . , after which it gradually lessens.” Similarly, in North America, wildfires were found to impose a cooling effect after the fire, according to Randerson et al. (2006, <https://doi.org/10.1126/>), McMillan et al. (2008, <https://doi.org/10.1029/2007gb003038>), Amiro et al. (2006, <https://doi.org/10.1016/j.agrformet.2006.02.014>), Rogers et al. (2015, <https://doi.org/10.1038/ngeo2352>), and Randerson et al. 2012, <https://doi.org/10.1029/2012gl051790>).

[Response] Following the referee’s suggestion, the surface radiative forcing effect of forest fires has been examined for a period of up to 14 years after fire. Please refer to Extended Data Fig. 3, Extended Data Fig. 4, Supplementary Text 2, lines 108–112, lines 116–118, lines 120–124 in the main text.

Consistent with the studies cited by the referee, we also found a persistent surface **radiative** cooling effect due to fire for up to 14 years after fire for summer, winter and the annual time scale, by accounting for both shortwave and longwave radiative forcing effects (Extended Data Fig. 3h). The surface **radiative** cooling in summer, however, does not contradict the observed surface **radiometric** warming, because the former is determined by pure radiative processes, whereas the latter incorporates both radiative processes, and non-radiative ones including latent heat, sensible heat and ground heat fluxes. In summer, the change in surface **radiometric** temperature was dominated by non-radiative processes that lead to surface warming, whereas in winter, the change was dominated by radiative processes that lead to surface cooling. On the annual time scale, for the initial several years after fire, summer warming outweighed winter cooling, leading to warming, but afterwards the reverse was true, with cooling on the annual time scale (Supplementary Text 2.1 and 2.3, lines 120–128 in the revised main text). Note that assessing radiometric warming requires a more complete analysis of the energy budget than a radiative assessment.

The observed surface **radiometric** warming in summer was supported by field observations in boreal forests (Liu , 2005) (line 215 in the revised main text) and has consequences for soil thermal and hydrological processes (lines 212–214 in the main text). As summer is the major fire season for the northern temperate and boreal region (Giglio et al., 2013) and forest fire occurrence is generally limited by the climate conditions near the land surface (van Wagner, 1987), theoretically there is still feedback between postfire surface warming in summer and fire occurrence which could be strengthened by the observed fire size effect (lines 221–224 in the main text), which is one of the key implications of our study.

References

- Alkama, R., Cescatti, A., 2016. Biophysical climate impacts of recent changes in global forest cover. *Science* 351, 600-604.
- Andela, N., Morton, D.C., Giglio, L., Chen, Y., van der Werf, G.R., Kasibhatla, P.S., DeFries, R.S., Collatz, G.J., Hantson, S., Kloster, S., Bachelet, D., Forrest, M., Lasslop, G., Li, F., Mangeon, S., Melton, J.R., Yue, C., Randerson, J.T., 2017. A human-driven decline in global burned area. *Science* 356, 1356-1362.
- Angers, V.A., Gauthier, S., Drapeau, P., Jayen, K., Bergeron, Y., 2011. Tree mortality and snag dynamics in North American boreal tree species after a wildfire: a long-term study. *International Journal of Wildland Fire* 20.
- Artes, T., Oom, D., de Rigo, D., Durrant, T.H., Maianti, P., Liberta, G., San-Miguel-Ayanz, J., 2019. A global wildfire dataset for the analysis of fire regimes and fire behaviour. *Sci Data* 6, 296.
- Astrup, R., Bernier, P.Y., Genet, H., Lutz, D.A., Bright, R.M., 2018. A sensible climate solution for the boreal forest. *Nature Climate Change* 8, 11-12.
- Balch, J.K., Brando, P.M., Nepstad, D.C., Coe, M.T., Silvério, D., Massad, T.J., Davidson, E.A., Lefebvre, P., Oliveira-Santos, C., Rocha, W., Cury, R.T.S., Parsons, A., Carvalho, K.S., 2015. The Susceptibility of Southeastern Amazon Forests to Fire: Insights from a Large-Scale Burn Experiment. *BioScience* 65, 893-905.
- Baltzer, J.L., Day, N.J., Walker, X.J., Greene, D., Mack, M.C., Alexander, H.D., Arseneault, D., Barnes, J., Bergeron, Y., Boucher, Y., Bourgeau-Chavez, L., Brown, C.D., Carriere, S., Howard, B.K., Gauthier, S., Parisien, M.A., Reid, K.A., Rogers, B.M., Roland, C., Sirois, L., Stehn, S., Thompson, D.K., Turetsky, M.R., Veraverbeke, S., Whitman, E., Yang, J., Johnstone, J.F., 2021. Increasing fire and the decline of fire adapted black spruce in the boreal forest. *Proc Natl Acad Sci U S A* 118.
- Barlow, J., Lagan, B.O., Peres, C.A., 2003. Morphological correlates of fire-induced tree mortality in a central Amazonian forest. *Journal of Tropical Ecology* 19, 291-299.
- Barrett, K., McGuire, A.D., Hoy, E.E., Kasischke, E.S., 2011. Potential shifts in dominant forest cover in interior Alaska driven by variations in fire severity. *Ecol Appl* 21, 2380-2396.
- Beck, P.S.A., Goetz, S.J., Mack, M.C., Alexander, H.D., Jin, Y., Randerson, J.T., Loranty, M.M., 2011. The impacts and implications of an intensifying fire regime on Alaskan boreal forest composition and albedo. *Global Change Biology* 17, 2853-2866.
- Beer, C., Lucht, W., Gerten, D., Thonicke, K., Schimmlus, C., 2007. Effects of soil freezing and thawing on vegetation carbon density in Siberia: A modeling analysis with the Lund-Potsdam-Jena Dynamic Global Vegetation Model (LPJ-DGVM). *Global Biogeochemical Cycles* 21.
- Boschetti, L., Roy, D.P., Giglio, L., Huang, H., Zubkova, M., Humber, M.L., 2019. Global validation of the collection 6 MODIS burned area product. *Remote Sens Environ* 235.
- Brown, J.K., DeByle, N.V., 1987. Fire damage, mortality, and suckering in aspen. *Canadian Journal of Forest Research* 17, 1100-1109.
- Cansler, C.A., McKenzie, D., 2014. Climate, fire size, and biophysical setting control fire severity and spatial pattern in the northern Cascade Range, USA. *Ecol Appl* 24, 1037-1056.

- Catry, F.X., Rego, F., Moreira, F., Fernandes, P.M., Pausas, J.G., 2010. Post-fire tree mortality in mixed forests of central Portugal. *Forest Ecology and Management* 260, 1184-1192.
- Chen, D., Shevade, V., Baer, A., Loboda, T.V., 2021. Missing Burns in the High Northern Latitudes: The Case for Regionally Focused Burned Area Products. *Remote Sensing* 13.
- Cui, W., Perera, A.H., 2008. What do we know about forest fire size distribution, and why is this knowledge useful for forest management? *International Journal of Wildland Fire* 17.
- de Groot, W.J., Cantin, A.S., Flannigan, M.D., Soja, A.J., Gowman, L.M., Newbery, A., 2013. A comparison of Canadian and Russian boreal forest fire regimes. *Forest Ecology and Management* 294, 23-34.
- Fairman, T.A., Bennett, L.T., Nitschke, C.R., 2019. Short-interval wildfires increase likelihood of resprouting failure in fire-tolerant trees. *J Environ Manage* 231, 59-65.
- Gauthier, S., Bernier, P., Kuuluvainen, T., Shvidenko, A.Z., Schepaschenko, D.G., 2015. Boreal forest health and global change. *Science* 349, 819-822.
- Giglio, L., Randerson, J.T., van der Werf, G.R., 2013. Analysis of daily, monthly, and annual burned area using the fourth-generation global fire emissions database (GFED4). *Journal of Geophysical Research: Biogeosciences* 118, 317-328.
- Hantson, S., Padilla, M., Corti, D., Chuvieco, E., 2013. Strengths and weaknesses of MODIS hotspots to characterize global fire occurrence. *Remote Sensing of Environment* 131, 152-159.
- Hantson, S., Pueyo, S., Chuvieco, E., 2015. Global fire size distribution is driven by human impact and climate. *Global Ecology and Biogeography* 24, 77-86.
- Hantson, S., Pueyo, S., Chuvieco, E., 2016. Global fire size distribution: from power law to log-normal. *International Journal of Wildland Fire* 25.
- Harden, J.W., Manies, K.L., Turetsky, M.R., Neff, J.C., 2006. Effects of wildfire and permafrost on soil organic matter and soil climate in interior Alaska. *Global Change Biology* 12, 2391-2403.
- Hart, S.J., Henkelman, J., McLoughlin, P.D., Nielsen, S.E., Truchon-Savard, A., Johnstone, J.F., 2019. Examining forest resilience to changing fire frequency in a fire-prone region of boreal forest. *Glob Chang Biol* 25, 869-884.
- Hayes, D.J., McGuire, A.D., Kicklighter, D.W., Gurney, K.R., Burnside, T.J., Melillo, J.M., 2011. Is the northern high-latitude land-based CO₂sink weakening? *Global Biogeochemical Cycles* 25, n/a-n/a.
- Hirschi, M., Seneviratne, S.I., Alexandrov, V., Boberg, F., Boroneant, C., Christensen, O.B., Formayer, H., Orłowsky, B., Stepanek, P., 2010. Observational evidence for soil-moisture impact on hot extremes in southeastern Europe. *Nature Geoscience* 4, 17-21.
- Holden, S.R., Berhe, A.A., Treseder, K.K., 2015. Decreases in soil moisture and organic matter quality suppress microbial decomposition following a boreal forest fire. *Soil Biology and Biochemistry* 87, 1-9.
- Jain, P., Castellanos-Acuna, D., Coogan, S.C.P., Abatzoglou, J.T., Flannigan, M.D., 2021. Observed increases in extreme fire weather driven by atmospheric humidity and temperature. *Nature Climate Change* 12, 63-70.

- Johnstone, J.F., Chapin, F.S., Hollingsworth, T.N., Mack, M.C., Romanovsky, V., Turetsky, M., 2010a. Fire, climate change, and forest resilience in interior Alaska. *Canadian Journal of Forest Research* 40, 1302-1312.
- Johnstone, J.F., Hollingsworth, T.N., Chapin, F.S., Mack, M.C., 2010b. Changes in fire regime break the legacy lock on successional trajectories in Alaskan boreal forest. *Global Change Biology* 16, 1281-1295.
- Keeley, J.E., 2009. Fire intensity, fire severity and burn severity: a brief review and suggested usage. *International Journal of Wildland Fire* 18.
- Keyser, T.L., Smith, F.W., Lentile, L.B., Shepperd, W.D., 2006. Modeling Postfire Mortality of Ponderosa Pine following a MixedSeverity Wildfire in the Black Hills: The Role of Tree Morphology and Direct Fire Effects. *Forest Science* 52, 530-539.
- Lee, X., Goulden, M.L., Hollinger, D.Y., Barr, A., Black, T.A., Bohrer, G., Bracho, R., Drake, B., Goldstein, A., Gu, L., Katul, G., Kolb, T., Law, B.E., Margolis, H., Meyers, T., Monson, R., Munger, W., Oren, R., Paw, U.K., Richardson, A.D., Schmid, H.P., Staebler, R., Wofsy, S., Zhao, L., 2011. Observed increase in local cooling effect of deforestation at higher latitudes. *Nature* 479, 384-387.
- Lehsten, V., Groot, W.J., Flannigan, M., George, C., Harmand, P., Baltzer, H., 2014. Wildfires in boreal ecoregions: Evaluating the power law assumption and intra-annual and interannual variations. *Journal of Geophysical Research: Biogeosciences* 119, 14-23.
- Liu, H., 2005. Changes in the surface energy budget after fire in boreal ecosystems of interior Alaska: An annual perspective. *Journal of Geophysical Research* 110, D13101.
- Lü, A., Tian, H., Liu, M., Liu, J. & Melillo, J. M. 2006. Spatial and temporal patterns of carbon emissions from forest fires in China from 1950 to 2000. *Journal of Geophysical Research* 111, D05313.
- McGuire, A.D., Anderson, L.G., Christensen, T.R., Dallimore, S., Guo, L., Hayes, D.J., Heimann, M., Lorenson, T.D., Macdonald, R.W., Roulet, N., 2009. Sensitivity of the carbon cycle in the Arctic to climate change. *Ecological Monographs* 79, 523-555.
- Morton, D.C., Defries, R.S., Randerson, J.T., Giglio, L., Schroeder, W., Van Der Werf, G.R., 2008. Agricultural intensification increases deforestation fire activity in Amazonia. *Global Change Biology* 14, 2262-2275.
- Piao, S., Nan, H., Huntingford, C., Ciais, P., Friedlingstein, P., Sitch, S., Peng, S., Ahlstrom, A., Canadell, J.G., Cong, N., Levis, S., Levy, P.E., Liu, L., Lomas, M.R., Mao, J., Myneni, R.B., Peylin, P., Poulter, B., Shi, X., Yin, G., Viovy, N., Wang, T., Wang, X., Zaehle, S., Zeng, N., Zeng, Z., Chen, A., 2014. Evidence for a weakening relationship between interannual temperature variability and northern vegetation activity. *Nat Commun* 5, 5018.
- Potter, S., Cooperdock, S., Veraverbeke, S., Walker, X., Mack, M. C., Goetz, S. J., Baltzer, J., Bourgeau-Chavez, L., Burrell, A., Dieleman, C., French, N., Hantson, S., Hoy, E. E., Jenkins, L., Johnstone, J. F., Kane, E. S., Natali, S. M., Randerson, J. T., Turetsky, M. R., Whitman, E., Wiggins, E., and Rogers, B. M.: Burned Area and Carbon Emissions Across Northwestern Boreal North America from 2001–2019, *EGUsphere* [preprint], <https://doi.org/10.5194/egusphere-2022-364>, 2022.

- Reich, P.B., Bermudez, R., Montgomery, R.A., Rich, R.L., Rice, K.E., Hobbie, S.E., Stefanski, A., 2022. Even modest climate change may lead to major transitions in boreal forests. *Nature* 608, 540-545.
- Rogers, B.M., Randerson, J.T., Bonan, G.B., 2013. High-latitude cooling associated with landscape changes from North American boreal forest fires. *Biogeosciences* 10, 699-718.
- Rogers, B.M., Soja, A.J., Goulden, M.L., Randerson, J.T., 2015. Influence of tree species on continental differences in boreal fires and climate feedbacks. *Nature Geoscience* 8, 228-234.
- San-Miguel-Ayanz, J., Moreno, J.M., Camia, A., 2013. Analysis of large fires in European Mediterranean landscapes: Lessons learned and perspectives. *Forest Ecology and Management* 294, 11-22.
- Schultz, N.M., Lawrence, P.J., Lee, X., 2017. Global satellite data highlights the diurnal asymmetry of the surface temperature response to deforestation. *Journal of Geophysical Research: Biogeosciences* 122, 903-917.
- Schulze, E.D., Wirth, C., Mollicone, D., von Lüpke, N., Ziegler, W., Achard, F., Mund, M., Prokushkin, A., Scherbina, S., 2012. Factors promoting larch dominance in central Siberia: fire versus growth performance and implications for carbon dynamics at the boundary of evergreen and deciduous conifers. *Biogeosciences* 9, 1405-1421.
- Sitch, S., Smith, B., Prentice, I.C., Arneeth, A., Bondeau, A., Cramer, W., Kaplan, J.O., Levis, S., Lucht, W., Sykes, M.T., Thonicke, K., Venevsky, S., 2003. Evaluation of ecosystem dynamics, plant geography and terrestrial carbon cycling in the LPJ dynamic global vegetation model. *Global Change Biology* 9, 161-185.
- van der Werf, G.R., Dempewolf, J., Trigg, S.N., Randerson, J.T., Kasibhatla, P.S., Giglio, L., Murdiyarso, D., Peters, W., Morton, D.C., Collatz, G.J., Dolman, A.J., DeFries, R.S., 2008. Climate regulation of fire emissions and deforestation in equatorial Asia. *Proc Natl Acad Sci U S A* 105, 20350-20355.
- Van Wagner, C. E. (1987). Development and structure of the Canadian forest fire weather index system (Vol. 35). Retrieved from <http://cfs.nrcan.gc.ca/publications?id=19927>
- White, J.C., Wulder, M.A., Hermosilla, T., Coops, N.C., Hobart, G.W., 2017. A nationwide annual characterization of 25 years of forest disturbance and recovery for Canada using Landsat time series. *Remote Sensing of Environment* 194, 303-321.
- Whitman, E., Parisien, M.A., Thompson, D.K., Flannigan, M.D., 2019. Short-interval wildfire and drought overwhelm boreal forest resilience. *Sci Rep* 9, 18796.
- Zhao, J., Wang, L., Hou, X., Li, G., Tian, Q., Chan, E., Ciais, P., Yu, Q., Yue, C., 2021. Fire Regime Impacts on Postfire Diurnal Land Surface Temperature Change Over North American Boreal Forest. *Journal of Geophysical Research: Atmospheres* 126.
- Zhu, C., Kobayashi, H., Kanaya, Y., Saito, M., 2017. Size-dependent validation of MODIS MCD64A1 burned area over six vegetation types in boreal Eurasia: Large underestimation in croplands. *Sci Rep* 7, 4181.
- Zong, X., Tian, X., Yao, Q., Brown, P.M., 2022. An analysis of fatalities from forest fires in China, 1951–2018. *International Journal of Wildland Fire* 31, 507-517.

Reviewer Reports on the First Revision:

Referees' comments:

Referee #1 (Remarks to the Author):

I thank the authors for their thorough revision to the article. The additional analysis were quite interesting and I believe strengthen the overall work. I have no major concerns left unaddressed at this point.

A few minor suggested changes:

1) Line 189, I am not familiar with fire regimes in China, but certainly the western US and Canada do have vast tracts of sparsely populated forest land. Despite the US having the largest fire suppression agency globally, the complexity of fuels, resource deployment/accessibility/priority, and significant increases in aridity and fire weather have superseded management efforts. These same areas have also seen a build up of fuel due to successful fire suppression in a cooler/wetter climate. My little suggestion is to remove the "in remote regions" since many of fires that have contributed to increased burned area are lightning caused.

2) Line 372, if room allows, perhaps a note that highlights the different time period for each region. Note also that for Canada and the US you are limiting the analysis to large fires only. By contrast, this does not seem true for China or the European countries. Hence, this might lead to a small bias in reporting on fire sizes.

Referee #2 (Remarks to the Author):

I appreciate the authors' effort to improve the original manuscript. However, I am not convinced that the size of the fire is the main driver of the increase in temperature. Forest mortality due to fire is what drives changes in biophysical responses. Therefore, I would argue that fire intensity and severity are the main drivers, which just happened to correlate positively with fire size. Since the authors already calculated FRP and LAI changes, it is unclear why fire size was chosen from the rest of the fire metrics. Fire size can only be a proxy of fire intensity, and there are more uncertainties in fire size data than in FRP. So why not use FRP in the first place?

The authors stated that the increase in temperature can lead to more fires. But if the increase is mostly observed within the first 4 years after the fire, and fires in northern latitude reoccur every 50-100 years, how can this change in temperature be relevant to future fire activity?

Line 170-171: $p=0.15$ does not indicate significance; therefore, why did the authors state an increase in mean fire size? Additionally, according to Extended data Fig 10, "The R^2 between regional surface warming and mean fire size is 0.12 ($n=14$, $p=0.23$, the student's t-test)." Therefore, there is no relationship between regional surface warming and mean fire size. While according to Fig 1 in the main

text, “The scaling relationship between the change in summer (June to August) surface radiometric temperature immediately (i.e., one year) after fire and fire size” is significant. I found these two sentences contradictory.

To understand the significance of the results, I would suggest reporting changes in % of the mean. For example, while the change in temperature is significantly increasing with fire size, how much change are we talking about? Does it change by 10% or 0.01%? At least from the figure, it seems like the change, while significant, is negligible annually.

Why the authors chose a 50x25 km window for unburned control pixels neighboring?

Figure 4: Why are some countries in yellow and others in blue?

Author Rebuttals to First Revision:

Responses to referee comments

We thank both referees for their constructive comments which have helped us to greatly improve the quality of the manuscript. Please find below our point-to-point responses. The substantially revised texts are marked in blue in the revised manuscript to make it easy to track our revisions.

Referee #1 (Remarks to the Author):

I thank the authors for their thorough revision to the article. The additional analysis were quite interesting and I believe strengthen the overall work. I have no major concerns left unaddressed at this point.

[Response] We appreciate the efforts of the referee and the positive overall comment on our study. Please find below our point-to-point responses to your comments.

A few minor suggested changes:

[Comment 1] Line 189, I am not familiar with fire regimes in China, but certainly the western US and Canada do have vast tracts of sparsely populated forest land. Despite the US having the largest fire suppression agency globally, the complexity of fuels, resource deployment/accessibility/priority, and significant increases in aridity and fire weather have superseded management efforts. These same areas have also seen a build up of fuel due to successful fire suppression in a cooler/wetter climate. My little suggestion is to remove the "in remote regions" since many of fires that have contributed to increased burned area are lightning caused.

[Response] The referee's comment made us realize that "remote regions" is indeed not completely correct. Simply removing it as suggested by the referee seems to make the sentence incomplete in its meaning. We hence have changed this sentence to: "*growing burned area and fire size there highlights the difficulty of overcoming climatic drivers and of preventing large fires in sparsely populated areas.*". We hope this is clearer.

[Comment 2] Line 372, if room allows, perhaps a note that highlights the different time period for each region. Note also that for Canada and the US you are limiting the analysis to large fires only. By contrast, this does not seem true for China or the European countries. Hence, this might lead to a small bias in reporting on fire sizes.

[Response] Following the referee's suggestion, we have added "*Note that, due to data availability, different time periods were considered for different countries.*" in the figure caption. Because the time periods for different countries are indicated in each panel of the figure, and because the caption room is limited, we have not listed the time period for each country or country group. We further agree with the referee that including only large fires for

Canada, USA and Australia but all fires for China and European countries can introduce a small bias: to warn the readers on this point, we have added in the figure caption the following text: “*The average fire size is not strictly comparable between countries since for Canada, USA and Australia only large fires (>2 km²) were included while for the European countries and China all fires were included. This limitation does not affect the examination of temporal trends, however.*”

Referee #2 (Remarks to the Author):

[**Comment 3**] I appreciate the authors' effort to improve the original manuscript. However, I am not convinced that the size of the fire is the main driver of the increase in temperature. Forest mortality due to fire is what drives changes in biophysical responses. Therefore, I would argue that fire intensity and severity are the main drivers, which just happened to correlate positively with fire size. Since the authors already calculated FRP and LAI changes, it is unclear why fire size was chosen from the rest of the fire metrics. Fire size can only be a proxy of fire intensity, and there are more uncertainties in fire size data than in FRP. So why not use FRP in the first place?

[Response] We deeply appreciate the efforts of the referee in reviewing our paper and the very valuable comments that have helped greatly enhance the manuscript quality.

We agree with the referee that fire intensity and fire severity co-vary with fire size, and that these could also be important drivers for the observed increasing postfire surface warming with fire size: a fact which we now explain in the revised manuscript (lines 143–171). However, because of the surface roughness effect, fire size is more than just a proxy for fire intensity in driving increasing postfire surface warming. This effect is related to landscape spatial heterogeneity that depends solely on the spatial extent of a burned patch, which can be independent of fire intensity or severity. Following larger fires, a spatially more homogeneous landscape will lead to smaller surface roughness, causing reduced turbulent air movement, less surface heat dissipation through evaporative cooling and hence a larger extent of warming. This effect is termed the ‘direct size effect’ in the revised manuscript. It was only briefly explained in the original manuscript but the explanation has now been substantially strengthened.

Given that our study uses observational records rather than controlled experiments, the effects of fire intensity/severity and fire size can’t be separated using statistical methods based on variance partitioning, such as multiple linear regression, because of the co-varying relationship between them shown by our analysis. However, we were able to find an additional dataset that can be used to set up a pseudo-experiment to tease apart the different effects. This data set, published by White et al. (2017), maps the location and size of

disturbance events of both stand-replacing forest fire and clearcut forest harvest at a 30m spatial resolution across Canada. We now use this dataset in combination with remote sensing data of biophysical variables to show that, for the case of forest harvest, land surface temperature in summer shows a warming effect which scales with harvest patch size similarly as for fire. However, in the case of harvest, the intensity or severity effect is negligible because all the mapped forest harvest events are clearcut harvest. In our opinion, this new analysis provides compelling evidence for the existence of a ‘direct size effect’ in driving post-disturbance (fire or harvest) land surface warming. Moreover, the analysis reveals that the scaling slope for fire is significantly larger than that for harvest, with their difference, accounting for 26–32% of the scaling slope for fire, possibly explained by an additional fire intensity effect (for example, enhanced surface charring in more severely burned and larger patches resulting in more summer warming). These results unequivocally show that for Canadian forest mainly dominated by evergreen needleleaf forests, the observed enhanced surface warming following large fires is predominantly due to a ‘direct size effect’ which is supplemented by additional effects of fire intensity/severity that co-vary with fire size.

In summary, two effects are involved in driving increasing postfire surface warming with fire size: the effect of landscape spatial heterogeneity that depends directly on disturbance patch size (the direct size effect), and the effect of fire behaviour that co-varies with fire size. We agree with the referee that in terms of the second effect described here, fire size could be indeed considered as a proxy. But when both effects are considered, we argue that fire size remains the most inclusive (and the most important according to the Canadian case study described above) variable to cover both effects and hence fire size retains the primary role in our revised manuscript.

On the other hand, our results presented in Fig. 3 do indeed show that when considering a wide range of fire regimes spanning different forest types (in contrast to the aforementioned Canadian case study focusing on evergreen needleleaf forest), both the magnitude of surface warming and its scaling with fire size are modulated by forest type. This observed modulation is consistent with fire severity among different forest types, suggesting a growing importance of fire severity compared to fire size in postfire biogeophysical responses when the share of broadleaf trees increases (Extended Data Fig. 9). This latter finding justifies proposing species selection to mitigate the surface climate impacts of growing forest fire size, which is a key implication of our results for boreal forest fire management.

Revisions in the updated manuscript incorporating the above content include: a) a detailed Supplementary Text 3 (5 pages) for the Canadian case study demonstrating the presence of a ‘direct size effect’ in driving post-disturbance surface warming for both forest harvest and fire and also the additional role of the fire intensity effect that co-varies with fire size. b) A new

section (>500 words) in the main text, entitled “Covariance between ‘direct size effect’ and fire behaviour”, where the two effects driving postfire surface warming are described. c) an additional block of text describing the growing importance of fire severity in driving postfire biogeophysical responses across different forest types and its implication for forest fire management (lines 190–197).

Given the existence of the ‘direct size effect’ in driving postfire surface warming, and considering the limitations in the observations and data availability of FRP, forest mortality and Δ LAI, and further considering that fire size has more direct implications for forest management than FRP or fire severity variables, fire size remains the key explanatory variable used in this study to explore the surface climate impacts of forest fire and the implications of increasing large fires on future climate and fire activity. In addition, the correlation between fire behaviour and fire size means that the effects of fire behaviour are implicitly accounted for when using fire size as the main explanatory variable. These points are now summarized in a block of text in the revised manuscript (lines 167–171). In addition, the advantages of using fire size, rather than FRP, forest mortality or Δ LAI, as the explanatory variable have been further detailed in a 4-page Supplementary Text 4.

[Comment 4] The authors stated that the increase in temperature can lead to more fires. But if the increase is mostly observed within the first 4 years after the fire, and fires in northern latitude reoccur every 50-100 years, how can this change in temperature be relevant to future fire activity?

[Response] We are grateful for the referee’s comment which encouraged us to (re-)think more thoroughly about the implications of our findings, enabling us to arrive at a more balanced view regarding the impacts of increasing forest fire size on future climate and fire activity, as detailed below:

We agree with the referee that on a local scale for a burned forest stand per se, postfire surface warming of a 4-year duration at the annual time scale (for summer in northern forests, which is also the fire season, however, the warming persists over the entire study period of 14 years) won’t have an immediate influence on the next fire at the same site, because it takes more than four years to accumulate enough fuel on the same site to burn again. However, this conclusion relies on two implicit assumptions: (1) a forest ecosystem, rather than grassland or shrub, will regrow on the same site. (2) most likely, a forest stand of the same pre-disturbance tree species will recover, or if not, and if the recovered forest has different tree species than pre-disturbance, it will follow a certain successional pattern to ultimately reach the same pre-disturbance tree species before the next fire occurs. These are the crucial assumptions behind the assumed stable fire cycle, which is 50–100 years in northern-latitude forests.

However, either of these assumptions can become invalid under a changing fire regime including growing fire size, and under the observed postfire biogeophysical feedbacks, including surface warming, a reduction in ET, and reductions in soil moisture as reported by previous studies (Harden et al., 2006; Holden et al., 2015). As we will articulate below, the increasing burned fire patch size per se, and the alteration of postfire environmental conditions, can both have important consequences for tree recruitment and forest recovery (or no-recovery). Although these consequences happen within a few years after fire, they have a decade-long legacy effect on the local fire cycle at a given site.

First, the recovery of a forest after fire depends on propagule (e.g., plant seeds) availability and subsequent germination and establishment. However, propagules reaching burned patches can be reduced for larger burned patches because of a longer dispersal distance (in high-severity fires the soil seed bank could be completely destroyed, or forests can be burned again before reaching reproductive maturity so that seed banks are not yet developed). Hence, forest recovery can be slowed or suppressed, or plants with long distance dispersal capabilities (sometimes non-native ones) or those that can resprout vegetatively, such as shrub, can gain advantages (Gill et al., 2022). In this case, increasing forest fire size can result in delayed or suboptimal forest recovery or even non-recovery, or a recovery with unexpected species. For example, Turner et al., (2003) showed that postfire aspen seedling recruitment in Yellowstone National Park highly depends on site proximity to adult aspen; Harvey et al., (2016) reports that postfire tree seedling establishment declined sharply with increasing distance to seed sources in subalpine forests of the Northern Rocky Mountains in the United States.

Second, postfire surface warming combined with reduced soil moisture, can negatively affect tree recruitment and stand regeneration, and if under drought events accompanied by a heatwave, can cause complete regeneration failure. The warming, combined with drying, can also alter growth rates of seedlings of different competing tree species, leading to changes in species composition of regenerating stands. Both effects occur within a few years after fire but will have consequences for future fire occurrence at the same site. Whitman et al., (2019) report that shortened fire return intervals with postfire warming and drought have reduced the seedling density of regenerating stands in northwestern Canadian boreal forests. Baltzer et al., (2021) show that drier climatic conditions and more severe fires consistently undermine resilience, often resulting in complete regeneration failure in North American black spruce forests. By conducting climate manipulation experiments in southern boreal forest, Reich et al., (2022), show that even modest warming (+1.6 °C, comparable to the reported postfire warming by the largest fires in Fig. 1a, but note that postfire local surface warming is additional to background climate warming caused by anthropogenic carbon emissions, which is already 1–3 °C compared to preindustrial times in northern latitudes according to the 6th IPCC Assessment Report, see Gulev et al., 2021) can substantially increase the juvenile

mortality of all species, especially boreal conifers. As a result, seedling growth of species currently abundant in southern boreal forests was substantially reduced following warming but the seedling growth of currently subordinate species was enhanced — modest warming can thus alter the dominant species of postfire regenerating stands, a consequence that will influence its fire vulnerability.

These local-scale effects mainly operate through postfire forest regeneration, but at a regional scale, the reported enhanced surface warming and reduced evapotranspiration following large fires can reduce regional cloud cover and facilitate the development of droughts and heatwaves (Teuling et al., 2017). The unburned forests neighbouring burned ones can also be heated through a neighbourhood effect. Our additional analysis shows that indeed the land surface does also become warmer over unburned forests close to burned patches and this effect can extend as far as one and a half kilometers from burned perimeters, with a persistent amplifying effect of fire size (Extended Data Fig. 14 in the revised manuscript). These climate feedbacks may contribute to increasing fire risk on a regional scale.

On the global scale, increasing fire size and increasing forest mortality that co-varies with fire size will impact future climate change by promoting carbon release in both live and dead biomass and by promoting postfire permafrost melting and the subsequent release of soil organic carbon in boreal regions. Note that although larger fires also lead to lower surface temperature in winter (i.e., a cooling effect), winter cooling and summer warming can have asymmetric effects on permafrost dynamics. In winter, in high-latitude areas, the whole soil profile starting from the surface is completely frozen (Liu, 2005), and a maximum of -0.6°C cooling has little effect on the already-frozen ground. But in summer, our reported maximum warming reached $+1.6^{\circ}\text{C}$ for the daily mean surface temperature ($+3^{\circ}\text{C}$ for daytime warming, see Supplementary Fig. 2). Field studies have also reported a mean annual ground surface temperature increase of $1\text{--}7^{\circ}\text{C}$ following fire in Alaska, Canada, Siberia and northeast China (Holloway et al., 2020), driving an observed increase in active layer thickness of up to 3 meters that can take decades after fire to recover (Holloway et al., 2020). It is clear that increasing soil temperature and a deepening active layer will promote soil organic matter decomposition and enhance soil CO_2 release (Schuur et al., 2015).

The above processes which promote terrestrial CO_2 release with increasing fire size will contribute to the build-up in atmospheric CO_2 and the resulting climate change. But, on the other hand, surface radiative cooling and atmospheric radiative cooling by fire-emitted aerosols will help to cool the climate. Hence, although the overall effect on climate change might be uncertain, any contributions to climate change will impact future fire activity over a large spatial domain.

The fire-size related processes described above, which contribute to changes in future climate and fire activity, are summarized in Fig.5 in the revised manuscript.

The referee questions our conclusion that “the increase in temperature can lead to more fires”. After revisiting all these processes as described above, we agree with the referee and share the same concern that not all these processes will lead to more fires. Some processes facilitate future fire occurrence (land surface warming and drying; forest regeneration failure), but some processes (e.g., changes in regeneration species) have rather uncertain effects. Hence in the revised manuscript, we have given a balanced description of all these different aspects. Nonetheless, our findings raise the point that these processes linked to increasing fire size can impact surface climate dynamics and fire activity and hence the phenomenon of growing fire size in several countries and its associated biophysical feedbacks should receive due attention.

The considerations discussed above have been integrated as three paragraphs (lines 226–250) in the revised discussion section of the manuscript. Additionally, a new figure (Fig. 5) is included to give a nice summary of the different processes. The abstract has also been revised to reflect the more balanced description of the implications of our findings for future fire dynamics.

[Comment 5] Line 170-171: $p=0.15$ does not indicate significance; therefore, why did the authors state an increase in mean fire size? Additionally, according to Extended data Fig 10, “The R^2 between regional surface warming and mean fire size is 0.12 ($n=14$, $p=0.23$, the student’s t-test).” Therefore, there is no relationship between regional surface warming and mean fire size. While according to Fig 1 in the main text, “The scaling relationship between the change in summer (June to August) surface radiometric temperature immediately (i.e., one year) after fire and fire size” is significant. I found these two sentences contradictory.

[Response] Thanks to the referee for pointing this out. Indeed, $p=0.15$ does not indicate a significant trend in the annual regional mean fire size; neither does $p=0.23$ indicate a significant correlation between the regional mean postfire summer surface warming and the regional mean fire size. After reflection, this block of result has been removed from the revised manuscript. Below we provide the reasoning behind this removal.

We first examined whether the scaling relationship between postfire summer surface warming (ΔT) and the logarithm of fire size ($\log_{10}(\text{size})$) holds for each year of 2003–2016 for the study domain (40°N–70°N) by including all forest types. We found that the scaling relationship holds for all individual years (except for 2004 p -value=0.08), but the scaling coefficient ($\beta_{\Delta T}$) and the intercept of the linear regression (β_0) show considerable variations (Table R1). The coefficient of variation values for $\beta_{\Delta T}$ and β_0 are 33% and 74%, respectively.

Table R1 | The binned regression between postfire summer ΔT and fire size for all individual years of the period 2003–2016. $\beta_{\Delta T}$, β_0 , and R^2 represent the linear regression slope, the intercept of the linear regression and the coefficient of determination, respectively. The p-values are obtained from student's t-test.

Year	$\beta_{\Delta T}$	β_0	R^2	p-value
2003	0.39	0.20	0.89	<0.01
2004	0.13	0.59	0.13	0.08
2005	0.76	0.07	0.92	<0.01
2006	0.62	0.08	0.94	<0.01
2007	0.78	-0.03	0.66	<0.01
2008	0.51	0.11	0.86	<0.01
2009	0.38	0.28	0.76	<0.01
2010	0.59	0.18	0.89	<0.01
2011	0.49	0.16	0.87	<0.01
2012	0.37	0.49	0.95	<0.01
2013	0.69	0.32	0.88	<0.01
2014	0.48	0.23	0.88	<0.01
2015	0.61	0.41	0.95	<0.01
2016	0.48	0.60	0.80	<0.01

However, the existence of the scaling relationship for all individual years, and for the case when data for all years are collectively examined (Fig. 1a in the main text), does not necessarily guarantee a strict linear relationship between the regional mean summer surface warming and the regional mean fire size. The mathematical basis is explained below:

Given that the scaling relationship holds for any of the years of 2003–2016, we have

$$\Delta T_{ij} = \beta_{0j} + \beta_{\Delta Tj} \cdot \log_{10}(\text{size})_{ij} + \varepsilon_{ij} \quad (1)$$

where the index j indicates the year, the index i indicates a given fire patch for the j^{th} year, and ε_{ij} is the random error of the linear regression. As the point located by the coordinate of mean values of the independent variable ($\overline{\log_{10}(\text{size})_j}$) and the dependent variable ($\overline{\Delta T_j}$) must lie exactly on the regression line, we hence have:

$$\overline{\Delta T_j} = \beta_{0j} + \beta_{\Delta Tj} \cdot \overline{\log_{10}(\text{size})_j} \quad (2)$$

For another year k we also have:

$$\overline{\Delta T_k} = \beta_{0k} + \beta_{\Delta T_k} \cdot \overline{\log_{10}(\text{size})_k} \quad (3)$$

By differentiating Equation (2) and Equation (3), and assuming further that $\beta_{0j} = \beta_{0k}$ and $\beta_{\Delta T_j} = \beta_{\Delta T_k} = \beta_{\Delta T}$, we can get:

$$\overline{\Delta T_j} - \overline{\Delta T_k} = \beta_{\Delta T} \cdot (\overline{\log_{10}(\text{size})_j} - \overline{\log_{10}(\text{size})_k}) \quad (4)$$

Equation (4) is the exact equation we need in order to reach a perfect linear relationship between the regional mean summer surface warming and the regional mean fire size. To make this equation valid, we need the two equations below to hold for any given two years of the period of 2003–2016:

$$\beta_{0j} = \beta_{0k} \quad (5)$$

$$\beta_{\Delta T_j} = \beta_{\Delta T_k} \quad (6)$$

For Equations (5) and (6) to be valid, the scaling coefficient ($\beta_{\Delta T}$) and the intercept of the linear regression (β_0) for all individual years need to be identical, which is a very strong condition. But, as shown in Table R1, considerable variations exist in these two parameters, which explains why a significant linear regression between the regional mean summer surface warming and the regional mean fire size was not obtained in our results.

Hence, as is concluded in the original manuscript (lines 167–169 in the original manuscript), the significantly increasing trend in the regional mean value of postfire summer surface warming is more driven by the growing proportion of coniferous forests in terms of burned area rather than the increase in the regional mean fire size, given that the latter is even not statistically significant, perhaps due to both the great interannual variations in fire size and the relatively short time span (14 years). However, when a longer term of >35 years was examined, significant trends in fire size were found on regional scale (Fig. 4 in the main text). Given that our study focuses on size-dependent warming rather than the difference in warming between forest types, we decide to remove this block of result in the revised manuscript.

We also wondered what factors are driving the variations in $\beta_{\Delta T}$ for different years, and we examined the correlation between $\beta_{\Delta T}$ and the regional mean values of FRP, fire-induced forest mortality, and the fraction of burned area covered by needleleaf forests, but no significant correlation was found for any factor. Our results thus confirm the robustness of the scaling relationship between postfire surface temperature change and fire size using diverse datasets, including those presented in Supplementary Text 1, but also show considerable

variations in the scaling coefficient ($\beta_{\Delta T}$) among different years. The driving factors for the latter remain somewhat opaque and will be left for future research.

[Comment 6] To understand the significance of the results, I would suggest reporting changes in % of the mean. For example, while the change in temperature is significantly increasing with fire size, how much change are we talking about? Does it change by 10% or 0.01%? At least from the figure, it seems like the change, while significant, is negligible annually.

[Response] For Fig. 1a in the main text (postfire surface warming in summer one year after fire for N40°–N70°), the mean surface warming (ΔT) for all fire patches is 0.89K. For every doubling in fire size, ΔT would increase by 0.15K, representing an increase of 16.9% of the mean ΔT which is still substantial. If we look at the change in ΔT across the whole gradient of fire size, ΔT increased by 4.6 times (460%) when fire size increased by 2.5 orders of magnitude from 1.25 km² (10^{0.1} km²) to 316 km² (10^{2.5} km²), which is also remarkable.

Table R2 summarizes the information on the average postfire ΔT one year after fire, the increase in ΔT as a percentage of the average ΔT (%) by doubling fire size, and the times of the increase in ΔT by varying fire size by 2.5 orders of magnitude for postfire surface warming for summer and the annual scale for different regions of N40°–N70°, Canada, USA and Australia. The increase in ΔT as a percentage of the average ΔT by doubling fire size has a range of 14.9%–29.3%. The times of the increase in ΔT by varying fire size by 2.5 orders of magnitude has a range of 3.0–17.7 times. These results illustrate remarkable changes in the magnitude of postfire surface warming with fire size.

Table R2 | Average postfire ΔT , the increase in ΔT as a percentage of the average ΔT (%) by doubling fire size and times of the increase in ΔT by varying fire size by 2.5 orders of magnitude.

	Average ΔT (K)		Increase in ΔT by doubling fire size as a percentage of the average ΔT (%)		Times of the increase in ΔT by varying fire size by 2.5 orders of magnitude	
	summer	annual	summer	annual	summer	annual
40°N-70°N	0.89	0.21	16.9	19.0	4.6	6.3
Canada	1.00	0.26	17.0	19.2	4.0	4.3
USA	0.94	0.41	29.8	29.3	17.4	17.7
Australia	0.47	0.47	14.9	17.0	3.0	4.2

In response to the referee's comments, we have added a sentence in the revised manuscript: "*with ΔT increasing by 0.15 \pm 0.01 K, or 17% of the average postfire warming (0.89 K) (Extended Data Table 1), for every doubling in fire size*" in the revised manuscript (line 79). Table R2 is provided as Extended Data Table 1 in the revised manuscript.

[Comment 7] Why the authors chose a 50x25 km window for unburned control pixels neighboring?

[Response] The size of the search window was determined following several previous studies which use similar 'space-and-time' or 'space-for-time' approaches to quantify surface radiometric temperature changes (ΔT) following forest cover change or forest fire disturbance. For example, a distance of 50x50 km was used by Alkama and Cescatti, (2016) in quantifying ΔT following forest cover change; a searching window of 50x28 km (longitude \times latitude) was used by Liu et al., (2019) in quantifying ΔT following fire disturbance; a searching window of 25x25 km was used in Duveiller et al., (2018) in quantifying ΔT following land cover change. Here, alternative searching windows of 25x25 km and 50x50 km were also used to quantify the scaling relationships between postfire summer and annual surface warming to fire size over the whole northern temperate and boreal forest. The scaling coefficients derived using these different search windows were almost identical, when the uncertainty of the regression slope is taken into account, indicating that our results are robust across a four-fold change in the size of the searching window.

The above information is summarised in lines 564–572 in the revised manuscript and in Extended Data Fig. 16.

[Comment 8] Figure 4: Why are some countries in yellow and others in blue?

[Response] The yellow and blue colours were used to differentiate Canada and USA because they are neighbours. In the revised manuscript this colour difference has been removed to avoid potential confusion. In the updated version of Fig. 4, the location of the USA is instead indicated by arrows pointing at Alaska and the continental US.

References

Alkama, R., Cescatti, A., 2016. Biophysical climate impacts of recent changes in global forest cover. *Science* 351, 600-604.

Baltzer, J.L., Day, N.J., Walker, X.J., Greene, D., Mack, M.C., Alexander, H.D., Arseneault, D., Barnes, J., Bergeron, Y., Boucher, Y., Bourgeau-Chavez, L., Brown, C.D., Carriere, S., Howard, B.K., Gauthier, S., Parisien, M.A., Reid, K.A., Rogers, B.M., Roland, C., Sirois, L., Stehn, S., Thompson, D.K., Turetsky, M.R., Veraverbeke, S., Whitman, E., Yang, J., Johnstone, J.F., 2021. Increasing fire and the decline of fire adapted black spruce in the boreal forest. *Proc Natl Acad Sci U S A* 118.

- Duveiller, G., Hooker, J., Cescatti, A., 2018. The mark of vegetation change on Earth's surface energy balance. *Nat Commun* 9, 679.
- Gill, N.S., Turner, M.G., Brown, C.D., Glassman, S.I., Haire, S.L., Hansen, W.D., Pansing, E.R., St Clair, S.B., Tomback, D.F., 2022. Limitations to Propagule Dispersal Will Constrain Postfire Recovery of Plants and Fungi in Western Coniferous Forests. *BioScience* 72, 347-364.
- Gulev, S.K., P.W. Thorne, J. Ahn, F.J. Dentener, C.M. Domingues, S. Gerland, D. Gong, D.S. Kaufman, H.C. Nnamchi, J. Quaas, J.A. Rivera, S. Sathyendranath, S.L. Smith, B. Trewin, K. von Schuckmann, and R.S. Vose, 2021: Changing State of the Climate System. In *Climate Change 2021: The Physical Science Basis. Contribution of Working Group I to the Sixth Assessment Report of the Intergovernmental Panel on Climate Change*. Cambridge University Press, Cambridge, United Kingdom and New York, NY, USA, pp. 287–422.
- Harden, J.W., Manies, K.L., Turetsky, M.R., Neff, J.C., 2006. Effects of wildfire and permafrost on soil organic matter and soil climate in interior Alaska. *Global Change Biology* 12, 2391-2403.
- Harvey, B.J., Donato, D.C., Turner, M.G., 2016. High and dry: post-fire tree seedling establishment in subalpine forests decreases with post-fire drought and large stand-replacing burn patches. *Global Ecology and Biogeography* 25, 655-669.
- Holden, S.R., Berhe, A.A., Treseder, K.K., 2015. Decreases in soil moisture and organic matter quality suppress microbial decomposition following a boreal forest fire. *Soil Biology and Biochemistry* 87, 1-9.
- Holloway, J.E., Lewkowicz, A.G., Douglas, T.A., Li, X., Turetsky, M.R., Baltzer, J.L., Jin, H., 2020. Impact of wildfire on permafrost landscapes: A review of recent advances and future prospects. *Permafrost and Periglacial Processes* 31, 371-382.
- Liu, H., 2005. Changes in the surface energy budget after fire in boreal ecosystems of interior Alaska: An annual perspective. *Journal of Geophysical Research* 110.
- Liu, Z., Ballantyne, A.P., Cooper, L.A., 2019. Biophysical feedback of global forest fires on surface temperature. *Nat Commun* 10, 214.
- Reich, P.B., Bermudez, R., Montgomery, R.A., Rich, R.L., Rice, K.E., Hobbie, S.E., Stefanski, A., 2022. Even modest climate change may lead to major transitions in boreal forests. *Nature* 608, 540-545.
- Schuur, E.A., McGuire, A.D., Schadel, C., Grosse, G., Harden, J.W., Hayes, D.J., Hugelius, G., Koven, C.D., Kuhry, P., Lawrence, D.M., Natali, S.M., Olefeldt, D., Romanovsky, V.E., Schaefer, K., Turetsky, M.R., Treat, C.C., Vonk, J.E., 2015. Climate change and the permafrost carbon feedback. *Nature* 520, 171-179.
- Teuling, A.J., Taylor, C.M., Meirink, J.F., Melsen, L.A., Miralles, D.G., van Heerwaarden, C.C., Vautard, R., Stegehuis, A.I., Nabuurs, G.J., de Arellano, J.V., 2017. Observational evidence for cloud cover enhancement over western European forests. *Nat Commun* 8, 14065.
- Turner, M.G., Romme, W.H., Reed, R.A., Tuskan, G.A., 2003. Post-fire aspen seedling recruitment across the Yellowstone (USA) Landscape. *Landscape Ecology* 18, 127-140.
- Whitman, E., Parisien, M.A., Thompson, D.K., Flannigan, M.D., 2019. Short-interval wildfire and drought overwhelm boreal forest resilience. *Sci Rep* 9, 18796.

Reviewer Reports on the Second Revision:

Referees' comments:

Referee #1 (Remarks to the Author):

The authors addressed my minor concerns. I think the study is both interesting and innovative. In its current version, it should be a high impact study.

There are some additional questions that a study like this raises, which is in the spirit of advancing science that the authors might consider here, but may also be worth additional study separate from this work.

The finding relating to broadleaf trees is not entirely surprising given the differences in flammability/fire-resistance to that of evergreen needleleaf forests. Since climate is a key determinate of forest types, it is still not completely clear to me (1) the degree to which climatology as being aliased for deciduous (e.g., places with wetter summer conditions), and (2) the practical nature in the policy suggestions of increasing broadleaf deciduous is climatically suitability and ecologically congruent with the wider geography.

Referee #3 (Remarks to the Author):

Overall, this manuscript presents convincing evidence of the importance of fire size (and associated fire severity) on the magnitude of post-fire biophysiological effects in northern forests.

The authors have thoroughly revised the manuscript based on the comments of the referees. All major comments of referee 2 have been addressed. In particular, the authors have devised a clever approach to address comment #3 (successfully separating between severity and size effects) using data on harvest patches, and added a well-balanced discussion to address comment #4 about the implications of their findings.

I have a few remaining suggestions for the authors to consider:

(1) Fire size sensitivity analysis: The authors explain in the methods why they are using binned averages for the regression. However, while I see an advantage of using binned averages for display in Figs. 1a, 2 and the associated ED Figs., I suggest to fit them with error bars to display the spread of the data. Furthermore, as shown in ED Figs 18 and 19, using the binned data for the regression analysis introduces a persistent bias towards a larger effect of fire size in all analyses. It also artificially narrows the confidence interval of the regression and increases the coefficient of determination (R^2). I don't see a reason why the original data shouldn't be used for all the regression analyses (incl. the displayed confidence interval), and the binned averages (including error bars) used for display only.

(2) Study area: In Fig. 1c it seems that the analysis is largely based on northern forests and indeed only data few points are contributed by temperate regions? I suggest to exclude Australia from the trend

analysis, or include it fully in all analyses, especially since the authors show a significant effect of forest type on the scaling factor.

(3) Implications: Considering the authors' suggestion of a “helped transition” to more broadleaf species, I’d like to hear their perspective on a couple of possible effects that may be good to consider:

- a. Effects on ecosystem-protected permafrost: Replacing coniferous with broadleaf species in boreal North America can have important implications for underlying permafrost, since conifers there often promote thick organic layers which provide thermal insulation. The authors state that larger fires threaten permafrost stability (which may be true, but is not specifically analysed here), but I wonder how these suggested ecological transitions may influence ecosystem-protected permafrost (see i.e. Ran et al., 2021).
- b. Spring flammability of broadleaf forest: Not a big concern, but I think the issue of spring fires could probably be mentioned. See Parisien et al. (2023) for a potentially useful reference.
- c. North American bias?: Broadleaf replacement is happening naturally in western boreal North America. With Mack et al (2021) and Mekonnen et al (2019) the authors cite two studies focused on Alaska only, so the statement “forest biome as a whole” (L.223) may be a bit of an overstatement - to my knowledge this has not been shown for Siberia yet? It would be interesting to get the author’s perspective on whether introduction of broadleaf species would be desirable also in regions where this is not naturally happening, and possible implications.

L.46: should other gases (i.e. methane) and aerosols (i.e. black carbon) be counted towards this impact, too?

L. 57: to be specific these datasets are not strictly independent, as they are also derived from MODIS

L. 190: I found the reference to fire regimes here a bit confusing as the authors have not specifically analysed fire regimes (there is also no reference to it in Fig. 3). My small suggestion is to take out the term fire regime, as the authors already explain well that forest type can influence the intensity of fires i.e. (as shown in prior studies).

L 201: which of the datasets is based on two decades of data? This is not explained or stated in L751-753 in the methods

L. 206: As explained above, I would suggest to either exclude Australia or extend the whole analysis to it

L214: climatic drivers or the legacy of long-term suppression/fire exclusion?

L. 248: permafrost thaws (melting is only used for ice)

L 435f: Considerable limitations of MCD64 in boreal regions have been shown by (Xu et al., 2022), (Chen et al., 2021) and (Scholten et al., 2022). Not saying that GFA would not be suitable to use, but I suggest to moderate this statement.

Fig. 3: What do the different letters indicate for the significance level? Do they all imply “significant at $P < 0.05$ ”? In that case, why use different letters? Also, it would be good to report the average and spread of fire sizes in the different forest types – could the higher β values for needleleaf forest be related to the fact that they are the only regions that facilitate large fires?

References

Chen, D., Shevade, V., Baer, A. E., & Loboda, T. V. (2021). Missing burns in the high northern latitudes:

The case for regionally focused burned area products. *Remote Sensing*, 13(20), 1–25.

<https://doi.org/10.3390/rs13204145>

Parisien, M. A., Barber, Q. E., Flannigan, M. D., & Jain, P. (2023). Broadleaf tree phenology and springtime wildfire occurrence in boreal Canada. *Global Change Biology*, April, 1–14.

<https://doi.org/10.1111/gcb.16820>

Ran, Y., Jorgenson, T., Li, X., Jin, H., Wu, T., Li, R., & Cheng, G. (2021). Biophysical permafrost map indicates ecosystem processes dominate permafrost stability in the Northern Hemisphere.

Environmental Research Letters, 18(1), 119–123. <https://doi.org/10.1088/1748-9326/ac20f3>

Scholten, R. C., Coumou, D., Luo, F., & Veraverbeke, S. (2022). Early snowmelt and polar jet dynamics co-influence recent extreme Siberian fire seasons. *Science*, 378(6623), 1005–1009.

<https://doi.org/10.1126/science.abn4419>

Xu, W., Scholten, R. C., Hessilt, T. D., Liu, Y., & Veraverbeke, S. (2022). Overwintering fires rising in eastern Siberia. *Environmental Research Letters*, 17(4), 045005. [https://doi.org/10.1088/1748-](https://doi.org/10.1088/1748-9326/ac59aa)

[9326/ac59aa](https://doi.org/10.1088/1748-9326/ac59aa)

Author Rebuttals to Second Revision:

Referees' comments:

[General Response] We thank both referees for their constructive comments which have helped us to improve the quality of the manuscript. Please find below our point-by-point responses.

Besides the revisions made in response to the referees' comments (detailed below), the following revisions have been made in order to allow a better focus on our key findings, given the limited length of the paper:

- (1) The descriptions of postfire winter cooling and the fire-size impact on winter biogeophysical processes have been shortened in the main text and moved to Supplementary Text 2, because we wish to focus on postfire summer surface warming (which results in annual warming) and because the fire-size impact in winter mostly only holds for radiative processes, given that little change occurred in winter non-radiative processes, including latent heat flux (ecosystem evapotranspiration) and the sum of ground heat flux and sensible heat flux.
- (2) The Supplementary text (original Supplementary Text 2.3) detailing long-term changes in postfire biogeophysical processes has been removed as we feel that the descriptions in the main text along with the Extended Data Figures (Extended Data Figs. 3, 4) are already sufficient. Too many Supplementary Texts on unimportant details makes the reading difficult.
- (3) According to the format guidelines of Nature, a maximum of 10 figures are allowed in the Extended Data Figures. We thus retained the most important 10 Extended Data Figures and moved others into the Supplementary Information.
- (4) The results of decreasing burned area and fire size in selected European countries and in China (original main text Fig. 4) have been removed, because we feel that they are not essential to make the point regarding growing fire size (concentrating on growing fire size in USA, Canada and Australia is enough). In order to meet Nature's requirement that a 6-page article can only have 4 display items, we turned the original main text Fig. 4 into Extended Data Fig. 10.
- (5) Fire-caused forest mortality has now been defined as including forest deaths occurring within 1 year after fire in Fig. 2d, to be consistent with the definitions of the other variables in Fig. 2. Given that there is a negligible difference in the forest mortality derived using a 1-year (in the revised manuscript) and that using a 2-year time window (in the original manuscript) (Supplementary Figs. 6,7; lines 484–488 of the revised manuscript), this is a minor change with no essential consequence but it allows the variables in Fig. 2 to be consistently defined.
- (6) The sections of the paper have been re-organized to make each section containing a reasonable amount of content and words, to make the paper easier to follow.

We believe these revisions help to make our paper more concise and more effective in conveying the key findings and their associated implications.

Referee #1 (Remarks to the Author):

The authors addressed my minor concerns. I think the study is both interesting and innovative. In its current version, it should be a high impact study.

[Response] We once again deeply appreciate the referee's effort to review our paper and are grateful for the positive, encouraging general comments. Please see our detailed response to your specific comment below.

[Comment 1] There are some additional questions that a study like this raises, which is in the spirit of advancing science that the authors might consider here, but may also be worth additional study separate from this work.

The finding relating to broadleaf trees is not entirely surprising given the differences in flammability/fire-resistance to that of evergreen needleleaf forests. Since climate is a key determinate of forest types, it is still not completely clear to me (1) the degree to which climatology as being aliased for deciduous (e.g., places with wetter summer conditions), and (2) the practical nature in the policy suggestions of increasing broadleaf deciduous is climatically suitability and ecologically congruent with the wider geography.

[Response] In response to the referee's comment, to explore possible climate control of forest distribution in our study domain, we analyzed changes in the mean monthly precipitation and surface air temperature during the growing season (April to October) over 1991–2020 (a period length of 30 years used to represent climatology) with increasing coverage of deciduous broadleaf (DBF) and mixed (MF) forests over our study domain of 40°N–70°N as a whole, and for the continents of North America and Eurasia separately (Fig. R1).

Figure R1 | Changes in growing-season (April to October) climate with increasing coverage of deciduous broadleaf forests (DBF) and mixed forests (MF) in northern forests of 40°N–70°N. Monthly mean precipitation and surface air temperature are shown for northern forests (a, b), North America (c, d) and Eurasia (e, f), respectively. Monthly precipitation and surface air temperature datasets with a spatial resolution of 0.5° published by the Climate Research Unit, University of East Anglia (CRU TS 4.07), were aggregated to 2°×2° using the average value of 0.5° cells within each 2° grid cell. For each 2° grid cell with a forest cover greater than 10%, forest coverage by DBF and MF, as well as the annual average values of climate variables for 1991–2020, were calculated. Subsequently, all 2° grid cells across the study domain were grouped into different bins with intervals of 10% based on the forest coverage of DBF and MF. Numbers on boxplots indicate sample size (i.e., the number of 2° grid cells). The center line of the boxplots represents the median value, with box limits indicating upper and lower quartiles and whiskers showing 1.5 × interquartile range.

Indeed, as pointed out by the referee, the growing coverage of broadleaf-containing forests largely corresponds to increasing precipitation and temperature during growing season, suggesting a top-down control of climate over forest distribution. Hence, we agree with the referee that the management suggestion of increasing broadleaf species is only applicable when the local climate is suitable.

On the other hand, we also found a large variation in the proportion of broadleaf-containing forests under similar climate conditions (Fig. R2), which does suggest the possibility of modifying forest composition at the local scale.

Figure R2 | Changes in the proportion of deciduous broadleaf forests (DBF) and mixed forests (MF) with growing-season (April to October) climate conditions in northern forests of 40°N–70°N. The analysis was carried out for northern forests as a whole (a, b), North America (c, d) and Eurasia (e, f), respectively.

Monthly precipitation and surface air temperature datasets with a spatial resolution of 0.5° published by the Climate Research Unit, University of East Anglia (CRU TS 4.07), were aggregated to 2°×2° using the average value of 0.5° cells within each 2° grid cell. For each 2° grid cell, the annual average values of climate variables for 1991–2020 were calculated. Subsequently, all 2° grid cells across the study domain were grouped into different bins with intervals of 20 mm month⁻¹ for monthly precipitation and 2 °C for surface air temperature, both during the growing season. The forest coverage by DBF and MF of 2° grid cells within each climate bin were then used as samples to make the boxplots. Numbers on boxplots indicate sample size (i.e., the number of 2° grid cells). The center line of the boxplots represents the median value, with box limits indicating upper and lower quartiles and whiskers showing 1.5 × interquartile range.

The practicality of the management suggestion to increase broadleaf species in boreal forests is supported by several aspects as detailed below:

- (1) Besides the top-down control of overall climate conditions, the distribution of broadleaf and mixed forests in boreal forests also depends on local topography and disturbance history, including fires. For example, in Canadian boreal forests, black spruce forests are typically found in lowland areas with poor drainage, while white spruce, mixed forests, and broadleaf forests prefer upland locations with better drainage, and jack pine forests dominate at xeric sites on sandy soils (Krawchuk et al., 2006; Schneider, 2013). Fire might be the single most important disturbance driving vegetation dynamics in boreal forests given its widespread occurrence, although wind, insect and drought disturbances also happen. Forest fire is an important ecological factor controlling both environments for vegetation establishment and vegetation succession in boreal regions (Abaimov et al., 2002). Following low- and medium-severity fires, coniferous stands often regenerate with the same species. However, following high-severity fires, increasing abundance of broadleaf forests has been reported for Alaskan (Beck et al., 2011; Mack et al., 2021), Canadian (Massey et al., 2023), and central Siberian forests (Schulze et al., 2005; Tautenhahn et al., 2016) based on field observations at postfire chronosequences and satellite observation. In addition, the phenomenon that enhanced biomass burning was followed by an expansion of broadleaf species in the boreal forests of North America and western Siberia was also revealed by centennial to millennial changes in vegetation and fire frequency archived in charcoal and pollen records in peat and lacustrine sediments (Feurdean et al., 2020; Kelly et al., 2013).
- (2) Nearly two thirds of boreal forests are managed, with these managed forests being mainly distributed in the warmer and wetter southern parts of the biome (Gauthier et al., 2015). These managed areas are also the places which are likely suitable for broadleaf forest growth from a climate-constraint point of view. In fact, for Russia and Canada, about 20% of industrial roundwood production is currently from hardwood (FAOSTAT, <https://www.fao.org/faostat/en/#data/FO>). Considering that increasing fires over the past two decades have augmented the loss of timber-producing forests globally (Bousfield et al., 2023), expanding the broadleaf fraction in the managed forests of the southern boreal zone might be a feasible option to curb

such losses. This would, however, result in an increasing share of hardwood in the wood production from these regions or a decrease in total wood production if all deciduous forests are solely used as fire breaks. In Canada, proposals to increase broadleaf forest as a management option to reduce fire risk have been made at both national and local scales. Girardin and Terrier (2015) suggested that a conversion rate of 0.1 to 0.2% year⁻¹ from coniferous to broadleaf forests, starting from 2020, would be sufficient to maintain burn rates at a constant level across much of the southern Canadian boreal forest. Such a conversion rate is similar to the change rate in broadleaf proportion caused by harvesting and industrialization during recent decades in Canadian boreal landscapes. Hirsch et al. (2004) showed that for a local forest area (300,000 ha) in west-central Alberta operated by a local sawmill and pulp mill company, increasing the proportion of broadleaf forest could reduce fire size.

- (3) The management suggestion of increasing broadleaf forest in southern boreal forests is also consistent with the predicted shift in forest composition driven by climate warming in this region. Several modelling studies have shown that future climate warming in boreal regions would promote the growth of broadleaf species in southern boreal forests, for example, the boreal forests of Alberta, Canada (Stralberg et al., 2018) and central Siberia (Williams et al., 2023). Although such a simulated forest composition shift is sometimes catalyzed by an interactive fire effect, which is supported by observations (described above), the model simulations also point to a negative feedback effect, tending to reduce fires, because of the low flammability of broadleaf trees. The latter effect is supported by reconstructions of forest composition and fire frequency over past centennial to millennial scales based on charcoal and pollen records from lake sediments over North American boreal forest (Girardin et al., 2013; Kelly et al., 2013). Hence, there is a fair chance that species conversion could be actively employed to reduce fire risk over southern managed boreal forests, before the occurrence of the highly severe fires that are needed for such conversion to happen naturally.
- (4) In the southern boreal forests, grassland and shrubland may increase, or even displace trees, on the southern edges of the boreal forest as a result of increasing drought under future warming (Kharuk et al., 2023; Stralberg et al., 2018). Such an increase of grassland and shrubland will promote fire occurrence. It has been demonstrated that an earlier snowmelt, caused by climate warming, promotes in particular human-caused spring forest fires in broadleaf-dominated forests in the southern part of the boreal forests (as noted by Referee #3, see our response there). However, there may be no easy, single solution of vegetation management to address all upcoming changes and their associated fire risks, although it seems safe to state that effective fire detection, fire suppression, and more stringent regulations to control human ignitions will definitely help. Increased grazing might be a potential vegetation management-based option to reduce the fire risk associated changes in fuel load (i.e., woody to grassy), but a detailed exploration of this point is beyond our current scope.

In our original manuscript (line 224), the two references (Mack et al., 2021; Mekonnen et al., 2019) used to support an increasing abundance of deciduous broadleaves after conifer forest fires are both for Alaskan forests. Referee #3 raised the question whether this is also true for Canadian forests and Siberian forests. While we found references supporting large-scale transition to broadleaf forests after fire in Canadian forests (Massey et al., 2023), the same phenomenon has been reported only for local scales in Siberian dark taiga (Schulze et al., 2005; Tautenhahn et al., 2016), although widespread broadleaf regeneration after fire, but not yet a complete species transition, was also reported in Siberian light taiga. As this is related to the feasibility of broadleaf species from a climate point of view and their establishment after fire, we kindly invite the referee to read also our detailed responses to the Comment 3 raised by Referee #3.

A key moderation in the revised manuscript, as a result of addressing this comment, is that we now strictly limit our management suggestion to warmer and wetter southern boreal forests which are under human management. Given the word limits of the paper, the aspects described above are summarized in the revised manuscript as (lines 205–217): “

Considering that about two-thirds of boreal forests, mainly in the warmer and wetter southern areas, are under human management³⁴, increasing broadleaf species at the time of stand regeneration seems a practical alternative to active fire suppression. Moreover, field- and remote sensing-based observations have shown that the ongoing intensification of the fire regime in North American boreal forests has led to an increased abundance of deciduous broadleaves^{35,36} at the cost of evergreen conifers, resulting in a natural control on fire frequency^{37,38}.

Nonetheless, given that the fire-mitigation effect of broadleaf species is mostly documented for North American crown fire systems^{10,39} and plausible in the Eurasian dark taiga where crown fires dominate⁴⁰, the effect of increasing broadleaves on fire frequency in light taiga, where surface fires dominate, requires further research. In addition, cautions should be taken, or careful local scale planning is needed, to increase the share of broadleaves in areas where broadleaf forests are a risk for human-dominated spring fires⁴¹, or where permafrost stability is partially maintained by the protective thick organic layer beneath needleleaf forests⁴².”

Referee #3 (Remarks to the Author):

Overall, this manuscript presents convincing evidence of the importance of fire size (and associated fire severity) on the magnitude of post-fire biophysiological effects in northern forests.

The authors have thoroughly revised the manuscript based on the comments of the referees. All major comments of referee 2 have been addressed. In particular, the authors have devised a clever approach to address comment #3 (successfully separating between severity and size effects) using data on harvest patches, and added a well-balanced discussion to address comment #4 about the implications of their findings.

[Response] We appreciate the referee's effort to review our paper and are grateful for the positive, encouraging general comments. Please see below our detailed responses to your comments, which have helped us to improve the quality of the manuscript.

I have a few remaining suggestions for the authors to consider:

[Comment 1] Fire size sensitivity analysis: The authors explain in the methods why they are using binned averages for the regression. However, while I see an advantage of using binned averages for display in Figs. 1a, 2 and the associated ED Figs., I suggest to fit them with error bars to display the spread of the data. Furthermore, as shown in ED Figs 18 and 19, using the binned data for the regression analysis introduces a persistent bias towards a larger effect of fire size in all analyses. It also artificially narrows the confidence interval of the regression and increases the coefficient of determination (R^2). I don't see a reason why the original data shouldn't be used for all the regression analyses (incl. the displayed confidence interval), and the binned averages (including error bars) used for display only.

[Response] Binned regression is frequently used to reduce the variation in the dependent variable and has been used to examine land surface temperature response to forest cover change (e.g., Alkama and Cescatti, 2016 in Science; Zhang et al., 2024 in Nature geosciences). Although the use of binned regression in this study does not fundamentally change the significance of fire size impact, we agree with the referee that it can mask underlying variations, and lead to artificially boosted R^2 and overly narrow confidence intervals (although, in our case, the confidence interval is mostly very little affected).

In the original manuscript, because of the reduction in variation due to binning, we were able to show general trends of surface warming with fire size initially and only later reveal one of the main drivers, i.e., the species composition of the forest. If we drop the binning, the variation is apparent immediately, so the only solution is to show spatial maps such that the spatial variation of the fire size impact is demonstrated right from the start. For this reason, in response to the referee's (and the editor's) suggestion, in the revised manuscript we have replaced almost all the binned regressions with spatially explicit original regressions within each 2° by 2° grid cell, except for a few occasions in the Supplementary figures (Supplementary Figs. 5, 6 and 25), where binned regressions are kept to enable a clear comparison of regressions using different datasets. In these cases, the slopes derived by both binned and original regressions are provided.

Below we summarize the major changes:

(1) The binned regression between summer ΔT and the logarithm of fire size, pooling all fire patches of the whole study domain of 40°N–70°N (original Fig. 1a), has been removed. The original regression (without binning) yields a $\beta_{\Delta T}$ significantly larger than zero ($p < 0.01$) with an R^2 of 0.07. The low R^2 and a large regression residual in ΔT is not that surprising, given that the fires in many grid cells in Fig. 1c, dominated by

mixed and deciduous broadleaf forests, show a negligible or insignificant fire size impact (i.e., $\beta_{\Delta T}$ is modulated by forest type). In contrast, for some North American and eastern Siberian 2° grid cells, which are dominated by needleleaf forests, R^2 values of the bi-variate regressions between summer ΔT and fire size can reach as high as 0.5 (revised Figs. 1c, 1d). This indicates that spatial variation which co-occurs with forest type distribution is an important source of the overall large spread in summer ΔT . Hence, we have decided that it's better to show the spatial distribution of $\beta_{\Delta T}$, since the large spread in ΔT is likely contributed to by variations in the fire size effect across space and among different forest types, and by co-varying factors including fire intensity (FRP) and fire severity (ΔLAI and forest mortality).

Indeed, a multiple linear regression model including fire size, forest type and their interactions shows a statistically significant fire size effect and yields an R^2 of 0.31 (Fig. 3f), which is still a decent value for a large-scale study such as this. The multiple linear regression model including fire size and co-varying fire intensity (FRP) and fire severity (ΔLAI and forest mortality) variables, which still shows a statistically significant effect of fire size, yields an R^2 of 0.47 (revised Supplementary Table 6), highlighting the contributions of variables other than fire size to the large spread in ΔT .

In addition, the R^2 of simple regressions ($\Delta T \sim$ fire size) derived by using GFA data seems to be also influenced by the data quality: for roughly the same region covered by boreal forests in US Alaska and western and central Canada, the three regional datasets show much larger R^2 values ($R^2=0.35-0.46$) than the two global datasets ($R^2=0.08-0.11$), including GFA, which is used in the main analysis of this study (Supplementary Figs. 10–12).

(2) Based on similar considerations to those outlined above, all binned regressions in Fig. 2 were replaced with spatial maps of the regression slope of fire size derived by applying original regressions (without binning) performed within each 2° grid cell. Please refer to the revised Fig. 2.

(3) The impact of fire size (β values) for different variables and different forest types in Fig. 3 (panels b, c, d & f) are now derived using multiple linear regression models based on original data (without binning). Please refer to the revised Fig. 3.

(4) Almost all Extended Data figures and Supplementary figures now use regressions based on original data (without binning), except for a few cases (Supplementary Figs. 5, 6 and 25) where binned regressions are used to enable a clear demonstration of the comparison of regressions using different datasets. In these cases, the slopes derived by both binned and original regressions are provided.

Overall, replacing binned regression with original regression maintains the robustness of our findings and conclusions and better highlights the spatial structure of the results which in turn is strongly driven by forest type. Hence the use of original regression provides more informative results than using binned regression.

[Comment 2] Study area: In Fig. 1c it seems that the analysis is largely based on northern forests and indeed only data few points are contributed by temperate regions? I suggest to exclude Australia from the trend analysis, or include it fully in all analyses, especially since the authors show a significant effect of forest type on the scaling factor.

[Response] We agree with the referee that the study domain of ‘northern temperate and boreal forest (40°N–70°N)’ (considerations for its selection are explained in lines 59–67) mainly contains boreal forest. The ‘temperate’ forests in this study domain are mainly those in western US and western Canada, and in western and central Europe. Although some of these regions belong to ‘boreal’, as defined in terms of latitudinal range (see below), from a climatic point of view they are actually ‘temperate’.

We have noticed that, in the literature, the definition of ‘northern lands’ is a little arbitrary, but most studies use a separating parallel falling between 30°N and 50°N. For example, Rogers et al. (2015) defined the ‘boreal region’ as land >40°N; Piao et al. (2014) defined ‘northern lands’ as land >30°N; McGuire et al. (2009) and Hayes et al. (2011) defined the ‘pan-arctic region’ or ‘northern high-latitude land’ as land >45°N. These examples support our definition of ‘northern temperate and boreal forests’ as land >40°N. Large tracts of temperate forests outside the domain of 40–70°N, but with noticeable climate-driven forest burned areas, are found in continental USA and Australia, justifying their inclusion in our analysis (the reason for including Australia is detailed below).

We prefer to keep Australia in both the trend analysis of fire size/burned area, and in the analysis of fire size impact on postfire ΔT , based on the following considerations: (1) Growing fire size in Australia is an important piece of evidence demonstrating widespread changes in fire size in temperate and boreal forests. (2) Unlike northern forests with large snow cover in winter, Australia provides a unique case showing how, when the winter snow albedo effect is weak, postfire surface temperature change could be positive even in winter, and is still amplified by fire size.

Both points of Australian results further strengthen the implications of our findings. Nonetheless, we agree with the referee that including Australia could lead to confusion as regards the forest management implications of our results. To minimize such potential confusion, in the revised manuscript, (1) we make clear that the management suggestions only apply in southern boreal forests which are under human management (lines 205–207), (2) the presence of the Australian analysis has been reduced to a minimum and its purpose clearly stated in lines 192–197: “Given that the reported amplified land surface warming by fire size also holds for the

central to western continental USA and Australia (Supplementary Figs. 2, 3), the implications of growing fire size on surface climate may extend beyond northern forests to other regions where climate warming can increase large fires (Australia was included in this part of the analyses only to make this point)”.

We hope the explanations given above and the associated revisions are acceptable. If not, we are happy to completely remove the Australian analysis from the main text and include it only in Supplementary Information.

[Comment 3] Implications: Considering the authors' suggestion of a “helped transition” to more broadleaf species, I’d like to hear their perspective on a couple of possible effects that may be good to consider:

a. Effects on ecosystem-protected permafrost: Replacing coniferous with broadleaf species in boreal North America can have important implications for underlying permafrost, since conifers there often promote thick organic layers which provide thermal insulation. The authors state that larger fires threaten permafrost stability (which may be true, but is not specifically analysed here), but I wonder how these suggested ecological transitions may influence ecosystem-protected permafrost (see i.e. Ran et al., 2021).

b. Spring flammability of broadleaf forest: Not a big concern, but I think the issue of spring fires could probably be mentioned. See Parisien et al. (2023) for a potentially useful reference.

c. North American bias?: Broadleaf replacement is happening naturally in western boreal North America. With Mack et al (2021) and Mekonnen et al (2019) the authors cite two studies focused on Alaska only, so the statement “forest biome as a whole” (L.223) may be a bit of an overstatement - to my knowledge this has not been shown for Siberia yet? It would be interesting to get the author’s perspective on whether introduction of broadleaf species would be desirable also in regions where this is not naturally happening, and possible implications.

[Response] We are grateful to the referee for making these points. They have helped us improve the management suggestions and avoid their potential drawbacks. As a result of addressing these comments, the concerning texts have been revised in the manuscript as follows (lines 205–217): “

Considering that about two-thirds of boreal forests, mainly in the warmer and wetter southern areas, are under human management³⁴, increasing broadleaf species at the time of stand regeneration seems a practical alternative to active fire suppression. Moreover, field- and remote sensing-based observations have shown that the ongoing intensification of the fire regime in North American boreal forests has led to an increased abundance of deciduous broadleaves^{35,36} at the cost of evergreen conifers, resulting in a natural control on fire frequency^{37,38}.

Nonetheless, given that the fire-mitigation effect of broadleaf species is mostly documented for North American crown fire systems^{10,39} and plausible in the Eurasian dark taiga where crown fires dominate⁴⁰, the effect of increasing broadleaves on fire frequency in light taiga, where surface fires dominate, requires further research. In addition, cautions should be taken, or careful local scale planning is needed, to increase the share

of broadleaves in areas where broadleaf forests are a risk for human-dominated spring fires⁴¹, or where permafrost stability is partially maintained by the protective thick organic layer beneath needleleaf forests⁴².”.

The brief revisions above, mainly due to the word limits of the paper, are based on our responses to each point raised by the referee detailed below.

a) We understand the referee’s concern, which we think may go beyond ‘ecosystem-protected’ permafrost to also include at least the ‘ecosystem-driven’ permafrost. According to Ran et al. (2021), ecosystem plays an important protective role in both permafrost types. Following the suggestion of Referee #1, we now limit our management suggestion of increasing broadleaf species to only managed boreal forests with a suitable climate for broadleaf establishment. Such forests are largely restricted to the southern boreal forests (Gauthier et al., 2015). While ‘ecosystem-protected’ permafrost makes up a relatively small fraction of the whole circumpolar permafrost, being mainly distributed along its southern edge, ‘ecosystem-driven’ permafrost occurs over large areas of managed boreal forests, mainly in Northwestern Canada and southern Siberia.

Field studies show that high-severity fires in North American boreal forests promote a transition from pre-fire coniferous dominance to postfire broadleaf colonization mainly because the fire-caused removal of the organic soil layer exposes mineral soil, which facilitates the germination of wind-dispersed broadleaf seeds including aspen and birch (Barrett et al., 2011; Johnstone et al., 2010). Hence, in practice, species conversion through forest management might involve partial removal of the organic layer followed by the establishment of broadleaf trees. Furthermore, the leaf litter of broadleaf trees decomposes more easily than that of conifers leading to a slower build-up of surface organic matter. Both aspects tend to reduce the thermal insulation of the soil organic layer which provides protection for the underlying permafrost. Hence, we agree that the referee’s concern is valid and should be discussed.

However, we argue that the negative effect on permafrost stability of increasing the share of broadleaves could be addressed by proper measures of establishing broadleaves or their compensatory effects: 1) the conversion to broadleaf does not have to be 100% over a large spatial extent, a practice which would severely reduce the protective role of the soil organic layer. Instead, if bands or paths of coniferous forests were to be harvested, the subsequent establishment of broadleaf trees in these strips could then serve as fire breaks to reduce or stop fire propagation in an otherwise homogeneous coniferous landscape. For example, aspen stands have been proposed being used for fuel break purposes in western US Rocky Mountain forests (DeByle et al., 1987; Fechner and Barrows, 1976). Such a practice would likely maintain the protective role of soil organic layer on the landscape scale. 2) broadleaf trees have a higher all year-round albedo than conifers and their radiative cooling effect might help to partly mitigate the (partly) lost protective role of the soil organic layer for underlying permafrost.

Nonetheless, the referee's comment served to make clear to us that our management suggestion of increasing broadleaf trees required a comprehensive evaluation. In particular, we saw a need to 1) compare the trade-off between the (reduced) fire risk and the associated permafrost dynamics with and without species conversion, 2) to cater for local conditions such as topography and permafrost stability when implementing the management suggestion.

b) We thank the referee for prompting us to be more cautious regarding the potential undesirable effects of increasing broadleaf trees in southern boreal forests. We agree with the referee that although broadleaf trees can, in general, reduce fire risk and fire spread, typically in mid-to-late fire season (summer and autumn) (Cumming, 2001; Nesbit et al., 2023), they don't play the same role in spring, in particular before leaf out. This point is now mentioned in the revised manuscript (see above).

c) In response to the referee's comment, we have removed the expression 'forest biome as whole' in the revised texts. We now limit the observation-supported, large-scale transition to broadleaf forests after fires to North American boreal forests, with references for both Alaskan and Canadian forests (Massey et al., 2023).

As for the situation in Siberia, following extensive literature survey (detailed below), our general conclusion is that indeed broadleaf replacement of conifers after fire at the whole stand scale is relatively rare in Siberia, but postfire broadleaf regeneration and establishment is rather widespread in all types of coniferous forests, mainly supported by field investigation and small-scale satellite imaging. The establishment of broadleaf species after fire as a natural process in Siberia suggests the possibility of a 'helped transition' to broadleaf forests, or the possibility of increasing broadleaf share to expand the area of mixed forests. An increased broadleaf share can largely help reduce the crown fire risk in dark taiga, but whether it can also reduce surface fires in light taiga remains unclear. However, from a biogeophysical point of view (i.e., postfire change in land surface temperature), our results indeed show that an increasing broadleaf share can reduce postfire surface warming and the associated fire size impact, even if it cannot effectively reduce the surface fire risk. As most fires in southern managed forests in Russia are caused by human (Kharuk et al., 2021; Mollicone et al., 2006) and there is chance that these fires could be mitigated by fire control or fuel treatment, we conclude that increasing broadleaf in Siberian managed dark taiga is probably feasible, but for light taiga where surface fire prevails, more research is needed to study its feasibility. Below we provide the details that lead to our conclusion.

1. Birch and aspen as 'fire invaders' and 'fire endurers' in Siberian boreal forests

Birch (*Betula spp.*) and aspen (*Populus spp.*) species are considered as 'invaders' and 'endurers' in boreal forests in terms of their response to fire. This seems being true in both North American and Siberian boreal forests as this is largely governed by their biological and ecological nature. Birch and aspen have winged or

hair-coated small seeds, which are dispersed long distance with wind and germinate best on exposed mineral and nutrient rich soils (Nikolov and Helmisaari, 1992). Their light-demanding nature means that their saplings need open space to grow. The fact that both conditions are made available by forest fires – open space and nutrient rich mineral soil – makes them pioneer species after fire disturbance, and in particular, after high-severity stand-replacing fires, in boreal forests (Johnstone et al., 2010; Schulze et al., 2005). They are hence considered as ‘invaders’ as they are able to rapidly colonize postfire open space (Wirth, 2005). In addition, both aspen and birch (including *Populus tremula* and *Betula pendula* in Siberia) are able to re-sprout after fire. In Siberia, *Betula* sp. can only resprout from stumps, whereas *Populus tremula* also has root suckers (Tautenhahn, 2014). These characteristics allow them ‘endure’ fire disturbances by quickly resprouting after fire, another reason making them early successional species after fire (Wirth, 2005).

The major species of *Betula* in Siberia are *Betula pubescens* and *Betula pendula*; the major aspen species is *Populus tremula*. These three species are widely distributed across almost the entire Eurasia. Kharuk et al. (2021) indicated that out of roughly 6 million km² forested area of Siberia, broadleaf birch and aspen cover ~670,000 km² and ~120,000 km², respectively. The broadleaf forest area is hence comparable with those of ‘dark taiga’ (~760,000 km²) and Scots pine stands (one form of ‘light taiga’, ~860,000 km²), two major conifer forests in Siberia, although the largest area is occupied by larch-dominated ‘light taiga’ (about 3,000,000 km² including sparse stands). This fact shows widespread ‘natural’ distribution of broadleaf trees in Siberia.

2. Regeneration of broadleaf species as a natural process in postfire succession of boreal Siberian forests

Broadleaf forests are largely concentrated in southern Siberia (our main text Fig. 2e), which is also shown by Kharuk et al. (2021, their Fig. 1), probably because broadleaves favour warmer areas (Fig. R1 in this response letter, see also our response to the comment of Referee #1). However, birch and aspen trees are also found widespread in northern Siberia, in particular in north-western and north-central Siberia but also in larch-dominated forests in north-eastern Siberia (Fig. 1 in Kharuk et al., 2021). Their presence, mostly due to establishment after fires, is confirmed by field observations spanning different forest types across the whole Siberia, including ‘dark taiga’ in western and central Siberia (Schulze et al., 2005; Tautenhahn et al., 2016), the transitional zone from dark taiga to light taiga in central Siberia (Schulze et al., 2012), larch-dominated ‘light taiga’ in central Siberia (Kharuk et al., 2016; Kharuk et al., 2011) and southeastern Siberia (Uemura et al., 1997; Zyryanova et al., 2007).

Western to central Siberia are dominated by ‘dark taiga’. The dominant evergreen coniferous species (*Pinus sibirica*, *Abies sibirica*, and *Picea obovata*) are considered as ‘fire avoiders’, which rarely have fires, but sustain crown fires under extreme drought and are easily killed by fire (Wirth, 2005). Schulze et al. (2005) used postfire chronosequence to show that after stand-replacing fires in dark taiga, stand succession follows a

typical sequence of *Betula/Poplar* → *Betula/Poplar-conifer* transition → Mixed boreal forest → *Picea-Abies*, very similar to the typical stand succession in North American boreal forests following crown fires. These successions were recently confirmed by Tautenhahn et al. (2016) using small-scale satellite imaging combined with field investigation. Tautenhahn et al. (2016) pointed out three possible forest succession trajectories in central Siberian dark taiga: ‘Evergreen self-replacement’ (regeneration of pre-fire species), ‘Deciduous to evergreen’ (the typical successional trajectory revealed by Schulze et al., 2005) and ‘Extended deciduous’ (complete replacement of evergreen conifer by broadleaf followed by its long-term persistence). In an additional simulation experiment they showed that under increasing fire severity and fire size, which are probable under climate warming, the trajectory of ‘Extended deciduous’ will become dominant because evergreen coniferous species rely on short-distance seed dispersal from unburned residual trees, which will become less under an intensified fire regime, whereas deciduous broadleaves have seeds with capabilities of long-distance dispersal.

Compared to ‘dark taiga’, in light taiga (*Pinus sylvestris* and *Larch spp.*) ecosystems, a full establishment of typical early-successional broadleaf forest after fire is less reported, but the widespread presence of broadleaf species in these ecosystems mentioned above indicate that there must be some form of broadleaf regeneration after fire. Light taiga are characterized by frequent surface fires with a fire return interval (FRI) of 30-100 years (Kharuk et al., 2008; Kharuk et al., 2011; Wirth, 2005) interspersed by infrequent stand-replacing fires with an FRI of 200-500 years (Schulze et al., 2012; Wirth, 2005). Typically, after surface fires, the regeneration was dominated by Scots pine and larch so that their monotypic stands are common (Wirth et al., 1999, for Scots pine; Zyryanova et al., 2007, for larch). But it is possible that dark taiga and broadleaf species can also dominate after fire in light taiga. For example, Abaimov et al. (2002) showed several possible postfire succession trajectories after fire in larch forests, including (1) ‘Uneven-age larch forest’ (self-replacement), (2) ‘Secondary birch forests’ → ‘Mixed forests with spruce prevailing’ → ‘Mixed spruce forests with larch and birch’ (or ‘Mixed larch forests with spruce and birch’). They also pointed out that more fires under a warming climate will enhance the competitiveness of birch (because of its re-sprouting capability) and spruce (mainly in northern larch system). Likewise, Zyryanova et al. (2007) pointed out in southeastern Siberian larch forests frequent surface fires maintain birch species because they can re-sprout, whereas larch saplings are killed.

For Scots pine-dominated light taiga, Furyaev et al. (2001) showed that after high-intensity fires which consume all of the forest floor and exposed mineral soil, the typical succession sequence is ‘fresh burns’ → ‘hardwood-pine and pine-hardwood young growth’ → ‘pole-size pine forests with birch and aspen admixture’ → ‘middle age and pre-mature pine forests with one or two age cohorts’ → ‘mature pine forests with two or more age cohorts’ → ‘over-mature pine forests thinned by fires with two or more age cohorts’, although the initial three stages can be stopped by fires which re-initiate the cycle.

The above evidence, mainly based on local field investigation or small-scale satellite imaging, suggests that broadleaf replacement of conifers after fire usually occur in dark taiga forests dominated by crown fires, which is similar to postfire succession in boreal North America where crown fires also dominate. For Siberian light taiga dominated by surface fires, postfire regeneration of broadleaf species do occur but a complete replacement of conifers by broadleaf species seems rare, and only after high-severity stand-replacing fires a complete replacement seems possible.

Over boreal North America, both extensive field investigations (Mack et al., 2021) and satellite observations (Beck et al., 2011; Massey et al., 2023), show that postfire broadleaf replacement of conifers occurs widely and over a large spatial scale, partly due to an intensified fire regime. However, over Siberia, satellite-based postfire changes in species composition over a large spatial scale are rarely reported (or even absent according to our literature survey). We suspect that this is probably because of two reasons: (1) for dark taiga such a species transition after fire is more probable, but fire occurrence is generally low in this system because dark taiga typically maintains a high surface fuel moisture and can only have crown fires at extreme drought. The low probability of fire occurrence in this system (also shown in our main text Fig. 1a) means that there are probably not enough fires available supporting large-scale satellite-based analysis. (2) over light taiga, surface fires prevail and enable broadleaf regeneration after fire but not a full broadleaf replacement of conifers, and therefore the signal of species change at the stand level might not be pronounced enough to be detected by currently mainstream medium-satellite land cover products (e.g., 30m Landsat-based data or 300m ESA CCI land cover product or 500m MODIS land cover product). However, with climate warming and the anticipated increase of drought events, fires in dark taiga and high-severity stand-replacing fires in light taiga might increase in the future, which will likely increase the chance of postfire species shift in Siberia (Tautenhahn et al., 2016).

3. Possibility of increasing broadleaf share to mitigate fire occurrence or postfire surface warming in Siberian boreal forests

The natural process of postfire early successional shift to a dominance of broadleaf species, or their regeneration without reaching dominance, makes it possible for a ‘helped transition’ to broadleaf species in Siberia. The question remains whether increasing the share of broadleaf forest would mitigate fire occurrence or postfire surface warming in this region.

The direct evidence of broadleaf trees (mainly birch and aspen, not the pyrophyte oak trees in Mediterranean Europe or Eucalyptus trees in Australia) in reducing fire risk in northern boreal and temperate regions mainly comes from North America in the current literature and is relatively rare in Siberia and boreal Europe. In the absence of direct evidence for Siberia we turn towards relevant insights acquired from other regions.

Cumming (2001) and Bernier et al. (2016), through comparing the distribution of different forest types in all lands and in burned areas only, concluded that broadleaf forests and mixed forests have a considerably lower chance of having fires than conifers in Canada. Debyle et al. (1987) shows that over 1970-1982, a period over which anthropogenic climate warming has not yet exerted clearly noticeable influences on wildfires, the burn rates of aspen (*Populus tremuloide*) stands were extremely low, resulting a fire return interval of 12,000 years in the interior western United States.

A recent systematic review (Nesbit et al., 2023), using both literature review and questionnaire of field fire managers, investigated the influence of quaking aspen (*Populus tremuloides*) on fire occurrence, behaviour and severity in North America. They found general evidence that the presence of aspen reduces fire occurrence, behaviour and severity because of several characteristics of aspen: (1) aspen-dominated stands have greater foliage moisture, at least during the spring and throughout the summer, than conifer stands (Wagner, 1977). (2) aspen stands have lower canopy bulk density and higher canopy base, and fewer small-diameter canopy fuel, making crown fires less likely. (3) aspen litter decomposes faster than needle litter, resulting in less surface fuel. There was evidence that aspen-dominated stands had higher soil moisture content than conifer-dominated stands in western Canada and in the interior western US. These features make surface fire less likely and less severe in aspen stands, in particular during summer.

These modern-time observations support a low flammability of broadleaf stands in North America. Reconstructions of fire and vegetation changes over the Holocene in Alaskan boreal forests, based on charcoal and pollen records from peat and lacustrine sediments, show that while increased fire activity since the mid-Holocene was found being associated with an expansion of black spruce and a reduction of birch, over a centennial time scale, increasing fire activities have resulted in increased aspen area which further dampened fire occurrence in the following period (Kelly et al., 2013). Similar evidence was also reported in eastern Canadian boreal forests but with a reverse direction: since 3000 years before present (BP), regional climate data show a decreasing trend in drought severity but fire frequency failed to show a corresponding decreasing trend, because increasing conifer forest area, at the cost of decreasing broadleaf areas, has compensated for the climate-driven decrease in fire activity (Girardin et al., 2013).

Based on these observations, several authors proposed the possibility of using broadleaf species as a 'fuel treatment' method to mitigate fire occurrence in North America. However, the direct evidence that broadleaf species can also mitigate fires in Siberia or boreal Europe remains very limited if not absent.

Although some studies on paleo fire and vegetation dynamics in Siberia also showed increasing abundance of broadleaf species corresponding to elevated fire activities during some time in history (Barhoumi et al., 2020; Feurdean et al., 2020), in general there is no evidence that increasing broadleaf forests can dampen climate-

moderated fire variability as being found in boreal North America (despite that inherent uncertainties exist in the charcoal and pollen records and their interpretations). There is one study (Feurdean et al., 2017), however, showing that in hemiboreal/boreal region in northeastern Europe, the expansion of temperate deciduous broadleaf species (*Ulmus laevis*, *U. glabra*, *Quercus robur*, *Tilia cordata*, *Corylus avellana*, *Fagus sylvatica*, *Carpinus betulus*) during the warm and dry Holocene Thermal Optimum (7500–4500 years BP) resulted in low fire activity that is contrary to what should have been driven by the climate conditions. This demonstrates the dampening effect on fire by increasing broadleaf forests.

Given that the fire mitigation effect of aspen (and other broadleaves including birch) in North American crown fire systems largely depend on the biological and ecological features of these species, and considering that the major aspen species in boreal Eurasia, the Eurasian aspen (*Populus tremula*), has a similar autecological and phenological characteristics to quaking aspen (*Populus tremuloides*), it may be reasonable to assume that increasing broadleaf species in Eurasian dark taiga forests (dominated by crown fires) can have a fire mitigation effect similar to North America.

However, whether increasing broadleaf species can mitigate surface fires remains unclear. In fact, from our results it seems that increasing broadleaf species in southern Siberia and southeastern Siberia are promoting low-severity surface fires (see black-circled areas in Fig. R3 below which is the same as main text Fig. 1) although these fires are most likely caused by humans. These low-severity fires result in a negligible reduction in postfire LAI or even slight increase in postfire LAI with also negligible postfire forest mortality (Extended Data Fig. 7), very likely resulting from the re-sprouting capability of broadleaf trees and their low vulnerability to fires. As a result, postfire surface warming in summer one year after fire is minimal. A re-production of Extended Data Fig. 9, but using data for Eurasia only (Fig. R4 below), clearly shows that with increasing abundance of broadleaf species, fire size, postfire decrease in LAI, forest mortality and postfire surface warming all show decreasing trend, confirming that increasing broadleaf, although may not mitigate fire occurrence, reduces postfire surface warming.

In the revised manuscript, (1) we have clearly describe that the increasing abundance of broadleaf species in response to an intensified fire regime is limited to North America (after adding new reference showing that this also happens over Canada and is hence not limited to Alaska); (2) the management suggestion regarding increasing broadleaf species has been moderated to indicate its plausibility in dark taiga in boreal Eurasia, but more search is needed regarding its feasibility in light taiga. These revisions are lines 211–214 of the revised manuscript (also shown above).

Figure R3 | The same figure as main text Figure 1.

Figure R4 | A reproduction of Extended Data Fig. 9 but using data of Eurasia only. Fire vulnerability is defined as the postfire change in LAI (ΔLAI , a), surface radiometric temperature (ΔT , e), and forest mortality (h). Simple linear regression models ($y = \alpha + \beta \times \log_{10}(\text{fire size})$) were fitted for each 2° grid cell with more than 10 fires to derive the fire size effect (β) on postfire changes in LAI ($\beta_{\Delta\text{LAI}}$, b), ecosystem evapotranspiration ($\beta_{\Delta\text{ET}}$, c), surface albedo ($\beta_{\Delta\alpha}$, d) and surface radiometric temperature ($\beta_{\Delta\text{T}}$, f) in summer one year after fire. Boxplots are shown for fire size (g), fire vulnerability (a, e, h) and fire size effect on biogeophysical changes (b, c, d, f) using samples of 2° grid cells falling in different bins of DBF and MF forest coverage (in 10% intervals). The number above each box indicates the number of 2° grid cells in each bin. The center line of the boxplots represents the median value, with box limits indicating upper and lower quartiles and whiskers showing $1.5 \times$ interquartile range.

[Comment 4] L.46: should other gases (i.e. methane) and aerosols (i.e. black carbon) be counted towards this impact, too?

[Response] Here the surface climate impacts are strictly limited to “CO₂ emissions” as stated in the original manuscript. We agree with the referee that other gases and aerosols should also be taken into account, but we are not aware of any studies comparing surface climate impacts (such as surface temperature change) between biogeochemical (by including CO₂, other gases and aerosols) and biogeophysical effects by fire.

The surface climate impacts of aerosols are mentioned at the end of the manuscript (lines 238–240). Previous studies have shown that vegetation fires in general exert a global radiative cooling effect on Earth, and future fires will cool the Earth even more (Ward et al., 2012), because aerosol-induced cooling exceeds CO₂-induced warming. Likewise, Randerson et al. (2006) also show that shortening fire return intervals in boreal forests (which might be happening now) can lead to enhanced negative radiative forcing at the global scale (Note that the radiative cooling does not contradict local near-surface warming, as explained in our study). Despite this evidence, recent studies continue to focus on the CO₂ emissions caused by recent surges in boreal fires (Zheng et al., 2023). Hence, explicitly mentioning CO₂ effects seems acceptable in the Introduction. Nonetheless, whether recent surges in large fires have modified the overall cooling effect of forest fires depends on various factors, including the relative proportion of CO₂ versus aerosol emissions and postfire vegetation change, and the investigation of the issue is beyond our current scope.

[Comment 5] L. 57: to be specific these datasets are not strictly independent, as they are also derived from MODIS.

[Response] We agree with the referee and thanks for pointing this out. The sentence concerned has been modified as (lines 162–164): “*The independence of the algorithms used to derive fire size, fire intensity and fire severity datasets provides robust evidence for the co-varying fire behaviour effects which contribute to increasing postfire surface warming with fire size.*”.

[Comment 6] L. 190: I found the reference to fire regimes here a bit confusing as the authors have not specifically analysed fire regimes (there is also no reference to it in Fig. 3). My small suggestion is to take out the term fire regime, as the authors already explain well that forest type can influence the intensity of fires i.e. (as shown in prior studies).

[Response] Following the referee’s suggestion ‘fire regime’ has been removed. In fact, the paragraph in question has been re-written in a more concise form (lines 182–187).

[Comment 7] L 201: which of the datasets is based on two decades of data? This is not explained or stated in L751-753 in the methods

[Response] We are sorry for this mistake. The original ‘two decades’ should have been ‘four decades’, as the shortest period was given by the data of USA (1984–2020), which covers 37 years (about four decades) rather than two decades. Thank you for highlighting this issue. This sentence has been revised to (lines 190–192): “*In the forests of Canada, the USA and Australia, both the mean and extreme fire size have increased by 65%–450% over the past four to six decades, along with an increase in burned area (Extended Data Fig. 10).*”

[Comment 9] L. 206: As explained above, I would suggest to either exclude Australia or extend the whole analysis to it

[Response] Please refer to our responses to your [Comment 2] above. If the arguments given there do not convince you, we could still remove Australia from the main text and instead put it in the Supplementary Information as part of the section on the snow effect on winter cooling (or winter warming after fire when there is little snow, as shown for Australia).

[Comment 10] L214: climatic drivers or the legacy of long-term suppression/fire exclusion?

[Response] Fires in forest ecosystems in the USA and Canada are usually not limited by fuel availability, as evidenced by the fact that the temporal variabilities of forest burned area in these two countries are closely linked to fire weather and water deficit, rather than to cumulative precipitation prior to fire occurrence (Abatzoglou et al., 2018). Likewise, Abatzoglou and Williams (2016) used modeling to demonstrate that human-caused climate change (increases in temperature and vapor pressure deficit) caused over half of the documented increases in fuel aridity since the 1970s and doubled the cumulative forest fire area since 1984 in western US forested landscapes. For Canada, Gillett and Weaver (2004) show a close link between the national burned area anomaly and fire-season temperature over 1920–2000 suggesting a long-term control of temperature on fire activity. Wang et al. (2015) further show that the annual frequency of fire spread days with extreme fire weather, on which the majority of burned area occurs, has increased over 1981–2010 due to past climate change and will continue to increase by 35%–400% by 2050 in response to future climate change. Although, as pointed out by the referee, several studies show increasing fuel loads and fire potential across western US forests due to past fire suppression (Marlon et al., 2012; Parks et al., 2015), which may increase the sensitivity of area burned to climate variability, there seems to be no clear evidence that fuel limitation has substantially disrupted the relationship between western US forest fire area and climate-caused fuel aridity (Abatzoglou and Williams, 2016). Moreover, recent studies ascribe the outbreaks of extremely large boreal fires in recent years to record-high temperature and water deficit (drought), combined with changes in lightning activities and arctic-front jet occurrence, all being climate factors (Scholten et al., 2022; Zheng et al., 2023).

Following the referee's suggestion, the sentence in question has been revised as (lines 198–200): *“growing burned area and fire size there highlight the difficulty of preventing large fires in sparsely populated areas, which are mostly climate-driven and possibly exacerbated by historical fire exclusion.”*

[Comment 11] L. 248: permafrost thaws (melting is only used for ice)

[Response] Thanks for pointing this out. We have made the change in the revised manuscript (line 235).

[Comment 12] L 435f: Considerable limitations of MCD64 in boreal regions have been shown by (Xu et al., 2022), (Chen et al., 2021) and (Scholten et al., 2022). Not saying that GFA would not be suitable to use, but I suggest to moderate this statement.

[Response] We thank the referee for making this point and for providing the relevant references. For the revised manuscript, we searched the literature for a quantitative description regarding the omission and the performance of MCD64A1 in boreal regions. The relevant texts in the Methods are revised as (lines 422–426): *“A global, systematic evaluation of the MCD64A1 burned area data showed that over boreal forest, the product has an omission error of 27% being roughly compensated for by a commission error of 24%, but in temperate forest it omits a large proportion of fires⁵⁴. Additional studies show that over boreal Eurasia, and over Canada and US Alaska, MCD64A1 underestimated burned area by about 20%^{55,56}.”*

Following this revision, the potential influence on fire size mapping of the omission of burned area in MCD64A1, and the comparison with regional high-quality fire patch datasets is also mentioned in the revised Methods as (lines 433–438): *“Nonetheless, the omission of burned area by MCD64A1 in boreal regions likely leads to underestimation of fire size, which is confirmed by comparing GFA with regional fire patch datasets of higher quality than GFA for Canadian and Alaskan boreal forests (Supplementary Text 1). However, the major conclusion of this study, i.e., forest fire size amplifies postfire surface warming, was tested by using the high-quality fire patch datasets and found to be robust (Supplementary Text 1)”*

[Comment 13] Fig. 3: What do the different letters indicate for the significance level? Do they all imply “significant at $P < 0.05$ ”? In that case, why use different letters? Also, it would be good to report the average and spread of fire sizes in the different forest types – could the higher β values for needleleaf forest be related to the fact that they are the only regions that facilitate large fires?

[Response] The letters indicate the statistical significance in the differences among forest types for the variables considered in Fig. 3. As significant differences among forest types have been found for all variables, there is no longer the need for these letters. Hence, in the revised Fig. 3, all such letters are removed to avoid confusion. Instead, the statistical significance of the differences is explained in the revised figure caption.

Following the referee's suggestion, the information on fire size is provided in the revised Extended Data Fig. 8 and Extended Data Fig. 9. Indeed, as speculated by the referee, fire size is smaller for DBF and MF than for needleleaf forests (Extended Data Fig. 8); fire size also shows a decreasing trend with increasing coverage of DBF and MF (Extended Data Fig. 9). However, the contrasts in fire size among different forest types and among different coverages of DBF and MF forests are far less pronounced than those in fire vulnerability (ΔLAI and forest mortality). Therefore, the modulation of forest type on the size impact of postfire surface warming (different $\beta_{\Delta T}$ values) mainly depends on the differences in the size impact on ΔLAI and forest mortality among different forest types, and the cascading postfire biogeophysical processes, rather than depending directly on fire size. For the same reason, the spatial patterns of mean fire size and extreme fire size (Fig. R5) have only a weak correspondence with the spatial pattern of $\beta_{\Delta T}$ (main text Fig. 1c).

Figure R5 | Spatial distribution of mean fire size (a) and the extreme fire size (95th percentile) (b) within each 2° grid cell across the study domain of northern temperate and boreal forests (40°N–70°N) for 2003–2016.

References

- Abaimov, A.P., Zyryanova, O.A., Prokushkin, S.G., 2002. Long-term investigations of larch forests in cryolithic zone of Siberia: brief history, recent results and possible changes under global warming. *Eurasian Journal of Forest Research* 5, 95-106.
- Abatzoglou, J.T., Williams, A.P., 2016. Impact of anthropogenic climate change on wildfire across western US forests. *Proc Natl Acad Sci U S A* 113, 11770-11775.
- Abatzoglou, J.T., Williams, A.P., Boschetti, L., Zubkova, M., Kolden, C.A., 2018. Global patterns of interannual climate-fire relationships. *Glob Chang Biol* 24, 5164-5175.
- Alkama, R., Cescatti, A., 2016. Biophysical climate impacts of recent changes in global forest cover. *Science* 351, 600-604.
- Barhouni, C., Ali, A.A., Peyron, O., Dugerdil, L., Borisova, O., Golubeva, Y., Subetto, D., Kryshen, A., Drobyshev, I., Ryzhkova, N., Joannin, S., Williams, J., 2020. Did long-term fire control the coniferous boreal forest composition of the northern Ural region (Komi Republic, Russia)? *Journal of Biogeography* 47, 2426-2441.
- Barrett, K., McGuire, A.D., Hoy, E.E., Kasischke, E.S., 2011. Potential shifts in dominant forest cover in interior Alaska driven by variations in fire severity. *Ecol Appl* 21, 2380-2396.

- Beck, P.S.A., Goetz, S.J., Mack, M.C., Alexander, H.D., Jin, Y., Randerson, J.T., Loranty, M.M., 2011. The impacts and implications of an intensifying fire regime on Alaskan boreal forest composition and albedo. *Global Change Biology* 17, 2853-2866.
- Bernier, P., Gauthier, S., Jean, P.-O., Manka, F., Boulanger, Y., Beaudoin, A., Guindon, L., 2016. Mapping Local Effects of Forest Properties on Fire Risk across Canada. *Forests* 7.
- Bousfield, C.G., Lindenmayer, D.B., Edwards, D.P., 2023. Substantial and increasing global losses of timber-producing forest due to wildfires. *Nature Geoscience*.
- Cumming, S.G., 2001. Forest Type and Wildfire in the Alberta Boreal Mixedwood: What Do Fires Burn? *Ecological Applications* 11, 97-110.
- DeByle, N.V., Bevins, C.D., Fischer, W.C., 1987. Wildfire Occurrence in Aspen in the Interior Western United States. *Western Journal of Applied Forestry* 2, 73-76.
- Fechner, G.H., Barrows, J.S., 1976. Aspen stands as wildfire fuel breaks. *Eisenhower Consortium Bulletin* 4, 26 p. Rocky Mountain Forest and Range Experiment Station, Fort Collins, Colo. .
- Feurdean, A., Florescu, G., Tanțau, I., Vanni re, B., Diaconu, A.-C., Pfeiffer, M., Warren, D., Hutchinson, S.M., Gorina, N., Galka, M., Kirpotin, S., 2020. Recent fire regime in the southern boreal forests of western Siberia is unprecedented in the last five millennia. *Quaternary Science Reviews* 244.
- Feurdean, A., Veski, S., Florescu, G., Vanni re, B., Pfeiffer, M., O'Hara, R.B., Stivrins, N., Amon, L., Heinsalu, A., Vassiljev, J., Hickler, T., 2017. Broadleaf deciduous forest counterbalanced the direct effect of climate on Holocene fire regime in hemiboreal/boreal region (NE Europe). *Quaternary Science Reviews* 169, 378-390.
- Furyaev, V.V., Vaganov, E.A., Tchepakova, N.M., Valendik, E.N., 2001. Effects of Fire and Climate on Successions and Structural Changes in The Siberian Boreal Forest. *Eurasian Journal of Forest Research* 2, 1-15.
- Gauthier, S., Bernier, P., Kuuluvainen, T., Shvidenko, A.Z., Schepaschenko, D.G., 2015. Boreal forest health and global change. *Science* 349, 819-822.
- Gillett, N.P., Weaver, A.J., 2004. Detecting the effect of climate change on Canadian forest fires. *Geophysical Research Letters* 31.
- Girardin, M.P., Ali, A.A., Carcaillet, C., Blarquez, O., Hely, C., Terrier, A., Genries, A., Bergeron, Y., 2013. Vegetation limits the impact of a warm climate on boreal wildfires. *New Phytol* 199, 1001-1011.
- Girardin, M.P., Terrier, A., 2015. Mitigating risks of future wildfires by management of the forest composition: an analysis of the offsetting potential through boreal Canada. *Climatic Change* 130, 587-601.
- Hayes, D.J., McGuire, A.D., Kicklighter, D.W., Gurney, K.R., Burnside, T.J., Melillo, J.M., 2011. Is the northern high-latitude land-based CO₂sink weakening? *Global Biogeochemical Cycles* 25, GB3018.
- Hirsch, K., Kafka, V., Todd, B., 2004. Using forest management techniques to alter forest fuels and reduce wildfire size: an exploratory analysis. Pages 175–184 in R.T. Engstrom, K.E.M. Galley, and W.J. de Groot (eds.). *Proceedings of the 22nd Tall Timbers Fire Ecology Conference: Fire in Temperate, Boreal, and Montane Ecosystems*. Tall Timbers Research Station, Tallahassee, FL. .
- Johnstone, J.F., Chapin, F.S., Hollingsworth, T.N., Mack, M.C., Romanovsky, V., Turetsky, M., 2010. Fire, climate change, and forest resilience in interior Alaska. *Canadian Journal of Forest Research* 40, 1302-1312.
- Kelly, R., Chipman, M.L., Higuera, P.E., Stefanova, I., Brubaker, L.B., Hu, F.S., 2013. Recent burning of boreal forests exceeds fire regime limits of the past 10,000 years. *Proc Natl Acad Sci U S A* 110, 13055-13060.
- Kharuk, V.I., Dvinskaya, M.L., Petrov, I.A., Im, S.T., Ranson, K.J., 2016. Larch Forests of Middle Siberia: Long-Term Trends in Fire Return Intervals. *Reg Environ Change* 16, 2389-2397.

- Kharuk, V.I., Ponomarev, E.I., Ivanova, G.A., Dvinskaya, M.L., Coogan, S.C.P., Flannigan, M.D., 2021. Wildfires in the Siberian taiga. *Ambio* 50, 1953-1974.
- Kharuk, V.I., Ranson, K.J., Dvinskaya, M.L., 2008. Wildfires dynamic in the larch dominance zone. *Geophysical Research Letters* 35.
- Kharuk, V.I., Ranson, K.J., Dvinskaya, M.L., Im, S.T., 2011. Wildfires in northern Siberian larch dominated communities. *Environmental Research Letters* 6.
- Kharuk, V.I., Shvetsov, E.G., Buryak, L.V., Golyukov, A.S., Dvinskaya, M.L., Petrov, I.y.A., 2023. Wildfires in the Larch Range within Permafrost, Siberia. *Fire* 6.
- Krawchuk, M.A., Cumming, S.G., Flannigan, M.D., Wein, R.W., 2006. Biotic and abiotic regulation of lightning fire initiation in the mixedwood boreal forest. *Ecology* 87, 458-468.
- Mack, M.C., Walker, X.J., Johnstone, J.F., Alexander, H.D., Melvin, A.M., Jean, M., Miller, S.N., 2021. Carbon loss from boreal forest wildfires offset by increased dominance of deciduous trees. *Science* 372, 280-283.
- Marlon, J.R., Bartlein, P.J., Gavin, D.G., Long, C.J., Anderson, R.S., Briles, C.E., Brown, K.J., Colombaroli, D., Hallett, D.J., Power, M.J., Scharf, E.A., Walsh, M.K., 2012. Long-term perspective on wildfires in the western USA. *Proc Natl Acad Sci U S A* 109, E535-543.
- Massey, R., Rogers, B.M., Berner, L.T., Cooperdock, S., Mack, M.C., Walker, X.J., Goetz, S.J., 2023. Forest composition change and biophysical climate feedbacks across boreal North America. *Nat Clim Chang* 13, 1368-1375.
- McGuire, A.D., Anderson, L.G., Christensen, T.R., Dallimore, S., Guo, L., Hayes, D.J., Heimann, M., Lorensen, T.D., Macdonald, R.W., Roulet, N., 2009. Sensitivity of the carbon cycle in the Arctic to climate change. *Ecological Monographs* 79, 523-555.
- Mekonnen, Z.A., Riley, W.J., Randerson, J.T., Grant, R.F., Rogers, B.M., 2019. Expansion of high-latitude deciduous forests driven by interactions between climate warming and fire. *Nat Plants* 5, 952-958.
- Mollicone, D., Eva, H.D., Achard, F., 2006. Human role in Russian wild fires. *Nature* 440, 436-437.
- Nesbit, K.A., Yocom, L.L., Trudgeon, A.M., DeRose, R.J., Rogers, P.C., 2023. Tamm review: Quaking aspen's influence on fire occurrence, behavior, and severity. *Forest Ecology and Management* 531.
- Nikolov, N., Helmisaari, H., 1992. Silvics of the circumpolar boreal forest tree species. In: Shugart HH, Leemans R, Bonan GB, eds. *A Systems Analysis of the Global Boreal Forest*. Cambridge University Press; 13-84.
- Parks, S.A., Miller, C., Parisien, M.-A., Holsinger, L.M., Dobrowski, S.Z., Abatzoglou, J., 2015. Wildland fire deficit and surplus in the western United States, 1984–2012. *Ecosphere* 6.
- Piao, S., Nan, H., Huntingford, C., Ciais, P., Friedlingstein, P., Sitch, S., Peng, S., Ahlstrom, A., Canadell, J.G., Cong, N., Levis, S., Levy, P.E., Liu, L., Lomas, M.R., Mao, J., Myneni, R.B., Peylin, P., Poulter, B., Shi, X., Yin, G., Viovy, N., Wang, T., Wang, X., Zaehle, S., Zeng, N., Zeng, Z., Chen, A., 2014. Evidence for a weakening relationship between interannual temperature variability and northern vegetation activity. *Nat Commun* 5, 5018.
- Ran, Y., Jorgenson, M.T., Li, X., Jin, H., Wu, T., Li, R., Cheng, G., 2021. Biophysical permafrost map indicates ecosystem processes dominate permafrost stability in the Northern Hemisphere. *Environmental Research Letters* 16.
- Randerson, J.T., Liu, H., Flanner, M.G., Chambers, S.D., Jin, Y., Hess, P.G., Pfister, G., Mack, M.C., Treseder, K.K., Welp, L.R., Chapin, F.S., Harden, J.W., Goulden, M.L., Lyons, E., Neff, J.C., Schuur, E.A., Zender, C.S., 2006. The impact of boreal forest fire on climate warming. *Science* 314, 1130-1132.
- Rogers, B.M., Soja, A.J., Goulden, M.L., Randerson, J.T., 2015. Influence of tree species on continental differences in boreal fires and climate feedbacks. *Nature Geoscience* 8, 228-234.
- Schneider, R.R., 2013. *Alberta's Natural Subregions Under a Changing Climate: Past, Present, and Future*.

- Scholten, R.C., Coumou, D., Luo, F., Veraverbeke, S., 2022. Early snowmelt and polar jet dynamics co-influence recent extreme Siberian fire seasons. *Science* 378, 1005-1009.
- Schulze, E.D., Wirth, C., Mollicone, D., von Lüpke, N., Ziegler, W., Achard, F., Mund, M., Prokushkin, A., Scherbina, S., 2012. Factors promoting larch dominance in central Siberia: fire versus growth performance and implications for carbon dynamics at the boundary of evergreen and deciduous conifers. *Biogeosciences* 9, 1405-1421.
- Schulze, E.D., Wirth, C., Mollicone, D., Ziegler, W., 2005. Succession after stand replacing disturbances by fire, wind throw, and insects in the dark Taiga of Central Siberia. *Oecologia* 146, 77-88.
- Stralberg, D., Wang, X., Parisien, M.A., Robinne, F.N., Sólymos, P., Mahon, C.L., Nielsen, S.E., Bayne, E.M., 2018. Wildfire-mediated vegetation change in boreal forests of Alberta, Canada. *Ecosphere* 9.
- Tautenhahn, S. (2014). Multiple successional pathways after fire in Siberian forests. PhD Thesis, Technische Universität Bergakademie, Freiberg.
- Tautenhahn, S., Lichstein, J.W., Jung, M., Kattge, J., Bohlman, S.A., Heilmeyer, H., Prokushkin, A., Kahl, A., Wirth, C., 2016. Dispersal limitation drives successional pathways in Central Siberian forests under current and intensified fire regimes. *Glob Chang Biol* 22, 2178-2197.
- Uemura, S., Kanda, F., Isaev, A.P., Tsuji, T., 1997. Forest structure and succession in southeastern Siberia. *Vegetation Science* 14, 119-127.
- Wagner, C.E.V., 1977. Conditions for the start and spread of crown fire. *Canadian Journal of Forest Research* 7, 23-34.
- Wang, X., Thompson, D.K., Marshall, G.A., Tymstra, C., Carr, R., Flannigan, M.D., 2015. Increasing frequency of extreme fire weather in Canada with climate change. *Climatic Change* 130, 573-586.
- Ward, D.S., Kloster, S., Mahowald, N.M., Rogers, B.M., Randerson, J.T., Hess, P.G., 2012. The changing radiative forcing of fires: global model estimates for past, present and future. *Atmospheric Chemistry and Physics* 12, 10857-10886.
- Williams, N.G., Lucash, M.S., Ouellette, M.R., Brussel, T., Gustafson, E.J., Weiss, S.A., Sturtevant, B.R., Schepaschenko, D.G., Shvidenko, A.Z., 2023. Simulating dynamic fire regime and vegetation change in a warming Siberia. *Fire Ecology* 19.
- Wirth, C., 2005. Wirth, C. Fire Regime and Tree Diversity in Boreal Forests: Implications for the Carbon Cycle. In *Forest Diversity and Function: Temperate and Boreal Systems*; Springer: Berlin/Heidelberg, Germany, pp. 309–344 (2005).
- Wirth, C., Schulze, E.D., Schulze, W., von Stunzner-Karbe, D., Ziegler, W., Miljukova, I.M., Sogatchev, A., Varlagin, A.B., Panvyorov, M., Grigoriev, S., Kusnetzova, W., Siry, M., Harges, G., Zimmermann, R., Vygodskaya, N.N., 1999. Above-ground biomass and structure of pristine Siberian Scots pine forests as controlled by competition and fire. *Oecologia* 121, 66-80.
- Zhang, Y., Wang, X., Lian, X., Li, S., Li, Y., Chen, C., Piao, S., 2024. Asymmetric impacts of forest gain and loss on tropical land surface temperature. *Nature Geoscience* 17, 426-432.
- Zheng, B., Ciais, P., Chevallier, F., Yang, H., Canadell, J.G., Chen, Y., van der Velde, I.R., Aben, I., Chuvieco, E., Davis, S.J., Deeter, M., Hong, C., Kong, Y., Li, H., Li, H., Lin, X., He, K., Zhang, Q., 2023. Record-high CO₂ emissions from boreal fires in 2021. *Science* 379, 912-917.
- Zyryanova, O.A., Yaborov, V.T., Tchikhacheva, T.L., Koike, T., Kobayashi, M., Matsuura, Y., Satoh, F., Zyryanov, V.I., 2007. The Structure and Biodiversity after Fire Disturbance in *Larix gmelinii*(Rupr.). *Eurasian Journal of Forest Research* 10, 19-29.

Reviewer Reports on the Third Revision:

Referees' comments:

Referee #3 (Remarks to the Author):

The authors have integrated my suggestions in the revised version with great care and detail. I particularly enjoyed reading the discussion of where and when increasing broadleaf forest coverage in northern forests can be feasible and most beneficial. The authors have discussed this in great detail in their answer to both reviewers, and I would encourage them to share this interesting and well-referenced discussion with the public, for example by publishing their reviews and rebuttals upon acceptance.

I have a few minor comments remaining (below), after which I believe the paper will be ready for publication. I believe this research makes an interesting and valuable contribution to the field.

Main text:

L63 Barbero et al (ref 16) is about CONUS, I would suggest to replace this with a northern forest specific reference

L85 Could use a reference

L104 Sentence is not clear - increase following an increase in fire size?

L124: I wonder if the warming in year for in ED Fig. 3a is significant? Visually, it looks like a very minor effect. Maybe adding asterisks for the data points representing significance could be useful. Also, it would be interesting to see the cumulative effect over time – i.e. is there a point in time at 10 years or so, when the winter cooling outweighs the summer warming?

L205-217: Perhaps mentioning very briefly an example such as the idea of strips of forests for fuel break (mentioned in the answers to the reviewers) could clarify that the authors are not calling for large-scale replacements in tree species across the entire domain but for well-targeted measures?

Figures:

Fig1:

(1) In panel f, it could be useful to align the zero values on the y axes

(2) Panel e: These values are based on total land area, not fire perimeters, correct? In that case it would be good to add a sentence to the methods how these averages are computed, for example, is this based on resampled or original data, and what data was used for computing land area? (in the methods it is only explained how forest types were computed for each fire)

Fig3:

(1) Panels a and e are missing error bars.

(2) Could the authors add p-values/significance levels (i.e. asterisks) to the panels for the slope values for each forest type (or alternatively mention in the caption that all are significant at $P < 0.05$ if this is the case)? Since the slopes of DBF are close to zero, I wonder what the p-values for these regressions are.

(3) Aligning the zero values for panels b, c, d, and f, or drawing a light line in the panels at the zero value

could help with readability.

Extended Data Fig.3 is missing error bars. Also, it would be interesting to know which of these changes are significantly different from zero (especially the annual average)

Extended Data Fig.5: Very difficult to see the stippling in panel b (if there is any?)

Extended Data Fig.6g: Should probably be MW per area? (since the grid cell area varies when plotting in degrees)

Extended Data Fig. 8c: Also MW per area for FRP? ARE FRP and spread rate based on averages per fire or is the variation within fires taken into account?

Extended Data Fig.10: Is it possible to add confidence intervals to the regression lines?

Methods:

L437: Suggest to specifically name the regional fire patch datasets that were used. Could also mention that the regional datasets are based on Landsat and MCD14 to clarify that these are not subject to MCD64 omission/commission errors

L452: Was the FRP normalized by pixel area? Given the large variability in pixel size with MCD14 this could affect the computation of average FRP.

L475: Is the 30m tree cover product for 2000 also from GFC?

L484/485: Would this not be a 3-year time window? The year of burning + the year after + two years after?

L492/ L498/ L.509/ L.535: How is valid data defined? Is there a quality flag or confidence rating provided that was used?

Section 2.1: Was temporal averaging applied? Monthly? Seasonal? Annual? How were “summer” and “winter” defined (i.e. for ED Fig2 and others, and in the main text)

Section 2.3:

(1) Same as for section 2.1 - what is the temporal time frame of postfire biogeophysical responses? Summer only, or also annual and winter? And for how many years after the fire?

(2) L636-639: When applying multiple linear regressions to grid cells with spatial correlations, it would be recommended to correct for the false discovery rate (see i.e. “field significance” issues discussed in Wilks 2016: <https://journals.ametsoc.org/view/journals/bams/97/12/bams-d-15-00267.1.xml>)

Supplementary text:

L30: Could mention that MTBS is also derived from Landsat

Author Rebuttals to Third Revision:

Referees' comments:

Referee #3 (Remarks to the Author):

[Comment 1] The authors have integrated my suggestions in the revised version with great care and detail. I particularly enjoyed reading the discussion of where and when increasing broadleaf forest coverage in northern forests can be feasible and most beneficial. The authors have discussed this in great detail in their answer to both reviewers, and I would encourage them to share this interesting and well-referenced discussion with the public, for example by publishing their reviews and rebuttals upon acceptance.

I have a few minor comments remaining (below), after which I believe the paper will be ready for publication. I believe this research makes an interesting and valuable contribution to the field.

[Response] We once again thank the referee for the constructive comments that help improve our manuscript quality. We are particularly pleased knowing that our discussions on the feasibility and benefits of increasing broadleaf forest coverage in northern forests are helpful. Following the referee's suggestion, we have chosen the transparent peer review so that all review materials will be openly available upon acceptance of the paper for publication. Please find below our detailed responses to your remaining minor comments, which helped further improve our manuscript.

Main text:

[Comment 2] L63 Barbero et al (ref 16) is about CONUS, I would suggest to replace this with a northern forest specific reference.

[Response] Following the referee's suggestion, we used the reference of Williams et al. (2023, Simulating dynamic fire regime and vegetation change in a warming Siberia. *Fire Ecol.* **19**, 33.) which is about northern forests. This study shows that annual burned area, mean fire size and fire intensity in Siberian forests are all projected to increase under future climate warming.

[Comment 3] L85 Could use a reference.

[Response] We have included the reference of Potter et al. (2023, Burned area and carbon emissions across northwestern boreal North America from 2001–2019.

Biogeosciences **20**, 2785-2804.), which has been cited in Methods for the evaluation of the MCD64A1 burned area data. Potter et al. (2023) compared burned area over Canada and US Alaska derived by MCD64A1 and by a regional dataset of ABoVE-FED and found that MCD64A1 underestimated burned area by about 20%.

[Comment 4] L104 Sentence is not clear - increase following an increase in fire size?

[Response] Thank you for pointing this out. We have revised the sentence as (lines 104–105): *“In line with surface warming, outgoing longwave radiation increased after fire, with greater increases following larger fires (Extended Data Figs. 1c, 1d).”*

[Comment 5] L124: I wonder if the warming in year for in ED Fig. 3a is significant? Visually, it looks like a very minor effect. Maybe adding asterisks for the data points representing significance could be useful. Also, it would be interesting to see the cumulative effect over time – i.e. is there a point in time at 10 years or so, when the winter cooling outweighs the summer warming?

[Response] Using one-sample two-tailed t-test, we found that the mean values of ΔT by averaging all fire patches, for any year after fire and irrespective of summer, winter, or annual mean, are all significantly different from zero ($p < 0.05$). In fact, all variables shown in Extended Data Fig. 3, except for summer $\Delta(H+G)$ for the 7th and 8th year after fire, have mean values significantly different from zero according to one-sample two-tailed t-test. This is because although the magnitudes of these variables are visually not so big, the sample size is often more than 50000, resulting a very small standard error of the mean and correspondingly a large value of t-statistics and a small p-value. In the revised Extended Data Fig. 3, we used solid dots to represent mean values significantly different from zero ($p < 0.05$) and empty dots for those not significantly different from zero ($p > 0.05$). The following sentence has been inserted in the caption of Extended Data Fig. 3: *“Solid dots represent mean values significantly different from zero ($p < 0.05$, two-tailed t-test), whereas empty dots represent those not significantly different from zero ($p > 0.05$, two-tailed t-test). The values of s.e.m. for all variables are too small to be visible and hence error bars showing s.e.m. are omitted.”*

Regarding the referee’s second question, it does not make sense to look at the

cumulative effect over time. Nonetheless, starting from the 5th year after fire, the winter cooling outweighs the summer warming, resulting in an annual effect of surface cooling. This has been now been clarified in the revised manuscript (lines 124–126) as: “*On the annual timescale, for the first four years after fire, the change in surface radiometric temperature was dominated by the summer signal showing size-dependent warming. Afterwards, the winter cooling became dominant*”.

[Comment 6] L205-217: Perhaps mentioning very briefly an example such as the idea of strips of forests for fuel break (mentioned in the answers to the reviewers) could clarify that the authors are not calling for large-scale replacements in tree species across the entire domain but for well-targeted measures?

[Response] Thanks for your suggestion. We have added the following sentence in lines 209–211 of the revised manuscript: “*For example, strips of coniferous forests could be, following harvest, replaced by broadleaf trees, which could then serve as firebreaks to reduce fire spread in an otherwise homogeneous coniferous landscape¹⁰.*” with the reference of “Nesbit, K. A., Yocom, L. L., Trudgeon, A. M., DeRose, R. J. & Rogers, P. C. Tamm review: Quaking aspen’s influence on fire occurrence, behavior, and severity. *For. Ecol. Manage.* **531**, 120752 (2023).”

Figures:

[Comment 7] Fig1:

- (1) In panel f, it could be useful to align the zero values on the y axes
- (2) Panel e: These values are based on total land area, not fire perimeters, correct? In that case it would be good to add a sentence to the methods how these averages are computed, for example, is this based on resampled or original data, and what data was used for computing land area? (in the methods it is only explained how forest types were computed for each fire).

[Response] We are grateful to the referee for making these points.

- (1) For the panel f, we have aligned the zero values on both sides of the y-axes and inserted a horizontal line indicating the zero value.
- (2) We are sorry for the unclarity in Panel e. In the panel the triangle legend shows the relative proportion of each forest type to the forest area over each 2° grid cell. The

colour opacity shows the proportion of forest area to land area. This has now been clarified in the revised caption of Fig. 1e. The relevant calculation processes have been described in the revised manuscript in lines 484–487: “*Based on the resampled 500m ESA CCI land cover data, annual proportions of forest area to total land area (including all land cover types except for water bodies) were calculated for each 2° grid cell for 2003–2016, followed by the calculation of multiannual mean proportion (shown as color opacity in Fig. 1e).*”, and lines 489–492: “*The relative proportions of ENF, DNF and MF+DBF against forest area over each 2° grid cell were calculated to show the spatial pattern of forest type composition across the study domain (Fig. 1e).*”

[Comment 8] Fig3:

- (1) Panels a and e are missing error bars.
- (2) Could the authors add p-values/significance levels (i.e. asterisks) to the panels for the slope values for each forest type (or alternatively mention in the caption that all are significant at $P < 0.05$ if this is the case)? Since the slopes of DBF are close to zero, I wonder what the p-values for these regressions are.
- (3) Aligning the zero values for panels b, c, d, and f, or drawing a light line in the panels at the zero value could help with readability.

[Response] (1) For panels a & e, one-tailed t-test shows that for all forest types, the mean value of ΔLAI (or ΔT) is significantly lower (or greater) than zero at $\alpha=0.05$, with negligible standard error of the mean (s.e.m.). The extremely small values of s.e.m. is because of the large sample size (>10000) (see also our response to Comment 5 above). When the values of s.e.m. are shown as error bars, the error bars are negligible (shown below as Fig. R1). Hence, we have explicitly stated the results of t-test in the revised caption of Fig. 3 as (lines 399–400): “*For all forest types, the mean value of ΔLAI (or ΔT) is significantly lower (or greater) than zero at $\alpha=0.05$ (one-tailed t-test), with negligible s.e.m.*”.

Fig. R1 The same as Fig. 3 but with panels a & e showing values of the standard error of the mean (s.e.m.) as error bars. The error bars are almost invisible because the s.e.m. values are very small.

Alternatively, we could show standard deviations of 2° grid cell values of a given forest type (below as Fig. R2, we first derived a spatial map similar to Fig. 1b for each forest type and then calculated the standard deviations of 2° grid cell values). But in this case, the error bars cross zero-value lines for MF, although one-sample t-test shows that both ΔLAI and ΔT for MF have mean values significantly different from zero. In this case the visual presentation is at odds with the result of t-test. For this reason, we think showing s.e.m. or explaining that s.e.m. values are negligible (which is our current choice) is consistent with the result of t-test.

Fig. R2 The same as Fig. 3 but with the panels a & e showing the standard deviation of all fire patches for a given forest type.

The relatively large standard deviations of summer ΔT in Fig. R2e above do not come as a surprise. Most previous studies investigating land surface temperature change following fire or forest cover change focus on showing the spatial maps of mean ΔT (e.g., Fig. 2 in Liu et al., 2018 Nature Communications; Fig. 1b in Rogers et al., 2013 Nature geoscience; Fig. 1 in Li et al., 2015 Nature Communications), similar to our Fig. 1b. The large standard deviations in ΔT following deforestation or forest gain in tropical regions were also reported in a recent study of Zhang et al. (2024, Nature geoscience). Their Fig. 1a & 1b [REDACTED] shows large standard deviations in ΔT and hence they used a binned regression approach to obtain the sensitivity of ΔT to changes in forest cover. For the same reason, the similar binned regression approach was also applied in Alkama and Cescatti (2016, Science). Despite the relatively large spread, a large sample size helped to identify a highly robust summer surface warming effect after fire in our study. Moreover, as shown in our Supplementary Fig. 10, with the increase of fire size, ΔT shows a clear departure from the zero value, showing an unequivocal warming effect when fires become large.

[REDACTED]

Based on the same consideration, we have revised Fig. 1f to remove the original error bars showing standard deviations but instead clearly stated in the revised Fig caption as (lines 379–380): “*The regional mean postfire summer ΔT (in violet, significantly greater than zero for all years at $\alpha=0.05$ by one-tailed t-test, with negligible s.e.m)*”. This is because summer ΔT for all years after fire are significantly greater than zero according to one-tailed t-test, but error bars showing standard deviations largely cross the zero-value line, which is visually at odds with the results of t-test.

Extended Data Fig. 3 also has a similar issue. We have provided the results of t-test showing the significance of the mean values but omitted the very small values of s.e.m.. Please refer to our response to Comment 5 and Comment 9 of the referee.

If the referee or the editorial team do not agree with our current choice of presenting errors (i.e. s.e.m. values are negligible), we could either show s.e.m. values as error bars (which is suboptimal because they will be almost invisible) or show standard deviations as error bars.

(2) Following the referee’s suggestion, in the revised Figs 3b, 3c, 3d, and 3f, we have indicated the significance of the regression slopes ($p<0.05$) for different forest types using asterisks. For all biophysical variables, except for DBF, the slopes of all forest types are significantly different from zero ($p<0.05$).

(3) We have inserted a dashed zero-line in the revised Figs 3b, 3c, 3d, and 3f.

[Comment 9] Extended Data Fig.3 is missing error bars. Also, it would be interesting to know which of these changes are significantly different from zero (especially the annual average).

[Response] Please refer to our response to Comment 5 regarding the significance of these mean values. We think showing s.e.m. would be consistent with the results of t-test but as the values of s.e.m. are very small (negligible on the figure), they are omitted.

Otherwise, showing the standard deviations of the values of 2° grid cells (similar to Fig. 1b) as error bars would look like Fig. R4 below. However, most error bars cross the zero-value line, which is visually at odds with the results of t-test showing that almost all mean values are significantly different from zero ($p < 0.05$).

Fig. R4 The same as Extended Data Fig. 3 but with error bars showing standard deviations of all fire patches. Solid dots represent mean values significantly different from zero ($p < 0.05$, two-tailed t-test), whereas empty dots represent those not

significantly different from zero ($p>0.05$, two-tailed t-test).

If the referee or the editorial team deem it as more appropriate to show the standard deviations or show s.e.m., we could do that.

[Comment 10] Extended Data Fig.5: Very difficult to see the stippling in panel b (if there is any?).

[Response] We thank the referee for pointing this out. There are in total 149 grid cells showing locally significant regressions ($p<0.05$). After the false discovery rate control ($\alpha_{FDR}=0.1$) this number becomes 78. The statistically significant cells are largely found in DNF-dominated Siberia but also partly found in North America. The stippling grid cells are not so obvious in the original figure likely because they are masked by the dark color of underlying map. We have revised the background map to a less dark color in order to allow the stippling grid cells being visible. The revised figure is reproduced below:

[Comment 11] Extended Data Fig.6g: Should probably be MW per area? (since the grid cell area varies when plotting in degrees).

[Response] We agree with the referee that the grid cell area varies when plotting in degrees. Following the referee's suggestion, we calculated the actual area of each fire pixel and based on this, calculated the FRP per area (MW km^{-2}) for each fire patch. The calculation process has been described in the revised manuscript (lines 472–478) as: "Although an active fire pixel has a nominal area of 1 km^2 , its actual area increases as the satellite sensor view moves away from the nadir. Hence, the view angle information was used to calculate the actual area of a given active fire pixel,

over which the given FRP was measured. More specifically, the area of an active fire pixel was calculated as the product of the along-scan (image scanning direction) pixel length and the along-track (satellite movement direction) pixel length, following Li et al.⁵⁷. The mean FRP per area ($MW\ km^{-2}$) of a fire event was finally obtained by dividing the total FRP of all active fire pixels within a fire patch by their total area.”

Based on the new method, we updated all relevant figures and tables: Extended Data Figs. 6 g-i, Extended Data Fig. 8, Supplementary Figure 19, and Supplementary Tables 5 and 6. Although the absolute values of FRP changed, the findings in these figures and tables remain unaffected.

[Comment 12] Extended Data Fig. 8c: Also MW per area for FRP? ARE FRP and spread rate based on averages per fire or is the variation within fires taken into account?

[Response] In the revised Extended Data Fig. 8c, the calculation of FRP accounts for changes in the area of active fire pixels as documented in our response to Comment 11 above. Regarding the referee’s second question: yes, both FRP and spread rate for a given fire patch are based on the average value per fire. The calculation of FRP on the fire patch level has been described in the response to Comment 11 above. For spread rate, the calculation of fire spread rate for a given fire patch has been described in lines 440–442 of the revised Methods: *“The spread rate for a given fire patch was calculated as the mean value of the spread rates for all underlying 500m burned pixels of a given patch”*.

The variations in FRP or spread rate within a single fire patch were taken into account when deriving their values on the fire patch level, but were not directly taken into account in Extended Data Fig. 8, because Extended Data Fig. 8 was based on fire patches rather than underlying 500m MODIS burned pixels. The latter point has been now made clear in the revised caption of Extended Data Fig. 8.

[Comment 13] Extended Data Fig.10: Is it possible to add confidence intervals to the regression lines?

[Response] Our key purpose here is to robustly estimate the temporal trends in burned area, extreme fire size and mean fire size, rather than predicting their values for a given year, which is usually the purpose of fitting regression models with confidence intervals provided for the dependent variable. The Theil-Sen estimation is a non-parametric method that can provide robust estimate for temporal trends when there are outliers in observations. It does not require that the residuals are normally distributed compared to ordinary least-squares regression. These advantages make Theil-Sen estimation idea for estimating temporal trends in burned area and fire size.

Following the derivation of the Sen's slope, an estimate for the intercept could be derived and hence a regression line could be drawn, but the purpose is to visually show the trend rather than to show how well the independent variable is predicted by the fitted model. Hence, we argue that confidence intervals around the fitted line is not needed in this case. Nonetheless, we agree with the referee that confidence intervals for the derived slope are needed. Hence, the 95th confidence intervals for the derived slopes are now provided in brackets in the revised Extended Data Fig. 10 directly. Please refer to the revised Extended Data Fig. 10.

Methods:

[Comment 14] L437: Suggest to specifically name the regional fire patch datasets that were used. Could also mention that the regional datasets are based on Landsat and MCD14 to clarify that these are not subject to MCD64 omission/commission errors.

[Response] Following the referee's suggestion, we have added the following sentences in the revised manuscript (lines 446–453): *“The first two regional datasets cover both Canadian and Alaskan forests and were generated by the Arctic-Boreal Vulnerability Experiment (ABoVE) project through combining fire perimeter information from the Alaskan Interagency Coordination Center (AICC) and Natural Resources Canada with either burn severity information derived from Landsat images (ABoVE-Landsat), or with burn date information from the MODIS MCD14ML active fire product (ABoVE-MODIS). The third one covers Alaska only and was obtained from the Monitoring*

Trends in Burn Severity (MTBS), which is based on Landsat images. These three datasets are not subject to the omission/commission errors associated with MCD64A1.”

[Comment 15] L452: Was the FRP normalized by pixel area? Given the large variability in pixel size with MCD14 this could affect the computation of average FRP.

[Response] Following the referee’s suggestion, the calculation of FRP for a given fire patch accounts for changes in the area of active fire pixels with latitude. Please see our response to Comment 11 above.

[Comment 16] L475: Is the 30m tree cover product for 2000 also from GFC?

[Response] Yes. This has now been noted in line 503 of the revised manuscript.

[Comment 17] L484/485: Would this not be a 3-year time window? The year of burning + the year after + two years after?

[Response] We initially counted the number of years by excluding the year of fire occurrence because it was included in any case. The referee’s comment made us realize that this naming is confusing. Hence, we adopted the naming suggested by the referee. The sentences in question have now been revised as (lines 511–515): *“fire-induced forest mortality was defined as including forest deaths within the year of fire and one year after fire (a 2-year time window). Although a 3-year time window, i.e., including forest deaths up to 2 years after fire and within the year of fire, was used in a previous study⁸, we found using a 3-year or 2-year time windows yielded negligible differences in the derived forest mortality”*.

[Comment 18] L492/ L498/ L.509/ 1.535: How is valid data defined? Is there a quality flag or confidence rating provided that was used?

[Response] For land surface temperature, we used the available observations with a reported error less than 2K, with no gap filling for missing data. For LAI, Albedo, and NDSI, by ‘valid’ we simply mean that the observations are available. In all three cases, we used all available observations with no gap filling for missing data. These points have been clarified in the revised manuscript (lines 521–522, 528–530, 539–540 and 565–566).

[Comment 19] Section 2.1 Was temporal averaging applied? Monthly? Seasonal?

Annual? How were “summer” and “winter” defined (i.e. for ED Fig2 and others, and in the main text).

[Response] We have added a paragraph right below the title of Section 2 to address these issues in the revised Methods (lines 568–574): *“This study quantified the impacts of forest fire on biogeophysical variables starting from one year to up to 14 years after fire, for both summer (June–August) and winter (December–February), as well as the annual time scale. The mean values for summer, winter and the annual of these variables were first calculated by averaging the observations with the respective time steps of different variables, over the corresponding time lengths of summer, winter and all year round (the annual), followed by the application of the space-and-time approach to derive the fire impacts (described below).”*

In addition, the definition of summer and winter have been added in the main text when they first appeared, and in all figures and tables.

[Comment 20] Section 2.3: (1) Same as for section 2.1 - what is the temporal time frame of postfire biogeophysical responses? Summer only, or also annual and winter? And for how many years after the fire?

[Response] This has been addressed in our response to Comment 19 above.

[Comment 21] (2) L636-639: When applying multiple linear regressions to grid cells with spatial correlations, it would be recommended to correct for the false discovery rate (see i.e. “field significance” issues discussed in Wilks 2016:

<https://journals.ametsoc.org/view/journals/bams/97/12/bams-d-15-00267.1.xml>).

[Response] We thank the referee for the suggestion. In the revised manuscript, for all main text figures (Fig. 1c, Fig. 2), Extended Data Figures (Extended Data Figs. 1, 5 and 6) and Supplement Figures (Supplementary Figs. 2, 3, 7, 8, 11, 13, 16, 17, 19, 20, 21, 23, 24 and 28), where spatial maps of significance test are presented, we have provided the additional information of field significance test corrected for the false discovery rate (FDR) using $\alpha_{\text{FDR}} = 0.1$. In all these revised figures, pixels with locally significant regressions ($p < 0.05$) have been stippled with both solid and empty dots, but pixels stippled with solid dots indicate those having additionally passed the field

significance test corrected for FDR ($\alpha_{\text{FDR}} = 0.1$). As is shown in these revised figures, in most cases, pixels passing field significance test corrected for FDR account for >80% of all pixels showing locally significant regressions, suggesting that using a more rigorous test corrected for FDR does not alter the robustness of our findings.

According to Wilks (2016), for observations with a moderate spatial auto-correlation, the global significance level (α_{global}) is about a half of α_{FDR} . Hence, using $\alpha_{\text{FDR}} = 0.1$ is approximately equivalent of assuming $\alpha_{\text{global}} = 0.05$. This assumption has also been used in previous studies investigating spatial maps of significance test (Jain et al., 2021; Zou et al., 2021). In addition, we went through recent papers using field significance test corrected for FDR, and we found that although $\alpha_{\text{FDR}} = 0.05$ has been used by a few studies (Portmann et al., 2022; Scholten et al., 2022), using $\alpha_{\text{FDR}} = 0.1$ is also very common (Burrell et al., 2020; Chiang et al., 2021; Gudmundsson et al., 2017; Jain et al., 2021; Zou et al., 2021; Zscheischler and Seneviratne, 2017). Hence, we conclude that using $\alpha_{\text{FDR}} = 0.1$ is appropriate and suitable for our study.

We did not completely replace local significance test with field significance test corrected for FDR because of two reasons: (1) To control for the false discovery probability whether a particular 2° grid cell shows significant regression or not is not our key concern. We argue that FDR control is probably more useful when trying to infer source areas that have a remote influence somewhere else. For example, detecting teleconnection is a common question in climate research. In this case, the key purpose is to precisely locate the areas (pixels) having a teleconnection impact. Hence, we would like to have a high confidence in pixels that do show a significant effect, justifying a control for FDR. In our study, the amplification effect of fire size, in particular for ENF, DNF and MF, is well shown in Fig. 3. The spatial maps are used to demonstrate spatial patterns rather than to precisely locate the pixels with significant fire size effect. Hence, a control for FDR is less strictly required. (2) Although we found previous studies that applied FDR, the control for FDR has not yet become the mainstream. For example, in the Working Group I report of the most

recent IPCC assessment (the 6th assessment report), the maps showing locally significant increase in surface temperature over time have not used field significance test corrected for FDR (e.g., Fig. 2.11 in Gulev et al. 2021). In a recent paper published on May 29 2024 in Nature (Han et al., 2024) the correction for FDR has not been used either.

Given that using field significant test corrected for FDR does not alter the robustness of our findings, we hope including both locally significant regressions and those additionally passing FDR control is acceptable. Otherwise, we can also show maps with only the test corrected for FDR.

The method to apply field significance test corrected for FDR has been briefly explained in the revised Methods (lines 677–683) as: *“Additionally, we applied a more rigorous field significance test corrected for the false discovery rate ($\alpha_{FDR}=0.1$) following Wilks 2016 (74). For observations showing moderate to strong spatial autocorrelation, the global significance level ($\alpha_{global}=0.05$) is about one half of α_{FDR} ⁷⁴. The test corrected for FDR was carried out using the stats.multitest module in the Python package statsmodels based on the Benjamini-Hochberg method. Pixels with locally significant regressions ($p<0.05$) and additionally passing the field significance test corrected for FDR, are both shown in our results.”*

Supplementary text:

[Comment 22] L30: Could mention that MTBS is also derived from Landsat.

[Response] We have provided this information in the revised Supplementary text (Lines 30–31) as: *“The third product used was the Monitoring Trends in Burn Severity (MTBS) dataset, which was based on Landsat images and covers both Alaska and the continental USA.”*

References

Alkama, R., Cescatti, A., 2016. Biophysical climate impacts of recent changes in global

- forest cover. *Science* 351, 600-604.
- Burrell, A.L., Evans, J.P., De Kauwe, M.G., 2020. Anthropogenic climate change has driven over 5 million km² of drylands towards desertification. *Nat Commun* 11, 3853.
- Chiang, F., Mazdidasni, O., AghaKouchak, A., 2021. Evidence of anthropogenic impacts on global drought frequency, duration, and intensity. *Nat Commun* 12, 2754.
- Gudmundsson, L., Seneviratne, S.I., Zhang, X., 2017. Anthropogenic climate change detected in European renewable freshwater resources. *Nature Climate Change* 7, 813-816.
- Gulev, S.K., P.W. Thorne, J. Ahn, F.J. Dentener, C.M. Domingues, S. Gerland, D. Gong, D.S. Kaufman, H.C. Nnamchi, J. Quaas, J.A. Rivera, S. Sathyendranath, S.L. Smith, B. Trewin, K. von Schuckmann, and R.S. Vose, 2021: Changing State of the Climate System. In *Climate Change 2021: The Physical Science Basis. Contribution of Working Group I to the Sixth Assessment Report of the Intergovernmental Panel on Climate Change* [Masson-Delmotte, V., P. Zhai, A. Pirani, S.L. Connors, C. Péan, S. Berger, N. Caud, Y. Chen, L. Goldfarb, M.I. Gomis, M. Huang, K. Leitzell, E. Lonnoy, J.B.R. Matthews, T.K. Maycock, T. Waterfield, O. Yelekçi, R. Yu, and B. Zhou (eds.)]. Cambridge University Press, Cambridge, United Kingdom and New York, NY, USA, pp. 287–422, doi:10.1017/9781009157896.004.
- Han, J., Liu, Z., Woods, R., McVicar, T.R., Yang, D., Wang, T., Hou, Y., Guo, Y., Li, C., Yang, Y., 2024. Streamflow seasonality in a snow-dwindling world. *Nature* 629, 1075-1081.
- Jain, P., Castellanos-Acuna, D., Coogan, S.C.P., Abatzoglou, J.T., Flannigan, M.D., 2021. Observed increases in extreme fire weather driven by atmospheric humidity and temperature. *Nature Climate Change* 12, 63-70.
- Li, Y., Zhao, M., Motesharrei, S., Mu, Q., Kalnay, E., Li, S., 2015. Local cooling and warming effects of forests based on satellite observations. *Nat Commun* 6, 6603.
- Liu, Z., Ballantyne, A.P., Cooper, L.A., 2019. Biophysical feedback of global forest

- fires on surface temperature. *Nat Commun* 10, 214.
- Portmann, R., Beyerle, U., Davin, E., Fischer, E.M., De Hertog, S., Schemm, S., 2022. Global forestation and deforestation affect remote climate via adjusted atmosphere and ocean circulation. *Nat Commun* 13, 5569.
- Rogers, B.M., Soja, A.J., Goulden, M.L., Randerson, J.T., 2015. Influence of tree species on continental differences in boreal fires and climate feedbacks. *Nature Geoscience* 8, 228-234.
- Scholten, R.C., Coumou, D., Luo, F., Veraverbeke, S., 2022. Early snowmelt and polar jet dynamics co-influence recent extreme Siberian fire seasons. *Science* 378, 1005-1009.
- Zhang, Y., Wang, X., Lian, X., Li, S., Li, Y., Chen, C., Piao, S., 2024. Asymmetric impacts of forest gain and loss on tropical land surface temperature. *Nature Geoscience* 17, 426-432.
- Zou, Y., Rasch, P.J., Wang, H., Xie, Z., Zhang, R., 2021. Increasing large wildfires over the western United States linked to diminishing sea ice in the Arctic. *Nat Commun* 12, 6048.
- Zscheischler, J., Seneviratne, S.I., 2017. Dependence of drivers affects risks associated with compound events. *Sci Adv* 3, e1700263.